# CD47-amyloid-β-CD74 signaling triggers adaptive immunosuppression in sepsis

Zhongxue Feng [ID] [1,8], Lijun Wang [ID] [1,8], Yang Li [ID] [1,8], Yonggang Wei[2], Yueyue Zhou[3], Siying Wang[1], Xiaoqi Zhang[4], Chunling Jiang[5], Xuelian Liao[6], Yan Kang [ID] [6✉], Fei Xiao [ID] [7✉] & Wei Zhang [ID] [1✉]

## Abstract

Sepsis is defined as life-threatening organ dysfunction caused by a dysregulated host response to infection. However, how this dysregulation occurs remains to be elucidated. In this study, we use single-cell RNA sequencing (scRNA-seq) and conventional RNA-seq to analyze the immune landscape of sepsis and observe that adaptive immunity is acutely and strongly suppressed. This systemic immunosuppression occurs not only in the peripheral blood but also in all other immune compartments, including the spleen, lymph nodes, and bone marrow. Clinical data show that these adaptive immunity-related genes may have the potential to be used to distinguish patients with sepsis from those with common infections. CD47 is found to play a pivotal role in this immunosuppression by inducing the production of amyloid-β (Aβ), which interacts with CD74 on B cells, leading to B-cell suppression and subsequent adaptive immunosuppression. Blocking CD47-Aβ signaling significantly reduces organ injury and improves the survival rate of septic mice by restoring phagocytic cell functions and alleviating B-cell suppression and adaptive immunosuppression.

Keywords Sepsis; Adaptive Immunosuppression; Amyloid-β (Aβ); CD47; SIRP
Subject Categories Immunology; Microbiology, Virology & Host Pathogen Interaction; Signal Transduction

## Introduction

Sepsis is defined as life-threatening organ dysfunction caused by a dysregulated host response to infection (Fleischmann-Struzek et al, 2020; Markwart et al, 2020; Rudd et al, 2020). The global incidence of sepsis is estimated to be 48.9 million, and the number of deaths related to sepsis is ~11 million, which is nearly twice the previous estimate (Rudd et al, 2020). In 2017, the World Health Organization (WHO) announced that developing methods for the prevention, diagnosis, and treatment of sepsis was the top priority of global medical and health work (Fleischmann-Struzek et al, 2020; Markwart et al, 2020; Rudd et al, 2020).

Previous studies on sepsis have focused mainly on inflammation and cytokine storms caused by infection. However, almost all drugs that target inflammatory factors have failed (Nedeva et al, 2019; Steinhagen et al, 2020), suggesting that inflammation may not fully reflect the immune status of the body during sepsis. The latest revision of the definition of sepsis was at the 3rd International Consensus Definitions for Sepsis and Septic Shock (Sepsis-3) meeting (Singer et al, 2016), in which the pathogenesis of sepsis was modified from "nonspecific inflammation" to "a dysregulated host response to infection" (Fleischmann-Struzek et al, 2020; Markwart et al, 2020; Rubio et al, 2019; Rudd et al, 2020; Singer et al, 2016). The task force concurred that the excessive attention given to inflammation may overlook other important immune changes and that the focus on sepsis should be shifted to the dysregulated host response (Ghnewa et al, 2020; Singer et al, 2016). However, essential questions remain to be answered: what is the immune landscape of this dysregulation, and how does it happen? The lack of a clear immune landscape in infection and sepsis generates ambiguities and conceptual challenges (Rubio et al, 2019).

In this study, we first used single-cell RNA sequencing (scRNA-seq) and bulk RNA-seq to outline the immune landscape of sepsis. We found that when sepsis occurred, adaptive immunity was acutely and strongly suppressed, and this immunosuppression occurred not only in the peripheral blood but also in other immune compartments, including the spleen, lymph nodes, and bone marrow. Clinical data indicated that these adaptive immunity-related genes may have the potential to be applied to distinguish patients with sepsis from those with common infections. We revealed that CD47, an immune checkpoint molecule, may play a pivotal role in triggering this immunosuppression. CD47 was highly expressed in newly emerging myeloid cells at sepsis onset. On the one hand, the binding of CD47 to its receptor SIRPα transmitted a "don't eat me" signal that weakened the phagocytic activity of myeloid cells. On the other hand, CD47-SIRPα signaling promoted the release of Aβ by myeloid cells, which interacted with

[1]Institute of Critical Care Medicine, State Key Laboratory of Biotherapy and Cancer Center, West China Hospital, Sichuan University, Chengdu, Sichuan, China. [2]Department of Liver Surgery, West China Hospital, Sichuan University, Chengdu, Sichuan, China. [3]Frontier Medical Center, Xin Chuan Road, Zhong He Street, 610212 Chengdu, Sichuan, China. [4]Department of Orthodontics, State Key laboratory of Oral Diseases, West China Hospital of Stomatology, Sichuan University, Chengdu, Sichuan, China. [5]Department of Anesthesiology, West China Hospital of Stomatology, Sichuan University, Chengdu, Sichuan, China. [6]Department of Critical Care Medicine, West China Hospital, Sichuan University, Chengdu, Sichuan, China. [7]Department of Intensive Care Unit of Gynecology and Obstetrics, West China Second University Hospital, Sichuan University, Chengdu, Sichuan, China. [8]These authors contributed equally: Zhongxue Feng, Lijun Wang, Yang Li. ✉E-mail: kangyan@scu.edu.cn; icuxiaofei@scu.edu.cn; zhangwei197610@wchscu.edu.cn

CD74 on B cells, leading to B-cell suppression and subsequent adaptive immunosuppression. Treatment with an anti-CD47 antibody, the amyloid-β inhibitor E2609, or CD47 gene knockout significantly improved the survival rate of septic mice. Our study may provide translational opportunities for the design of immunotherapy for sepsis.

# Results

## scRNA-seq to explore the changes in circulating immune cells in mice with mild or critical infection (sepsis)

Rather than simply comparing healthy controls with sepsis patients, our interest is in what happens to the host's immune system when a mild infection occurs. Why are most infections mild infections that do not cause severe symptoms, whereas sepsis causes not only multiple organ dysfunction but also death? A "mild infection group", namely, infection that does not cause severe symptoms or death, was inserted between the healthy control and sepsis groups to address these questions. Following this concept, we used the cecal ligation and puncture (CLP) model, which is widely used in sepsis studies (Hubbard et al, 2005), and divided the mice into sham, mild infection (0% mortality) and sepsis (50–70% mortality) groups according to the degree of infection (Fig. EV1; detailed experimental procedures in "Methods" section). The study design is shown in Fig. 1A.

Peripheral blood mononuclear cells (PBMCs) play an important role in the immune response to infection. We used 5' tag scRNA-seq based on the 10x Genomics platform to explore the changes in PBMCs isolated from mice with infections of different severities. After rigorous quality control (Appendix Fig. S1), a total of 32,409 cells were obtained. Unbiased graph-based clustering identified two major populations, namely, lymphocytes and myeloid cells (Appendix Fig. S2). All cells were further classified into 10 clusters according to the expression of cell markers reported in the literature (Fig. 1B,C; Appendix Fig. S3): $CD4^+$ T cells, $CD8^+$ naive T cells, $CD8^+$ effector T cells, naive B cells ($CD19^+CD27^-$), memory B cells, polymorphonuclear/monocytic myeloid-derived suppressor cells (PMN/M-MDSCs), monocytes/macrophages (Mos/Mφs), natural killer (NK) cells and platelets. The cell clusters were visualized by t-distributed stochastic neighbor embedding (t-SNE) (Fig. 1D) and uniform manifold approximation and projection (UMAP) nonlinear dimensionality reduction (Fig. 1E).

Compared with defining T/B cells, which have clear and generally accepted markers, defining myeloid cells is more complicated. According to the scRNA-seq results, the expression characteristics shared by all myeloid cells were $CD11b^+$, $CD14^+$, $CD16^+$, and $Cebpb^{hi}$ (Appendix Fig. S4). MDSCs are a large class of immunosuppressive myeloid-derived cells that play important roles in the immune response and have received special attention in many fields, particularly cancer and infection research. Myeloid cells are classified into three major categories: PMN-MDSCs, M-MDSCs, and Mos/Mφs. The common features of M-MDSCs and Mos/Mφs were $Ms4a4a^+$, $CD115^+$, and $F4/80^+$, and the molecular markers used to distinguish M-MDSCs from Mos/Mφs were F13a1 and Fabp4. F13a1 was specifically expressed in M-MDSCs, and Fabp4 was used to identify Mos/Mφs (Appendix Fig. S5).

Neutrophils were removed from the blood by standard Ficoll gradient centrifugation to isolate PBMCs (Appendix Fig. S6). The characteristic expression patterns of PMN-MDSCs (Appendix Fig. S7) were similar to those of neutrophils, except that PMN-MDSCs had altered buoyancy and appeared in the buffy coat together with other PBMCs (Bronte et al, 2016; Dumitru et al, 2012). Completely distinguishing PMN-MDSCs from neutrophils may not be possible, as they may be derived from similar progenitors but with different buoyancies and functions. Judging from the characteristic expression patterns of PMN-MDSCs (Appendix Fig. S7B,C), such as $S100a9^{hi}$ (Frohberger et al, 2020; Zhao et al, 2012) and $Il1r2^+$ (Schluter et al, 2018; Vambutas et al, 2009), PMN-MDSCs are likely to perform immunosuppressive functions, in contrast to the canonical role of neutrophils.

## The number of myeloid cells increases while the number of lymphocytes decreases during sepsis

As shown in Fig. 2A,B, the most noticeable changes were in the proportions of myeloid cells and lymphocytes. The t-SNE analysis (Fig. 2C) revealed that the numbers of all myeloid cells, including PMN-MDSCs, M-MDSCs and Mos/Mφs, were significantly increased, whereas the number of lymphocytes decreased as the degree of infection worsened. The proportion of myeloid cells increased from ~10 to 50%, whereas the proportion of lymphocytes decreased from 90 to 45% (Appendix Fig. S8).

The changes in the cell frequency were relative, and we used flow cytometry and microbead counters to calculate the absolute changes in the numbers of each cell population. As shown in Fig. 2D–F, the absolute number of myeloid cells increased in CLP mice, whereas the number of lymphocytes decreased significantly. $CD19^+$ B cells were the cell population with the most drastic changes in the number, with the mean number decreasing from ~1140 cells/μl to 126 cells/μl (Fig. 2G). No significant change was found in the numbers of the $CD4^+$ and $CD8^+$ T-cell subsets (Fig. 2H–J). These results suggest that the decrease in the number of lymphocytes was attributed mainly to the decrease in the number of $CD19^+$ B cells.

## The expression of MHC class II genes decreases during sepsis

The expression of MHC class II genes followed a similar pattern to that of T/B-cell-related genes (Appendix Figs. S9 and 10). In humans and mice, MHC genes are referred to as human leukocyte antigen (HLA) and H2 genes, respectively. The expression of the H2-Aa (corresponding to HLA-DQA), H2-Ab1 (HLA-DQB1), H2-DMa (HLA-DMA), H2-DMb (HLA-DMB), H2-Eb (HLA-DR), H2-Oa (HLA-DOA), and H2-Ob (HLA-DOB) genes decreased stepwise as the severity of infection worsened, and the H2 genes were expressed mainly in B cells rather than in myeloid cells (Appendix Fig. S10). Flow cytometry revealed that the median fluorescence intensity (MFI) of MHC class II molecules on B cells decreased by ~50%, whereas no significant change was detected in myeloid cells (Appendix Fig. S11), suggesting that the downregulation of MHC II molecules was due mainly to a reduction in the number of B cells expressing these molecules and, in part, to an ~50% reduction in the expression of these molecules on B cells.

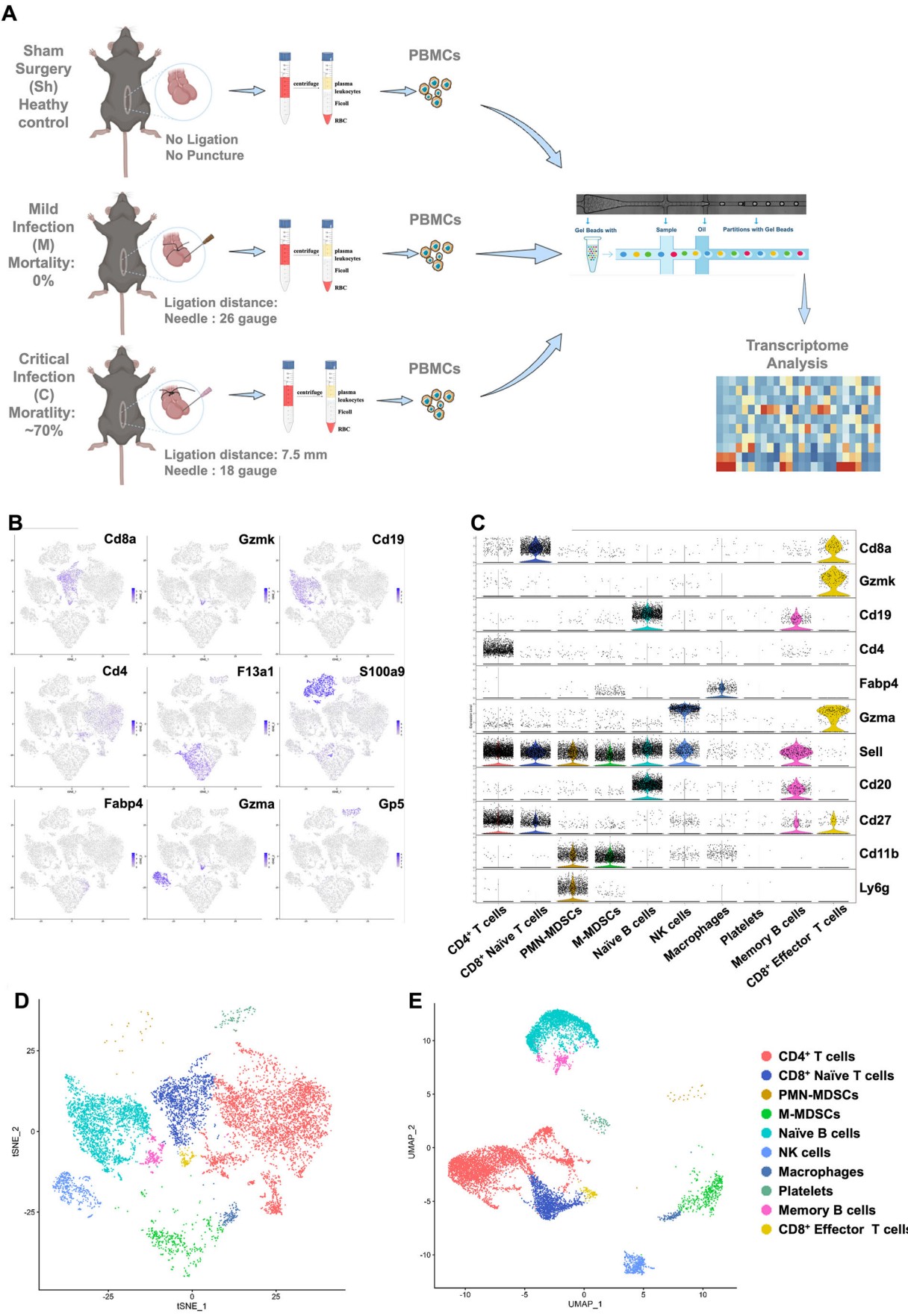

**Figure 1. Identification of clusters of PBMCs by scRNA-seq.**

(A) Study design: CLP mice were divided into 3 groups (Sham, Mild, and Critical) based on the length of the cecum ligation and the size of the puncture hole. Blood samples were collected 8 h after the CLP surgery, and neutrophils were discarded after Ficoll gradient centrifugation (1.080–1.085 g/l) to isolate PBMCs. PBMCs from three mice per group (Sham, Mild, Crit) were pooled and subjected to scRNA-seq. After rigorous quality control, a total of 32,409 PBMCs were obtained and analyzed using 5′ tag scRNA-seq with the 10x Genomics platform. (B) Canonical cell markers that were used to identify PBMC clusters: CD4+ T cells (CD3+CD4+), CD8+ naive T cells (CD3+CD8a+Sell+Ccr7+), CD8+ effector T cells (CD3+CD8a+Gzmk+) (Zhang et al, 2020), naive B cells (CD19+CD27−), memory B cells (CD19+CD27+) (Cao et al, 2020), polymorphonuclear myeloid-derived suppressor cells (PMN-MDSCs, CD11b+Ly6g+Ly6c$^{low}$S100a9$^{hi}$Il1r2+) (Bronte et al, 2016; Dumitru et al, 2012; Frohberger et al, 2020; Schluter et al, 2018; Vambutas et al, 2009; Zhao et al, 2012), monocytic myeloid-derived suppressor cells (M-MDSCs, CD11b+Ly6g−ly6c$^{hi}$CD115+Fabp4−F13a1+) (Bronte et al, 2016; Movahedi et al, 2008; Veglia et al, 2018), monocytes/macrophages (Mos/Mφs, CD11b+Ly6g−ly6c$^{hi}$CD115+Fabp4+F13a1−) (Breloer and Fleischer, 2008; Lechmann et al, 2002; Liao et al, 2020), natural killer (NK) cells (Gzma+) (Zhang et al, 2020), and platelets (Gp5+) (Lanza, 2006). (C) Violin plots displaying gene expression profiles derived from scRNA-seq results of pooled PBMCs from three mice per group. (D, E) t-SNE and UMAP plots showing the ten identified PBMC clusters.

## Adaptive immunosuppression is a systemic immune response in central to peripheral immune compartments

We then investigated whether, in addition to those from peripheral blood, immune cells from other immune compartments, including the central (bone marrow) and secondary lymphoid tissues (spleen and lymph nodes), exhibit similar changes in the expression of these genes. Peripheral blood, bone marrow, spleen, and lymph node samples were collected 8 h after the sham or CLP surgery. Bulk RNA-seq was performed to detect changes in gene expression in immune cells from all these tissues (Appendix Figs. S12 and 13). DEG and Gene Ontology (GO) enrichment analyses revealed that the most significantly enriched GO terms were related mainly to immune and defense responses to organisms (Appendix Table S1). The expression of T/B-cell-related genes and MHC class II genes decreased in a stepwise manner in central (bone marrow) and secondary lymphoid tissues (spleen and lymph nodes) and peripheral blood PBMCs (Fig. 3A–P).

The decreases in the expression of T/B-cell marker genes in the spleen, lymph nodes, and bone marrow suggested that the decrease in the number of lymphocytes was a systemic immune response. We performed bulk RNA-seq on kidney, liver, and lung tissues to analyze the expression profiles of immune cells. The FPKM values of T-cell-related genes, including CD3g, CD4, and CD8a, and B-cell-related genes, including Pax5 and CD19, were <1 in the kidney and liver but >1 in the lungs, indicating that T/B cell may mainly infiltrate the lungs, with limited presence in the liver and kidneys. On the other hand, as a pan marker of myeloid cells, the expression of CD11b in the kidney, liver, and lung increased stepwise (Appendix Fig. 14). These results suggest that T/B cells may mainly infiltrate the lungs when sepsis occurs, while myeloid cells migrate to all major organs, including the kidneys, liver, and lungs, and the degree of migration increased with the severity of infection.

*Klebsiella pneumoniae* (*K.p.*) is a gram-negative, oxidase-negative, rod-shaped bacteria and one of the most common clinical pneumonia pathogens. In addition to the CLP model, we established a pneumonia-induced sepsis model by injecting *K.p.* directly into the lungs of mice via tracheal intubation (Appendix Fig. S15). We collected peripheral blood samples from mice 8 h, 24 h, 3 d and 7 d after CLP surgery or *K.p.* administration to examine the expression of adaptive immunity-related genes in PBMCs. Real-time RT–PCR revealed that the expression levels of adaptive immunity-related genes continued to decrease at 24 h after CLP or *K.p.* infusion compared with those at 8 h (Fig. EV2). The results showed that adaptive immune suppression in the CLP- or *K.p.*-induced sepsis model was most obvious at ~24 h. Interestingly,

the expression of adaptive immunity-related genes recovered after 7 days compared with 24 h, regardless of CLP or *K.p.* infusion, suggesting that the adaptive immune system of these mice was in a state of recovery.

## Certain adaptive immunity-related genes may have the potential to differentiate patients with sepsis from those with common infections

The abovementioned genes exhibiting the most drastic changes in expression, such as Pax5/CD19 for B-cell functions, CD8A for T cells, and HLA genes for antigen presentation, may be used as auxiliary indicators for the diagnosis of infection or to distinguish patients with common infection from those with sepsis. We collected blood samples from 20 healthy volunteers and 51 patients with newly diagnosed infections to test this hypothesis. The entry criterion for enrollment was that the patient was diagnosed with an "infection". The detailed information of all participants was extracted later according to their hospital number. The characteristics of the patients are listed in Dataset EV1, including a brief case history, SOFA score, blood pressure, WBC counts and other information when the patients were admitted to the emergency department, as well as their progress and outcome. Eleven of the 51 patients with infection were diagnosed with sepsis, 5 of whom were diagnosed in the emergency room, and 6 of whom were hospitalized in a specialized ward. The diagnosis of sepsis was made after a SOFA score was calculated for the patient after sufficient clinical and laboratory results had been collected, including PaO2/FiO2 (mmHg), platelet count (×10$^9$/L), bilirubin level (μmol/L), mean arterial pressure, GCS, and creatinine level (μmol/L).

Expression levels of the target genes were detected using droplet digital PCR (copies/μl). The number of mRNA copies of the B-cell-related genes Pax5, CD5, and CD19; T-cell-related gene CD8A; and MHC-related genes CIITA, HLA-DQ1, HLA-DR, and HLA-DO were significantly decreased in patients with infections. The receiver operating characteristic (ROC) curve revealed that these genes may be used to distinguish septic patients from those with common infections, with area under the curve (AUC) values ranging from 0.85 to 0.97 (Fig. 4A–H). As a canonical marker of myeloid cells, CD11b expression increased when the healthy control group and the common infection group were compared, but did not differ significantly between patients with common infection and sepsis (Fig. 4I). We also included 10 patients with non-infectious post-traumatic/operative illnesses who had SOFA scores >2 (ranging from 3 to 8), and found that the immune-related

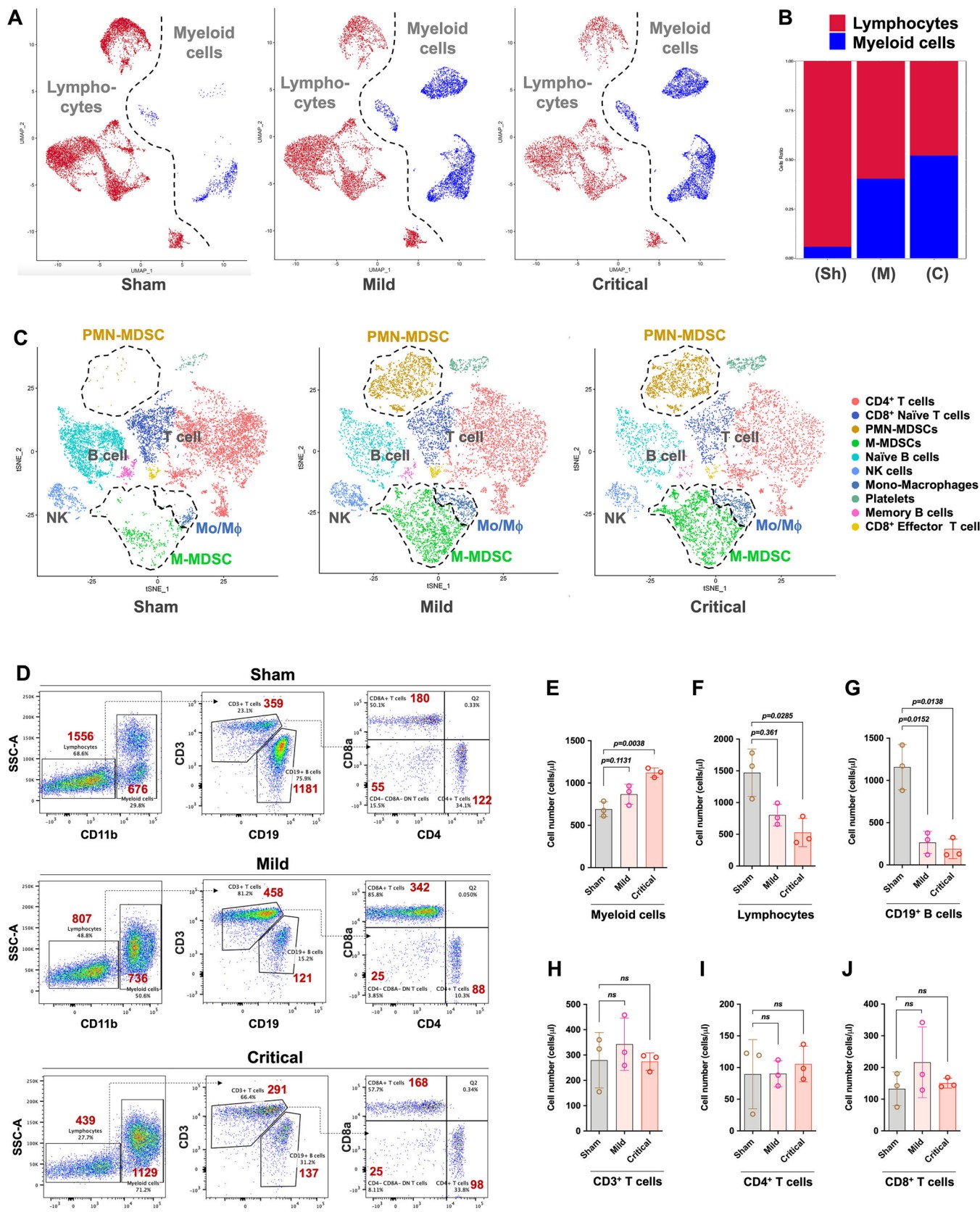

◄ **Figure 2. During sepsis, the number of myeloid cells increases, whereas the number of lymphocytes decreases.**

(**A, B**) UMAP analysis showing the changes in the proportions of lymphocytes and myeloid cells as the severity of infection worsened. (**C**) t-SNE analysis showing a stepwise decrease in the number of myeloid cells (including PMN-MDSCs, M-MDSCs, and Mos, framed by the dashed line) and an increase in the number of lymphocytes (including T/B cells) as the severity of infection worsened. (**D**) Flow cytometry data showing the ratio of each population (gating: CD11b⁺ myeloid cells, CD11b⁻CD19⁺ B cells, CD11b⁻CD3⁺CD4⁺ T cells, CD11b⁻CD3⁺CD8⁺ T cells, and CD11b⁻CD3⁺CD4⁻CD8⁻ double-negative T cells); the absolute cell number was labeled next to the corresponding cell population. (**E–J**) Quantification of the number of cells in each population. Data are presented as mean ± SD ($n = 3$, biological replicates). Significance was assessed by the two-tailed parametric Welch's $t$ test. The exact $P$ values are shown within the graphs.

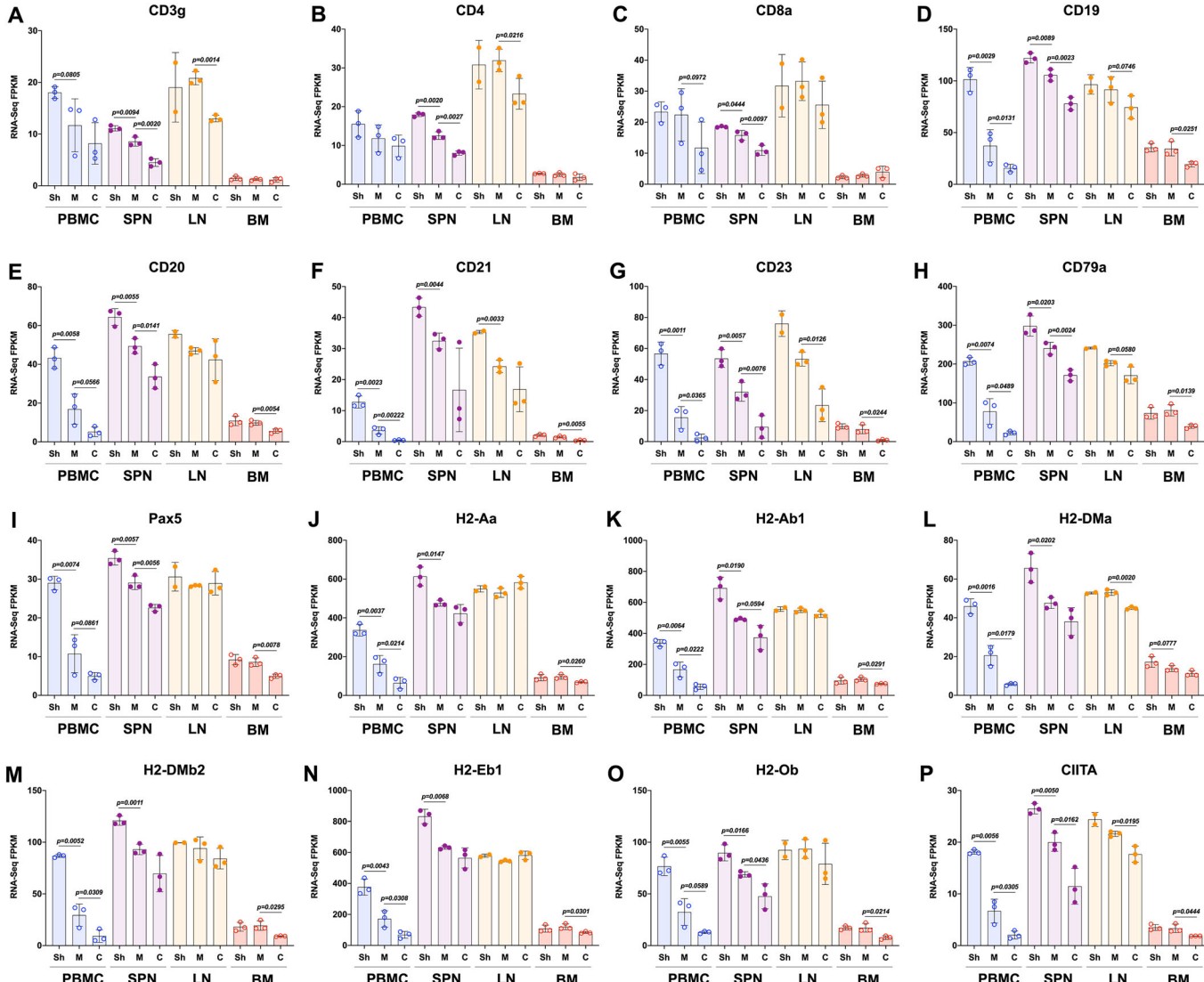

**Figure 3. Adaptive immunosuppression is a systemic immune response.**

(**A–P**) Samples from immune compartments, including the peripheral blood (PBMCs), bone marrow (BM), spleen (SPN), and lymph node (LN), were collected 8 h after the sham or CLP surgery and analyzed by RNA-seq (3 parallel samples per group). The FPKM (Fragments Per Kilobase of transcript per million mapped reads) value corrects for the sequencing depth and gene length and is used to represent the expression level of a transcript. The abscissa shows the sample sources from animals with different severities of infection (Sh, sham; M, mild infection; and C, critical infection), and the ordinate shows the FPKM value of a chosen gene (reported as the means ± SDs). Data are presented as mean ± SD ($n = 3$, biological replicates). The adjacent columns were compared using one-tailed Welch's $t$ test, and $P$ values deemed significant are shown between the adjacent groups.

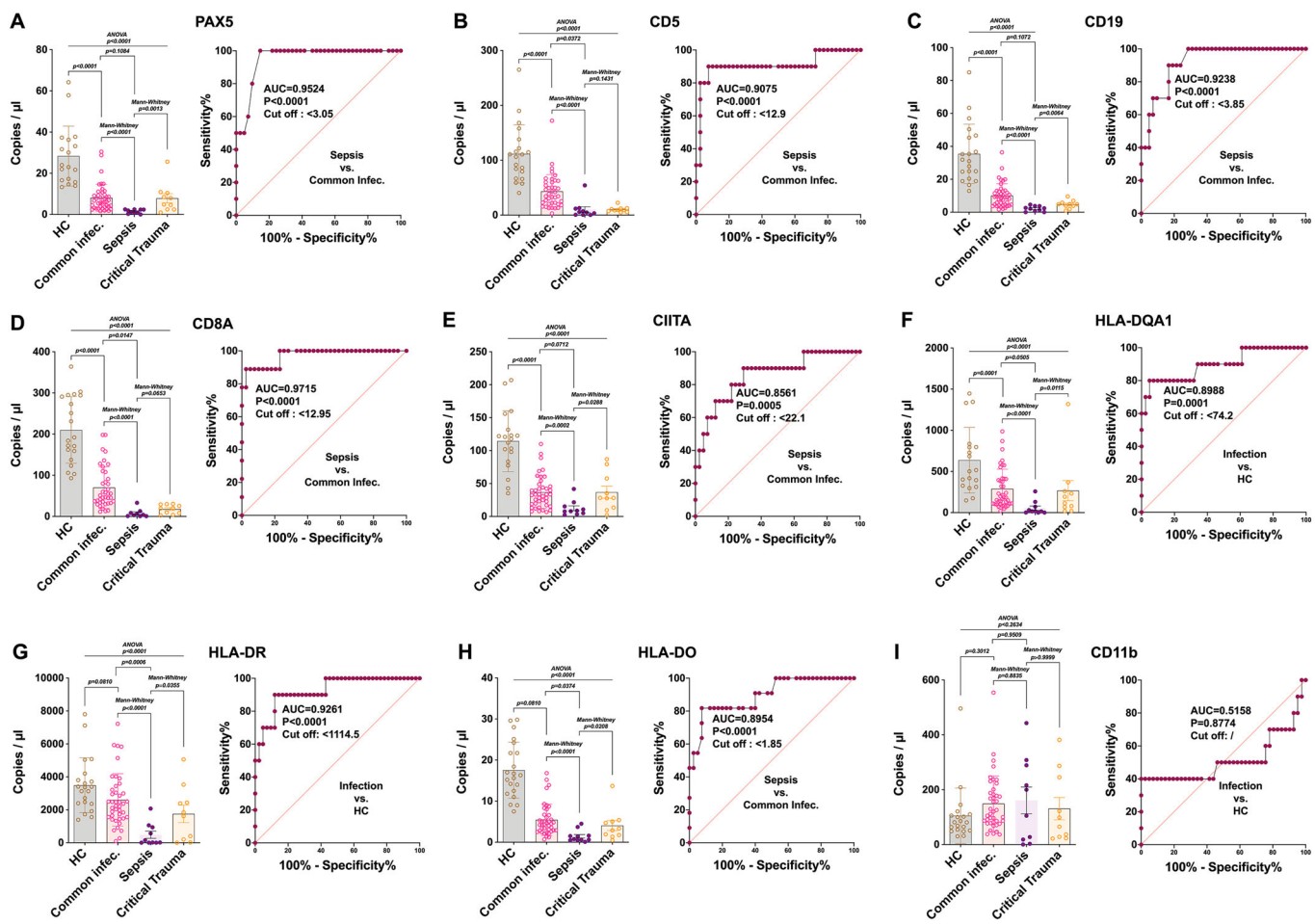

**Figure 4. Certain adaptive immunity-related genes may be used to distinguish patients with common infections from patients with sepsis.**

(A–I) The number of mRNA copies of Pax5, CD5, CD19, CD8A, CIITA, HLA-DQ1, HLA-DR, HLA-DO, and CD11b were measured by droplet digital PCR. Left panels: Scatter bar with SD showing the copy number (copies/µl) of each candidate gene in healthy volunteers ($n = 20$), patients with common infections ($n = 40$), patients with sepsis ($n = 11$), and patients with non-infectious post-traumatic/operative illness ($n = 10$). Significance is assessed by one-way ANOVA followed by Tukey's multiple comparison test. Because the datasets from the common infection group, sepsis group, and critical trauma group did not pass the normality test, the two adjacent groups were compared using the Mann–Whitney U test. The exact P values are shown above the bars. Right panels: ROC curve analysis to evaluate the diagnostic performance of a candidate gene for differentiating patients with common infection from patients with sepsis. The AUC values, P values, and cutoff values are annotated on the ROC curve. Source data are available online for this figure.

gene expression profiles of these post-traumatic/operative patients were similar to those of patients with common infections, except that the expression levels of CD5 and CD8A were low, similar to those of patients with sepsis (Fig. 4A–I).

## CD47 may play an important role in adaptive immunosuppression in sepsis

We explored the mechanism behind this rapid and widespread adaptive immunosuppression in sepsis. Immune checkpoint molecules, such as PD-1, are involved in cancer immunosuppression and are anticipated to play a role in immunosuppression in sepsis. However, scRNA-seq revealed fewer than 30 PD-1-positive cells among ~10,000 PBMCs, with no significant increase during mild or critical infections (Fig. 5A,B). Validation using dataset GSE151263 (Jiang et al, 2020; Data ref: Jiang et al, 2020) showed very low PD-1/PD-L1 expression in human PBMCs (Fig. EV3A).

Similarly, the expression of classic immunosuppressive factors, CTLA-4 and IL-10, is also low, aligning with the patterns observed for PD-1/PD-L1 (Fig. EV3B–D). Based on the low expression level and changing trend, these genes are unlikely to cause immunosuppression in sepsis.

We hypothesized that if a gene capable of causing such widespread and strong immunosuppression, it should meet three conditions: first, it should be widely expressed in immune cells in PBMCs, lymph nodes, spleen, and bone marrow, and its expression level should be sufficiently high; otherwise, it will not be able to produce the wide range of immunosuppression observed. Second, its characteristic expression pattern should be opposite those of adaptive immune-related genes. Third, it should perform immunosuppressive functions, similar to the characteristics of immune checkpoint molecules.

After screening our sequencing data, we identified a candidate gene, CD47, that met the above criteria. CD47 was expressed in

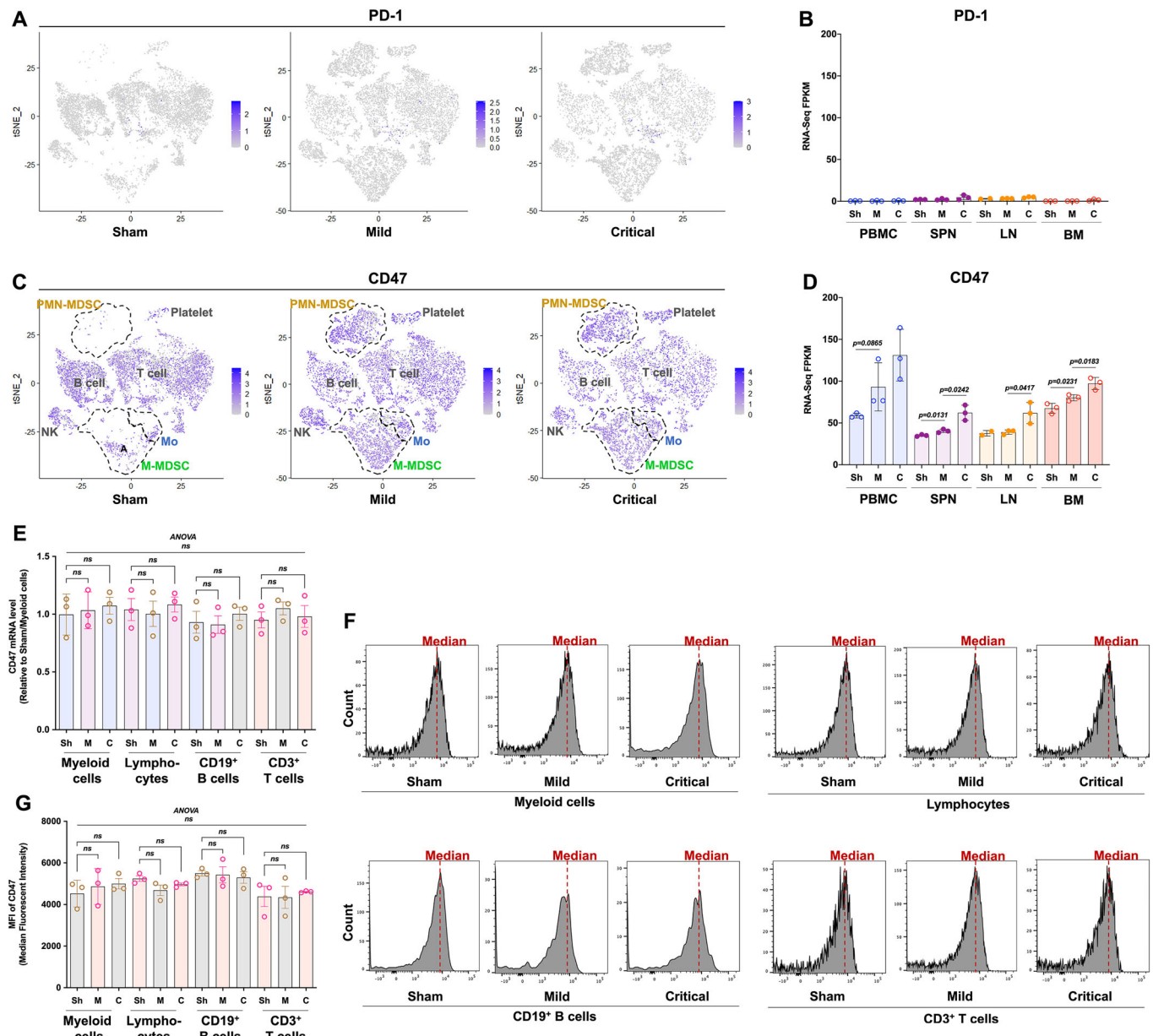

**Figure 5. CD47 is a potential mediator of adaptive immunosuppression in subjects with sepsis.**

(A, C) t-SNE plots showing the expression profiles of PD-1 and CD47 in PBMCs. (B, D) Scatter bar showing FPKM values of PD-1 and CD47 in immune compartments (PBMCs, LNs, SPN, and BM). Data are presented as mean ± SD (n = 3, biological replicates). The adjacent columns were compared using one-tailed Welch's t test, and P values deemed significant are shown between the adjacent groups. (E) CD47 mRNA levels in different cell populations (myeloid cells, lymphocytes, CD19+ B cells, and CD3+ T cells) are shown as mean ± SD (n = 3, biological replicates). (F) Flow cytometry data showing the MFI values of different cell populations (myeloid cells, lymphocytes, CD19+ B cells, and CD3+ T cells). (G) The values of MFI are shown as mean ± SD (n = 3, biological replicates). Significance in (E, G) was assessed by one-way ANOVA followed by Tukey's multiple comparison test. The exact P values are shown above the bars. Source data are available online for this figure.

almost all types of immune cells (Fig. 5C), and its expression characteristics were opposite those of genes related to adaptive immunity. The stepwise increases in CD47 expression were significant in the spleen, lymph nodes, bone marrow, and PBMCs (Fig. 5D).

We performed real-time RT‑PCR and flow cytometry to analyze the expression profile of CD47. Myeloid cells, B cells, and T cells were sorted using CD11b, CD19, and CD3, and the mRNA level of

CD47 was detected using real-time RT‑PCR (Fig. 5E). The expression of CD47 on the surface of each cell population was detected using flow cytometry, where the MFI represents the expression level of CD47 on the surface of different cell subsets (Fig. 5F,G). Both the CD47 mRNA and protein expression levels were not significantly different in different cell populations, and no significant changes in different cell populations were found when sepsis occurred (Fig. 5E–G). Thus, the increase in the overall CD47

mRNA level detected via RNA-seq is likely due to an increase in the number of cells expressing CD47 rather than an increase in CD47 expression in a certain cell type. Considering the high expression of CD47, which is widespread in almost all immune cells and thus it is regarded as a new immune checkpoint molecule and a novel target in cancer immunotherapy studies (Li, 2019; Logtenberg et al, 2020; Takimoto et al, 2019; Veillette and Tang, 2019), we speculated that CD47 may play an important role in adaptive immunosuppression in sepsis.

## CD47-SIRPα signaling reduces the phagocytotic activity of myeloid cells

CD47 is composed of five transmembrane regions and a cytoplasmic domain, and its main receptor is SIRPα. The binding of CD47 and SIRPα transmits a "don't eat me" signal to prevent CD47-expressing cells from being phagocytosed by myeloid cells expressing SIRPα (Barclay and Van den Berg, 2014; Logtenberg et al, 2020). Interestingly, although CD47 is widely expressed, its receptor, SIRPα, is specifically expressed in CD11b$^+$ myeloid cells but not in other cells (Fig. 6A). These findings suggest that the activation of the CD47-SIRPα signaling pathway may weaken the phagocytic function of myeloid cells and that newly emerged myeloid cells that simultaneously express CD47 and SIRPα may not be able to perform phagocytic functions properly.

We sorted myeloid cells using anti-CD11b$^+$ magnetic beads and tested their phagocytic and efferocytotic functions using FITC-conjugated *Escherichia coli* and apoptotic Jurkat cells. Myeloid cells from septic mice presented severely impaired phagocytotic function, and treatment with the anti-CD47 beads significantly restored the phagocytic (Fig. EV4) and efferocytotic function (Appendix Fig. S16) of myeloid cells. A CD47-KO mouse strain was generated (Appendix Fig. S17) to further investigate whether CD47 affects the clearance of bacteria in the blood. No colonies were found in sham or mild infection groups, but up to 10$^7$ CFUs were detected in the CLP group. Anti-CD47 antibody treatment markedly reduced bacterial load, and CD47-KO mice post-CLP showed similar effects (Fig. 6B,C).

We tested major organs (kidney, liver, lung) and found CLP-induced systemic infection likely caused bacterial spread. Blood agar culture showed high bacterial levels in CLP group organs compared to sham, with minimal bacteria in sham lungs, likely due to colonization. Anti-CD47 antibodies significantly reduced bacterial load (Appendix Fig. S18), suggesting CLP-induced abdominal infection spreads bacteria to other organs. This indicates the host's battle against pathogens is not limited to the initial infection site but may spread to major organs, including the lungs, liver, and kidneys, and cause multi-organ dysfunction.

Neutrophils dominate peripheral blood leukocytes post-bacterial infection, while PBMCs lack neutrophils due to Ficoll centrifugation (Appendix Fig. S6). Thus, neutrophils were excluded from scRNA-seq to avoid masking gene expression changes in other cells. Whole-blood analysis classified myeloid cells into Ly6G$^{high}$Ly6C$^{high}$ neutrophils, Ly6G$^{high}$Ly6C$^{medium}$ PMN-MDSCs, and Ly6G$^-$ monocyte-lineage cells, which were further divided into Ly6C$^+$F4/80$^+$ M-MDSCs and Ly6C$^-$F4/80$^+$ Mos/Mφs (Appendix Fig. S19). Flow cytometry showed CLP surgery significantly increased neutrophils, PMN-MDSCs, and M-MDSCs, and Anti-CD47 treatment reduced PMN-MDSCs and M-MDSCs but not neutrophils or Mos/Mφs, potentially enhancing bacterial clearance (Fig. 6D–G).

## CD47 induces the production of amyloid-β, which leads to B-cell suppression

Data showed that drastically downregulated adaptive immunity genes were mainly B-cell-related and MHC class II genes. scRNA-seq cell–cell interaction analysis identified two candidate genes likely mediating MDSCs-B-cell signaling: amyloid-β precursor protein (APP) in MDSCs and CD74 in B cells. The algorithm predicted high communication probability between APP and CD74 (Fig. 7A). APP has been reported to interact with CD74 directly (Matsuda et al, 2009) and mediates endothelial cell-macrophage communication (Liu et al, 2024). The interaction between APP and CD74 was verified by immunoprecipitation and western blot (IP-WB) assays (Fig. 7B).

scRNA-seq results showed CD74 was expressed in all B cells and a subset of myeloid cells, aligning with MHC class II gene profiles (Fig. 7C). Interestingly, APP expression was limited to myeloid cells (Fig. 7D), which was almost identical to SIRPα and CD11b profiles (Fig. EV4). This suggests CD47-SIRPα signaling may induce APP expression in myeloid cells, with APP cleavage products potentially interacting with CD74 on B cells, leading to B-cell suppression.

We measured APP mRNA in myeloid cells and its proteolytic product, amyloid-β (Aβ), in culture medium. RT-PCR showed higher APP mRNA in CLP-derived myeloid cells versus controls (Fig. 7E), and ELISA revealed elevated Aβ1-42 in culture medium of CLP-derived myeloid cells (Fig. 7F). This indicated CLP-derived myeloid cells produced more APP and Aβ than normal cells. Given APP and SIRPα's myeloid-specific expression (Figs. 7D and EV4), we hypothesized CD47-SIRPα signaling upregulates APP. Anti-CD47 antibody markedly reduced APP mRNA and Aβ levels (Fig. 7E,F), suggesting CD47 signaling promotes Aβ production.

We explored whether CD47-Aβ signaling suppresses B cells. Pax5 and Ciita are key transcription factors that regulate B-cell-related genes. Pax5 controls ~170 genes critical for B-cell signaling, differentiation, and maturation; its reduction can halt or reverse B-cell differentiation, reducing B-cell numbers and impairing function (Cobaleda et al, 2007; Hagman and Lukin, 2007; LeBien and Tedder, 2008; Revilla et al, 2012). Ciita induces the de novo transcription of MHC class II genes and increases constitutive MHC class I gene expression (Devaiah and Singer, 2013). Our results suggest that myeloid cells that arise during sepsis may secrete Aβ, suppressing B cells by inhibiting Pax5 and Ciita expression in an endocrine/paracrine manner.

We tested this hypothesis using a Transwell system. CD11b$^+$ myeloid cells were cultured in the upper chamber, and B cells in the lower chamber. Coculture with CLP-derived myeloid cells significantly suppressed Pax5/CD19 (B-cell-related) and Ciita/H2-Aa (MHC-related) gene expression in normal B cells, similar to B cells isolated directly from CLP mice (Fig. 7G–J). Since the chambers were separated, this suppression should be mediated by secreted factors, consistent with Aβ's role as a secreted molecule.

We used E2609 (elenbecestat), a β-secretase (BACE) inhibitor developed for Alzheimer's disease, to block Aβ production. ELISAs confirmed E2609 inhibited Aβ production in CLP-derived myeloid cells (Fig. 7K). This effect was also observed in vivo: systemic administration of E2609 (6.5 mg/kg, i.p.) or anti-CD47 (2.5 mg/kg,

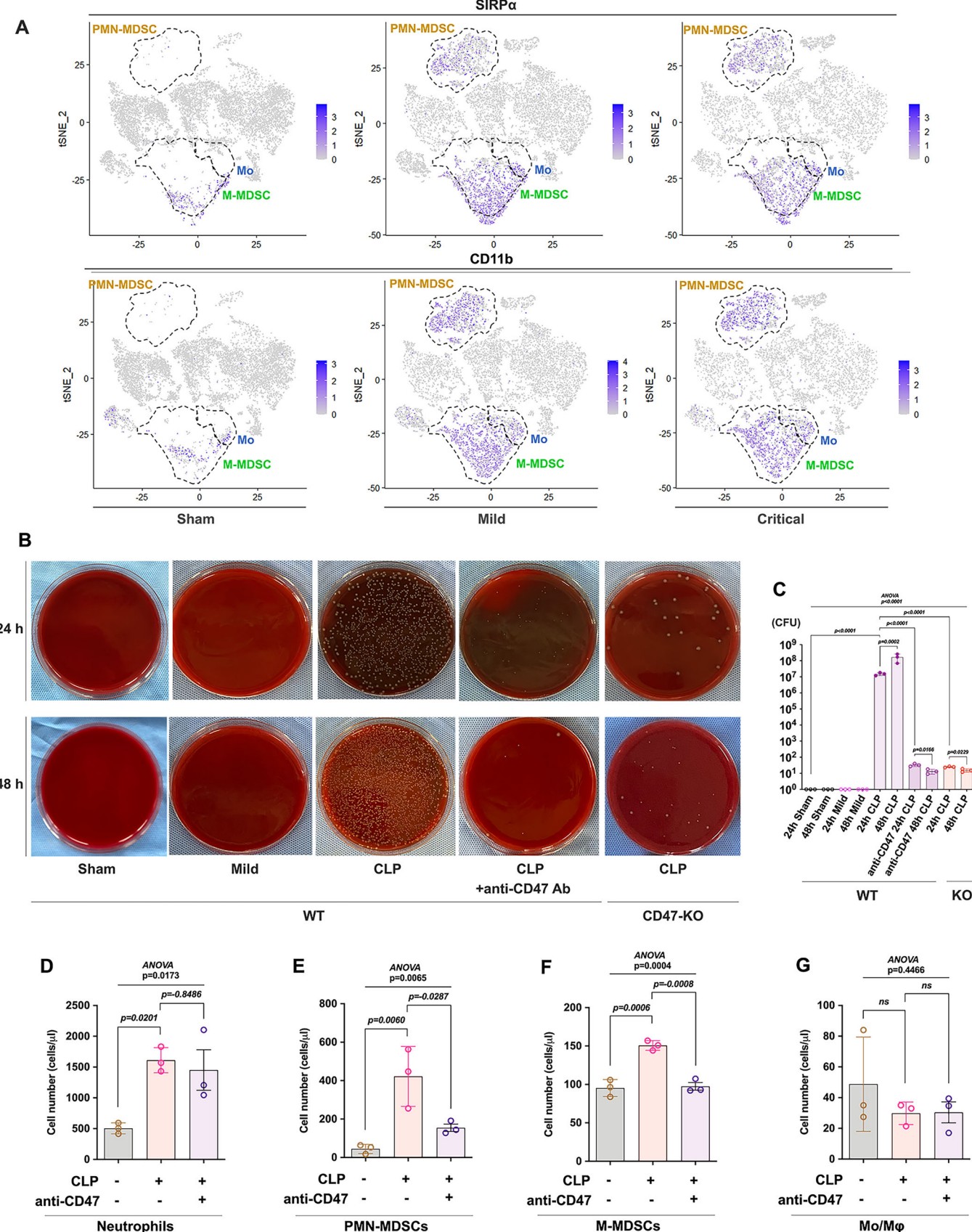

◀ **Figure 6. CD47-SIRPα signaling reduces the phagocytotic activity of myeloid cells.**

(A) The scRNA-seq results revealed that the expression profiles of SIRPα and CD11b were almost identical. (B) Blood samples were collected from WT mice that received sham, mild CLP, or critical CLP surgery or from CD47-KO mice that received critical CLP surgery, and the sample collection time was 24 h or 48 h after surgery. Blood agar plates were used to detect the bacterial load in the blood. (C) The bacterial load (CFUs) in each group was quantified. (D–G) Quantification of the number of cells in each population. Data are shown as mean ± SD ($n = 3$, biological replicates) and were assessed by one-way ANOVA followed by Tukey's multiple comparison test. The exact $P$ values are shown above the bars. Source data are available online for this figure.

i.v.) reduced peripheral blood Aβ levels in CLP mice to levels comparable to or below those of healthy controls (Fig. 7L).

Anti-CD47 antibody or E2609 was injected 1 h post-CLP surgery, and immune cells were isolated 8 h later from blood, spleen, lymph nodes, and bone marrow to assess whether blocking CD47 or Aβ could restore B-cell- and MHC-related gene expression in vivo. In CLP mice, transcription of B-cell-related and MHC class II genes was severely reduced. Anti-CD47 or E2609 treatment partially restored transcription of these genes (Fig. EV5). Flow cytometry showed both treatments significantly increased B-cell numbers in CLP mice (Appendix Fig. S20).

## Blocking CD47-Aβ signaling reduces organ injury, lowers inflammatory mediator levels, and significantly improves survival of septic mice

These results suggest blocking CD47-Aβ signaling may alleviate B-cell suppression, improving adaptive immunosuppression and potentially treating sepsis. We tested this using an anti-CD47 monoclonal antibody (Clone MIAP301). In the CLP group (IgG isotype injection), 6 of 20 mice survived 15 days post-surgery (30% survival). Anti-CD47 antibody (2.5 mg/kg, single i.v. injection) administered 8 h post-surgery increased survival to 80% (16 of 20 mice) (Fig. 8A). We also tested the Aβ inhibitor E2609 and CD47-KO mice in the CLP model. E2609-treated WT mice (6.5 mg/kg, twice/day for 3 days, i.p.) and CD47-KO mice showed significantly higher survival than controls, though slightly lower than the anti-CD47 group (Fig. 8A). Similar therapeutic effects were observed in a K.p.-induced sepsis model, though slightly less effective than in the CLP model, possibly due to the greater bacterial "shock" from a single large infusion (Fig. 8B). CD74-KO mice undergoing CLP surgery showed a lower survival rate, supporting the hypothesis that CD74 blockade leads to B-cell suppression (Appendix Fig. S21).

We analyzed pathological changes in the lungs and kidneys, organs most prone to acute injury (ALI and AKI). Whole-mount scans of H&E-stained sections were performed to avoid bias. CLP mice showed widespread lung and kidney damage. In the lungs, alveolar edema, inflammatory cell infiltration, and thickened alveolar interstitium were observed (Fig. 8C,D). In the kidneys, renal tubules were dilated, brush borders were thin or absent, and debris and vacuoles were present in the tubular lumens (Fig. 8E,F). Anti-CD47 antibody or E2609 treatment reduced lung inflammation, alleviated alveolar exudate, and improved alveolar interstitium thickness. Renal tubular dilation was reduced, brush borders were partially restored, and debris decreased. CD47-KO mice post-CLP showed similar improvements to those treated with anti-CD47 or E2609 (Fig. 8C–F).

Histological changes aligned with blood urea nitrogen (BUN), creatine (CREA), uric acid (UA), alanine transaminase (ALT), aspartate aminotransferase (AST), and fibrinogen levels, indicating

reduced acute organ injury after CD47-Aβ signaling blockade (Appendix Fig. 22A–F). Systemic inflammation was assessed by measuring TNF-α, IL-1β, IL-6, and MCP-1, which are hallmark inflammatory mediators of sepsis. ELISA showed CLP surgery significantly elevated serum levels of these cytokines. Anti-CD47 antibody or E2609 treatment significantly reduced their levels. CD47-KO mice exhibited similar reductions, consistent with histological findings (Appendix Fig. 22G–J).

## Discussion

The definition of sepsis was substantially revised in Sepsis-3 (2016), removing SIRS from the diagnostic criteria and shifting the core pathogenesis from "nonspecific inflammation" to "dysregulated host response to infection"(Singer et al, 2016). Adaptive immunosuppression is an important component of this dysregulated response, but how it occurs has not been clearly elucidated. Our data show that when sepsis occurs, adaptive immunity is acutely and strongly suppressed, marked by increased MDSCs, decreased lymphocytes, and downregulated adaptive immune genes. CD47 is pivotal: its receptor SIRPα, expressed on myeloid cells, weakens phagocytosis when bound. Concomitantly, CD47-SIRPα signaling promotes Aβ release from myeloid cells, which causes severe B-cell suppression and subsequent adaptive immunosuppression. Blocking CD47-Aβ signaling significantly reduces organ injury and improves the survival of septic mice by alleviating B-cell suppression.

When sepsis occurs, both T and B cells are reduced in number and impaired in function (Wang et al, 2022). T/B-cell immunodeficient mice (e.g., Rag$^{-/-}$, SCID, nude) show markedly worse sepsis survival (Feng et al, 2023; Joshi et al, 2012; Kadowaki et al, 2013; Kelly-Scumpia et al, 2011; Reim et al, 2009; Shelley et al, 2003). Interestingly, in critically patients with severe trauma leading to multiple organ dysfunction (SOFA > 2), some similar characteristics were also shown, such as a significant decrease in T/B-cell-related markers including CD8A/CD5/CD19, suggesting that trauma, like severe infection, may cause acute suppression of adaptive immunity, making the host more vulnerable to pathogens and further worsening the patient's condition. Some immunosuppressive factors, such as IL-10, have been reported to be involved in the immunosuppression of sepsis (Zhou et al, 2024). However, according to the scRNA-Seq results of the CLP mouse model and the human scRNA-Seq public dataset, the mRNA level of IL-10 in PBMCs of healthy volunteers and sepsis patients was low and did not show significant differences. This may be because IL-10, as a secretory cytokine, is released into the blood upon sepsis onset but not continuously produced in immune cells, so obvious changes in the mRNA level were not observed. The possible role of immunosuppressive factors such as IL-10 in sepsis requires more research.

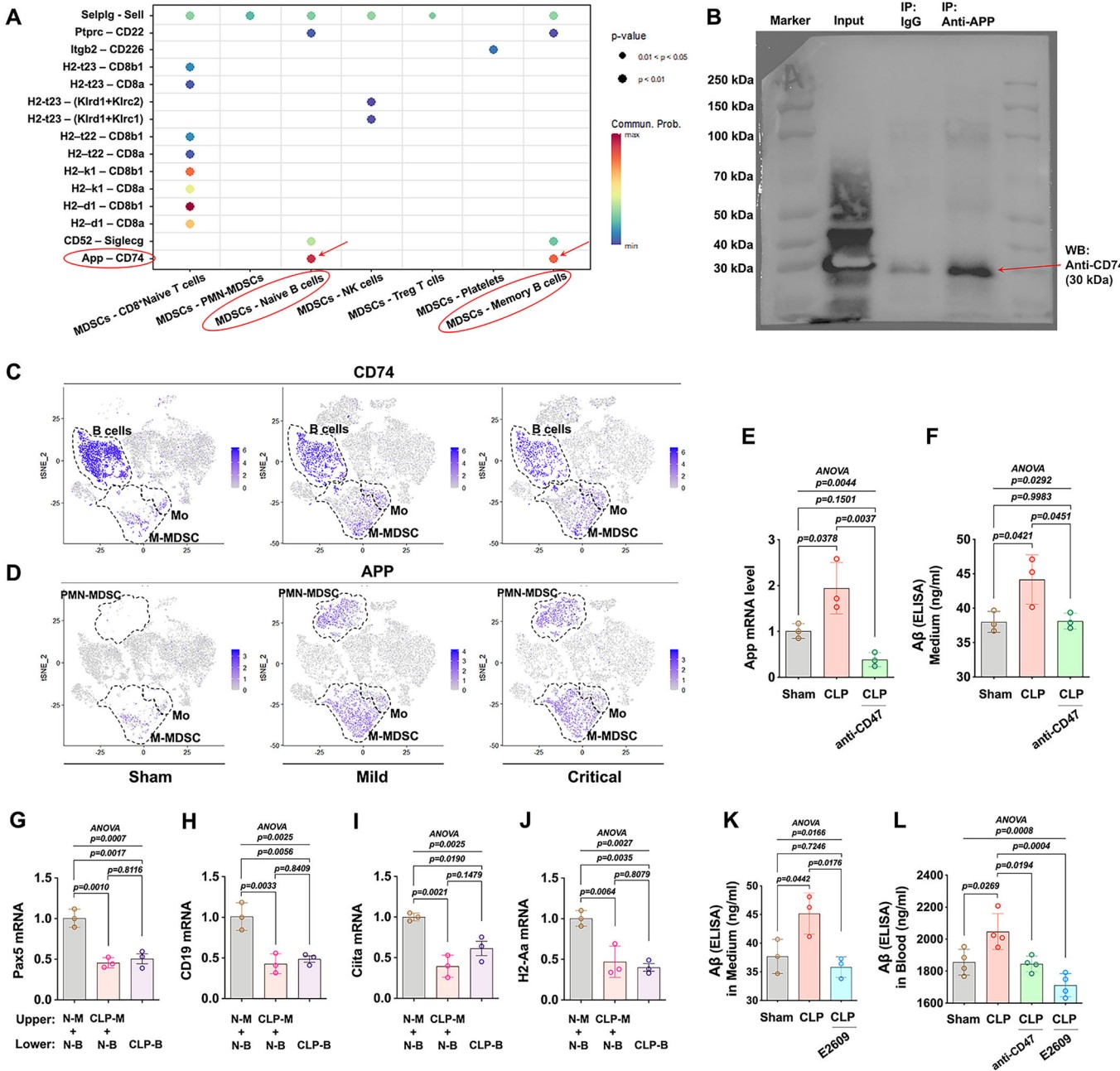

**Figure 7. CD47 induces the production of amyloid-β, which leads to B-cell suppression.**

(A) Cell–cell interaction analysis showing the predicted interaction between App from MDSCs and CD74 from B cells. (B) IP-WB assay: the cell lysates were immunoprecipitated with an anti-App antibody and probed with an anti-CD74 antibody. The input lane was loaded with the cell lysate that was not incubated with any antibody. The IgG lane was loaded with cell lysate and incubated with an IgG isotype control antibody as a negative control. (C, D) t-SNE plots showing the expression profiles of CD74 and App in PBMCs. (E) The mRNA level of App in myeloid cells and (F) the protein concentration of Aβ1–42 (ng/ml) in the blood of control mice, CLP mice or CLP mice treated with the anti-CD47 antibody. (G–J) Transwell assay: Myeloid cells were cultured in the upper chamber, and B cells were cultured in the bottom chamber. N-M myeloid cells from normal mice, CLP-M myeloid cells from CLP mice, N-B B cells from normal mice, CLP-B B cells from CLP mice. The mRNA levels of CD19, Pax5, H2-Aa, and Ciita in B cells from the bottom chamber were measured using real-time RT–PCR. (K, L) ELISA was performed to measure the concentration of the Aβ1-42 peptide in the culture medium of myeloid cells (in vitro) and in the blood of healthy or CLP mice treated with the anti-CD47 antibody or E2609 (in vivo). Data of (E–L) are shown as mean ± SD ($n = 3$, biological replicates). Significance was assessed by one-way ANOVA followed by Tukey's multiple comparison test. The exact $P$ values are shown above the bars. Source data are available online for this figure.

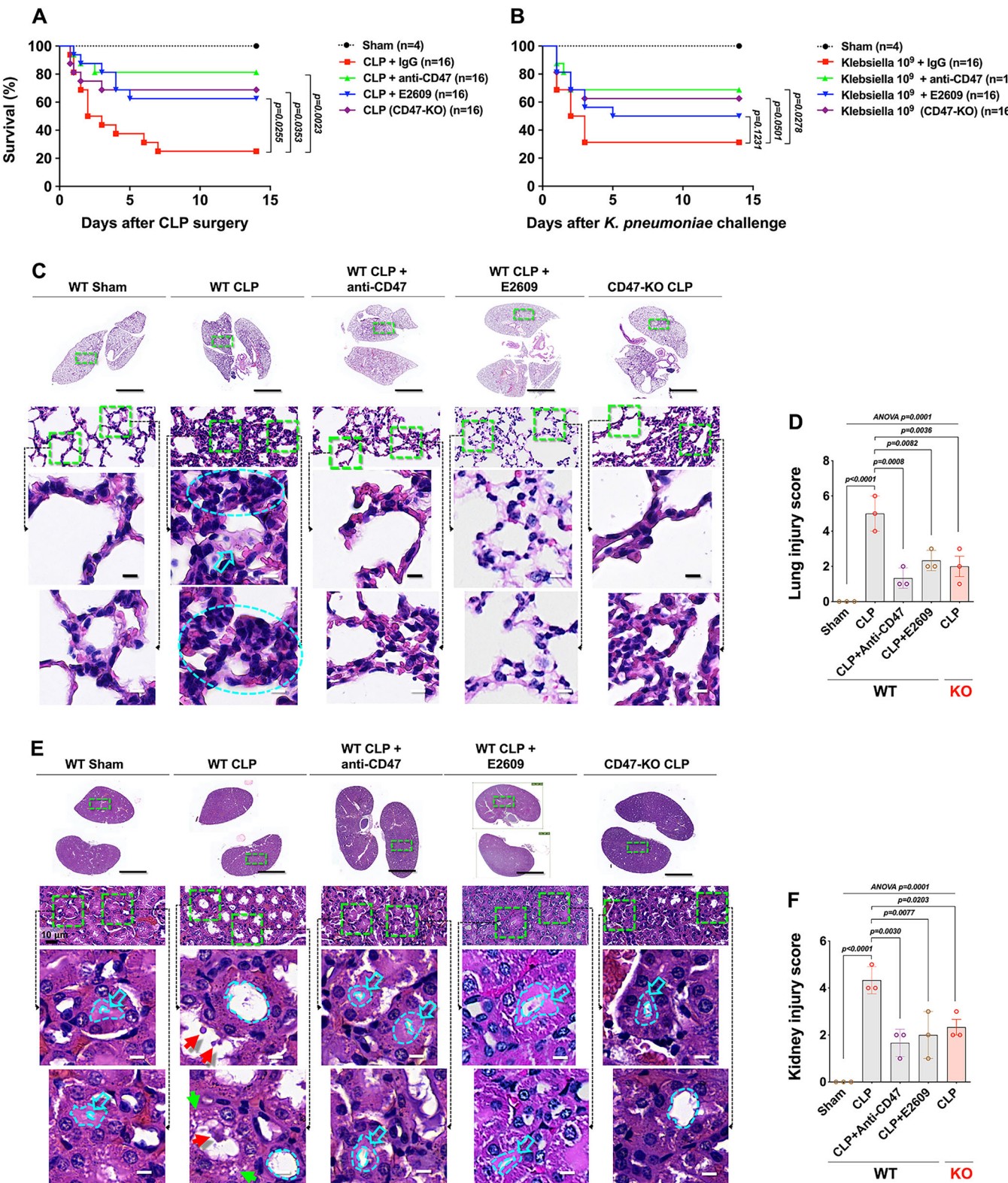

Although the "uncontrolled inflammatory response" theory dominated sepsis research since the 1970s, it was challenged as early as the 1990s (Lederer et al, 1999; Natanson et al, 1994; Oberholzer et al, 2001; Weighardt et al, 2000). Hotchkiss et al, (2013) later emphasized immunosuppression's equal importance to inflammation in sepsis pathogenesis (Hotchkiss et al, 2013). Reduced HLA-DR levels, first reported in 1994 (Lin et al, 1994), have since become a key biomarker for sepsis immune status evaluation. Regarding MDSCs in sepsis, their anti-inflammatory and immunosuppressive roles are widely accepted, but their impact

**Figure 8.  Blocking CD47-amyloid-β signaling improves the survival of septic mice by alleviating B-cell suppression and reducing organ injury.**

(**A, B**) Therapeutic experiment. The mice were divided into 5 groups: sham ($n = 5$); CLP/K.p. + IgG isotype control antibody (2.5 mg/kg, i.v., single injection 1 h after CLP surgery, $n = 20$); CLP/K.p. + anti-CD47 antibody (2.5 mg/kg, i.v., single injection 8 h after CLP/K.p., $n = 20$); CLP/K.p. + E2609 (6.5 mg/kg, i.p., first injection 8 h after CLP/K.p., twice a day for 3 days, $n = 16$); and CD47-KO mice that underwent CLP/K.p. The survival benefit was evaluated via a Kaplan–Meier survival analysis, and significance was assessed by the log-rank test. The P values are shown next to the connecting lines between the CLP-only group and the treatment groups. (**C**) Scans of whole lung sections: the cyan arrow shows alveolar edema and inflammatory cell infiltration in the alveoli. The dashed ellipse indicates the thickening of the alveolar interstitium. (**D**) The lung injury score was calculated according to extravascular features and alveolar features. (**E**) Scans of whole kidney sections: the hollow arrows show thick brush borders on the inner side of the renal tubule in control mice (the dotted curve outlines the area of the brush border of the renal tubules). In CLP mice, the lumens of the renal tubules were markedly dilated, and the brush borders were significantly thinned. The red arrows point to exfoliated cells or debris, and the green arrows point to vacuoles within the tubular walls. (**F**) Kidney injury scores were calculated according to the thickness of the brush border and the degree of interstitial and tubular injury. Scale bars: full-slice scan, 1000 μm; magnified field of view, 5 μm. Data in (**D, F**) are shown as mean ± SD ($n = 3$, biological replicates). Significance was assessed by one-way ANOVA followed by Tukey's multiple comparison test. The exact P values are shown above the bars. Source data are available online for this figure.

on survival remains debated (Lai et al, 2014). Early studies suggested MDSCs improve septic mouse survival (Derive et al, 2012; Sander et al, 2010), while recent findings indicate reducing MDSCs alleviates immunosuppression and enhances survival (Chen et al, 2022; Liang and Shen, 2020; Zhang et al, 2021). MDSCs primarily regulate T cells via IL-10 (Bah et al, 2018) or PD-L1 (Ruan et al, 2020), but their effect on B cells in sepsis was unclear. This study reveals a pathway where MDSCs suppress B cells: pathogen invasion triggers MDSC generation, which secrete Aβ via CD47-SIRPα signaling, leading to B-cell suppression and adaptive immune suppression. This mechanism links innate immune activation and adaptive immune suppression, offering a potential explanation for dysregulated immune responses in sepsis.

Acute organ injury (lung, kidney, liver, etc.) is common in sepsis, but its mechanism remains unclear. According to the classic sepsis pathogenesis theory, such injury is likely caused by a systemic inflammatory response. This theory suggests that even without direct infection, the over-activated immune response triggered by the primary infection site (e.g., CLP-induced abdominal infection) may cause "collateral damage" to organs beyond the initial infection site. However, bacterial detection of the lung, kidney, and liver revealed high bacterial burdens, indicating that CLP-induced systemic infection may spread to these organs and that the organ injuries may be because these organs themselves have become "battlefields" rather than "collateral damage" caused by the immune system. This may be a potential mechanism to explain multi-organ injuries in sepsis.

In recent years, efforts have been made to establish more sepsis models in addition to CLP, including the use of K.p., Staphylococcus aureus, Pseudomonas aeruginosa, influenza virus, etc. Methods include airway intubation and bacterial/viral injections to simulate pneumonia-induced sepsis (Byrnes et al, 2023; Joshi et al, 2024; London et al, 2010; Luna et al, 2009; Secher et al, 2019; Zhao et al, 2024). In clinical practice, bacterial infections like K.p. often cause active pneumonia in long-term hospitalized patients. However, inducing sepsis in normal mice requires a large bacterial dose, while using immunodeficient mice (e.g., nude or SCID mice) does not align with clinical reality, as sepsis patients generally do not have congenital T/B-cell immunodeficiency. Thus, using a high bacterial dose to induce pneumonia sepsis remains a practical alternative until an ideal model is developed. Future studies may consider fixing bacteria in clots and gradually releasing them to avoid the shock caused by a single large amount of bacterial injection.

CD47 is currently becoming a "hot" target for cancer immunotherapy. To date, clinical trials have been initiated with more than 20 antibody drugs that block CD47 as immunotherapies

for cancers (Li, 2019; Logtenberg et al, 2020; Takimoto et al, 2019; Veillette and Tang, 2019). From an evolutionary perspective, CD47 is expressed on almost all cell types (Andrechak et al, 2019; Van et al, 2006), suggesting that it may serve as an "ID card" that marks all self-cells, whereas SIRPα is expressed only on myeloid cells with phagocytic function (Barclay and Van den Berg, 2014), suggesting that it may act as a "Police badge" for phagocytic cells. Once CD47 and SIRPα recognize each other, phagocytes expressing SIRPα spare CD47-expressing cells without phagocytosing them. Under certain physiological conditions, normal cells may also undergo phagocytosis by downregulating the level of CD47 expression; for example, as erythrocytes age, they lose CD47 expression and are phagocytosed by splenic macrophages (Burger et al, 2012; Oldenborg et al, 2000). Therefore, self-recognition, avoiding self-attack, and clearing aging or dying self-cells may be important for the evolution of CD47 and SIRPa.

On the other hand, CD47 expression is upregulated during infection, and the increase is more pronounced during sepsis. If this change is a natural response established during evolution, what is its benefit for the host in combatting infection? According to our data, Aβ might be the answer. Various cleaved forms of Aβ have been found to have antimicrobial functions. Aβ exerts antimicrobial activity against 8 common and clinically relevant microorganisms (Soscia et al, 2010), and the amyloidogenic peptides Aβ1-42, Aβ2-42, and Aβ3p-42 are able to bind to the surfaces of various bacterial strains and Candida albicans and exert potent antimicrobial activities against all these pathogens (Spitzer et al, 2016). The common β-sheet structure shared by Aβ and other host defense peptides can interact and form channels on the membranes of microbes, which may be responsible for its antimicrobial function (Jang et al, 2011; Kagan et al, 2012). Therefore, CD47-SIRPα signaling may promote the release of Aβ from myeloid cells to defend against microbial invasion.

So, is CD47-SIRPα-promoted Aβ secretion beneficial or harmful during infection? The answer may not be easy to determine. Not all immune responses established during evolution benefit the host under all circumstances, such as inflammation, which can cause a variety of pathological changes, especially autoimmune syndromes and cytokine storms in patients with sepsis. We hypothesize that in the case of mild infections, Aβ secretion, along with other innate immune responses, may provide the first line of defense for the host to eliminate microorganisms. On the other hand, in the case of sepsis, excessive Aβ causes severe B-cell suppression, resulting in adaptive immunosuppression, which is detrimental to survival in sepsis. When the CD47-SIRPα-Aβ pathway is blocked by inhibiting CD47, B-cell suppression may be reversed, thereby activating

adaptive immunity, promoting antigen recognition and presentation, antibody production, and ADCC effects, etc., so that innate immunity and adaptive immunity can cooperate more effectively to fight against infection by foreign pathogens. Because the binding of CD47 and SIRPα transmits a "don't eat me" signal, blocking CD47 can directly increase the phagocytic activity of myeloid cells, and this process may not require the direct involvement of B cells.

Traditional concept generally regards the adaptive immune system, in which T/B cells play a major role, and the innate immune system, in which myeloid cells play a major role, as two separate and unrelated immune systems, and that B cells are not involved in bacterial sepsis at all. This concept needs to be updated based on recent advances. Currently, more than 600 papers on B cells and sepsis, and more than 1600 papers on T cells and sepsis can be found on PubMed. A recent study reported that patients with PICS and sepsis had reduced naive and memory B cells and proliferated plasma cells (Sun et al, 2025). Our recently published study found that adaptive immune suppression caused by impaired T cells is closely associated with sepsis-related death (Yang et al, 2025). It has been widely reported that adaptive immune suppression caused by T/B-cell suppression is strongly associated with sepsis mortality. B cells and T cells may participate in the pathogenesis of sepsis through pathogen recognition, antigen presentation, antibody production, direct antibody killing and antibody-assisted immune cell killing, immune cell assistance, and immune memory. In humans, it takes 4–7 days for the adaptive immune system to mount a significant response, and in mice it may work faster, but it generally takes effect after the innate immune response. More studies are needed to explore the potential mechanisms by which B- and T-cell-mediated adaptive immunity participate in sepsis.

Together, our findings may provide translational opportunities for the design of sepsis diagnoses and immunotherapies. CD47 is likely a promising target for sepsis treatment. Certain adaptive immunity-related genes may have the potential to be used to distinguish patients with sepsis from those with common infections. Since the sample size is not large enough, it can be regarded as a preliminary proof-of-concept experiment. In the next stage, these marker combinations will be continuously optimized by deleting or adding new markers. We hope to eventually obtain a set of immune markers to establish an immune scoring system for the diagnosis and immunophenotyping of sepsis.

# Methods

### Reagents and tools table

| Reagent/resource | Reference or source | Identifier or catalog number |
| --- | --- | --- |
| **Experimental models** | | |
| Jurkat cells | | |
| C57BL6/J (M. musculus) | Beijing HFK Bio-Technology, Co., LTD | N/A |
| CD47 knockout mice | Gempharmatech Co., Ltd (Jiangsu, China) | N/A |

| Reagent/resource | Reference or source | Identifier or catalog number |
| --- | --- | --- |
| Klebsiella pneumoniae | Prof. Zhenling Wang, Sichuan University, China. | Cat # BAA-1706 |
| FITC-conjugated Escherichia coli | Prof. Hao Yang, Sichuan University, China | N/A |
| Jurkat cell line (Clone E6-1) | Pricella | Cat # CL-0129 |
| **Antibodies** | | |
| Anti-mouse CD47 (IAP) (in vivo blockade, clone MIAP301) | BioXcell | Cat #BE0270 |
| Elenbecestat (BACE1 inhibitor E2609) | Selleck | Cat #s8600 |
| Anti-CD3 | BD Bioscience | Cat # 557984 and # 746368 |
| Hamster IgG2 λ isotype | BD Bioscience | Cat # 557985 |
| Anti-CD19 | BD Bioscience | Cat # 557398 and # 753680 |
| Rat IgG2a κ isotype control | BD Bioscience | Cat # 553929 |
| Anti-CD4 | BD Bioscience | Cat # 563151 |
| Anti-CD8 | BD Bioscience | Cat # 551162 |
| Anti-CD45 | BD Bioscience | Cat # 557659 |
| Anti-CD47 | BD Bioscience | Cat # 563584 |
| Anti-CD11b | BD Bioscience | Cat #.557397 |
| Anti-Ly6G | BD Bioscience | Cat # 560601 |
| Anti-Ly6C | BD Bioscience | Cat # 563011 |
| Anti-MHC II | BD Bioscience | Cat # 563415 |
| Anti-Mouse CD74 | BD Bioscience | Cat # 555317 |
| APP Rabbit mAb | ABclonal | Cat # No. A17911 |
| Mouse anti-rabbit IgG (conformation-specific) (L27A9) mAb (HRP Conjugate) | CST | Cat # 5127S |
| Rabbit anti-mouse IgG (light chain-specific) (D3V2A) mAb (HRP Conjugate) | CST | Cat # 5800S |
| **Oligonucleotides and other sequence-based reagents** | | |
| PrimeScript™ RT reagent kit with gDNA eraser (Perfect Real Time) | TAKARA | Cat # RR047Q |
| iTaq™ Universal SYBR® Green Supermix | Bio-Rad | Cat # 1725124 |
| HiScript II Q RT SuperMix for qPCR | Vazyme | Cat # R222-01 |
| QX200™ ddPCR™ EvaGreen Supermix | Bio-Rad | Cat # 1864033 |
| **Primers (M. musculus)** | | |
| 18S | This study | F: GCAATTATTCCCCATGAACG R: GGCCTCACTAAACCATCCAA |

| Reagent/resource | Reference or source | Identifier or catalog number |
|---|---|---|
| CD3g | This study | F: ACTGTAGCCCAGACAAATAAAGC R: TGCCCAGATTCCATGTGTTTT |
| CD4 | This study | F: AGGTGATGGGACCTACCTCTC R: GGGGCCACCACTTGAACTAC |
| CD8a | This study | F: CCGTTGACCCGCTTTCTGT R: CGGCGTCCATTTTCTTTGGAA |
| Cd19 | This study | F: CTTGGTATCGAGGTAACCAGTCA R: ACAATCACTAGCAAGATGCCC |
| Pax5 | This study | F: CCATCAGGACAGGACATGGAG R: GGCAAGTTCCACTATCCTTTGG |
| Ciita | This study | F: AGCAGGCCAAGACTTACATGA R: CAGGCTGACATAGAGTCCTGT |
| H2-Eb1 | This study | F: GCGGAGAGTTGAGCCTACG R: AGGCCCGTGGACACAATTC |
| H2-Aa | This study | F: TGGGAGGCTGGACTAAGAGG R: GGGGCTGGGATCTCAGGTT |
| CD47 | This study | F: TGGTGGGAAACTACACTTGCG R: CGTGCGGTTTTTCAGCTCTAT |
| APP | This study | F: TCCGAGAGGTGTGCTCTGAA R: CCACATCCGCCGTAAAAGAATG |
| CD21 | This study | F: TTGCTGCCAAAGTATTCTTTTGC R: TGGAGGTTTCTAAGCAGGTGATA |
| Primers (human) | | |
| PAX5 | This study | F: CTGATCTCCCAGGCAAACAT R: TTGCTCATCAAGGTGTCAGG |
| CD5 | This study | F: CTCACCCGTTCCAACTCGAAG R: TGGCAGACTTTTGACGCTTGA |
| CD19 | This study | F: GGCCCGAGGAACCTCTAGT R: TAAGAAGGGTTTAAGCGGGGA |
| CD8A | This study | F: ATGGCCTTACCAGTGACCG R: AGGTTCCAGGTCCGATCCAG |
| CIITA | This study | F: CTCTACTCAGAACCCGACAC R: ATTGGCATAAGCCTCCCT |
| HLA-DQA1 | This study | F: CTACAACTCTACCGCTGCTA R: GACTGACTGCCCATTGCT |
| HLA-DR | This study | F: TTTCTATCCAGGCAGCATT R: TGTTTCCAGCATCACCAG |
| HLA-DO | This study | F: CATCTGCATCGTGGACAACA R: AAATGGTCAGGCTGGGAAT |
| CD11b | This study | F: CTGCCGCCATCATCTTAC R: CTGGGTGCCCTTGACATT |
| Chemicals, enzymes, and other reagents | | |
| RPMI-1640 | Gibco | Cat # C11875500BT |
| DMEM high glucose | Gibco | Cat # C11995500BT |
| TRYPSIN-0.25% EDTA | Life | Cat # 25200056 |
| Fetal bovine serum (FBS) | Gibco | Cat # 10099141C |
| Penicillin-Streptomycin | HyClone | Cat # SV30010 |
| PBS | BioSharp | Cat # BL302A |
| Red Blood Cell Lysis Buffer | Beyotime | Cat # C3702 |

| Reagent/resource | Reference or source | Identifier or catalog number |
|---|---|---|
| Ficoll lymph separation solution | Solarbio | Cat # P8620 |
| Trypan blue | Sigma-Aldrich | Cat # T8154 |
| Chromium Next GEM Chip G Single-Cell Kit | 10x Genomics | Cat # PC-1000127 |
| Chromium Next GEM Single-Cell 5′ Library and Gel Bead Kit v1.1 | 10x Genomics | Cat # PC-1000167 |
| Chromium Single-Cell 5′ Feature Barcode Library Kit | 10x Genomics | Cat # PC-1000080 |
| Chromium Single-Cell 5′ Library Construction Kit | 10x Genomics | Cat # PC-1000020 |
| Chromium i7 Multiplex Kit | 10x Genomics | Cat # PN-120262 |
| Library Amplification Kit | 10x Genomics | Cat # PC-1000249 |
| Target Hybridization Kit | 10x Genomics | Cat # PC-1000248 |
| Chromium Controller and Next GEM Accessory Kit | 10x Genomics | Cat # PC-1000204 |
| Chromium X and Accessory Kit | 10x Genomics | Cat # PC-1000331 |
| Chromium X Upgrade Package | 10x Genomics | Cat # PC-1000330 |
| Bovine Serum Albumin (BSA) | BioFroxx | Cat # 9048-46-8 |
| DAPI | Solarbio | Cat # C0060 |
| TRIzol | Invitrogen | Cat # 15596026 |
| DEPC | Beyotime | Cat # R0022 |
| Isopropyl alcohol ((CH3)2CHOH) | KE SHI | Cat # 67-63-0 |
| Trichloromethane (CHCl3) | KE SHI | Cat # 67-66-3 |
| Ethanol (CH3CH2OH) | KE SHI | Cat # 64-17-5 |
| Methanol (CH3OH) | KE SHI | Cat # 67-56-1 |
| Lipopolysaccharides from E. coli (O55:B5) | Sigma-Aldrich | Cat # L5418 |
| Protease Inhibitor Cocktail (100×) | Bimake | Cat # B14001 |
| IP lysis buffer | Bimake | Cat # P0013 |
| Protein A/G magnetic beads | MCE | Cat # HY-K0202 |
| ECL | ThermFisher | Cat # A38555 |
| Software | | |
| R version 4.0.1 system: x86_64, mingw32 | The R Foundation | https://www.r-project.org |
| R Studio | Open source | Open source |

| Reagent/resource | Reference or source | Identifier or catalog number |
|---|---|---|
| CellRanger | 10x Genomics | http://10xgenomics.com/ |
| CellChat | Jin et al (2021) | http://www.cellchat.org/ |
| GraphPad Prism | N/A | N/A |
| ImageJ | N/A | N/A |
| Photoshop | N/A | N/A |
| Adobe Illustrator 2024 | N/A | N/A |
| **Other** | | |
| Animal Genomic DNA Quick Extraction Kit | Beyotime | Cat # D0065MS |
| Trucount Tubes | BD Bioscience | Cat # No.340334 |
| CD11b microbeads, mouse | Miltenyi Biotec | Cat # 130-097-142 |
| CD19 microbeads, mouse | Miltenyi Biotec | Cat # 130-121-301 |
| CD3 positive selection kit | Selleck | Cat # B90020 |
| BD Pharmingen™ FITC Annexin V Apoptosis Detection Kit I | BD Bioscience | Cat # 557984 |
| Blood agar plate | HKM | Cat # 024070 |
| autoMACSTM Pro washing solution | Miltenyi Biotec | Cat # 130-092-987 |
| Aβ1-42 ELISA Kit-96T | Kmaels (Shanghai) Biological Technology Co. | Cat # DRE-M0900c |
| Mouse TNF-alpha Quantikine ELISA Kit | R&D | Cat # MTA00B |
| Mouse CCL2/JE/MCP-1 Quantikine ELISA Kit | R&D | Cat # MJE00B |
| Mouse IL-6 Quantikine ELISA Kit | R&D | Cat # M6000B |
| Mouse IL-1 beta/IL-1F2 Quantikine ELISA Kit | R&D | Cat #MLB00C |

## Ethics

The observational study using human blood samples for measurements of the expression of adaptive immunity-related genes was approved by Sichuan University on March 5, 2021 (Approval No. MR 2021_S053). Informed consent was obtained from all subjects. The animal experiments were approved by the Animal Ethics Committee of Sichuan University and performed according to institutional and national guidelines (Approval No. 2021-S-012).

## Human data

A total of 20 healthy volunteers and 51 patients with newly diagnosed infections were included in the study. The patients were initially diagnosed with an infection after being admitted to the emergency

department. The detailed information of all participants was extracted later according to their hospital number. The characteristics of the patients are listed in Dataset EV1, including a brief case history, SOFA score, blood pressure, WBC counts, and other information obtained when the patients were admitted to the Emergency Department, as well as their progress and outcomes. The diagnosis of sepsis was made after a SOFA score was calculated for the patient after sufficient clinical and laboratory results had been collected, including the $PaO_2/FiO_2$ (mmHg), platelet count ($\times10^9$/L), bilirubin level (μmol/L), mean arterial pressure, GCS, and creatinine level (μmol/L). Blood samples (3 ml) were collected in sodium heparin-coated vacutainer tubes (BD Biosciences) within 24 h of the subject's arrival at the emergency department. The blood samples were temporarily stored at 4 °C for 1–1.5 h, and PBMCs were isolated using a density gradient centrifugation protocol (standard Ficoll gradient). The total RNA was extracted from the PBMCs using TRIzol reagent and subjected to two-step droplet digital PCR (ddPCR).

## Cecal ligation and puncture (CLP)

In brief, Balb/c or C57/BL6 mice (8–10 weeks, 20–25 g) were obtained from the Beijing HFK Bio-Technology.co., LTD. The mice were housed under specific pathogen-free (SPF) conditions for one week to adapt prior to the experiment and anesthetized with an intraperitoneal (i.p.) injection of pentobarbital sodium (50 mg/kg). An abdominal midline incision was made, and the cecum was isolated and ligated 2.5 mm (mild infection model) or 7.5 mm (critical infection model) from the cecal tip. The ligated cecal stump was then perforated once using 26-gauge (mild) or 18-gauge (critical) needles. For the sham surgery, the cecum was isolated, but no ligation or perforation was performed. After the operation, the cecum was returned to its normal intra-abdominal position, and the abdomen was closed with running 5–0 Prolene sutures. Sham-operated mice underwent identical procedures, except for the CLP.

All the mice that underwent CLP surgery received antibiotics and underwent fluid resuscitation. Briefly, the mice received a single intramuscular injection of ciprofloxacin at a dose of 20 mg/kg and were subjected to subcutaneous fluid resuscitation with 800 μl of saline immediately after surgery, as described in our previous study (Xiao et al, 2018).

## *Klebsiella pneumonia*-induced sepsis model

We established a pneumonia-induced sepsis model by injecting *Klebsiella pneumoniae* directly into the lungs of mice via tracheal intubation. We tested four different doses of $10^7$ CFUs, $10^8$ CFUs, $10^9$ CFUs and $10^{10}$ CFUs. Klebsiella pneumoniae strain was provided by Professor Zhenling Wang (Sichuan University, China).

## LPS-induced endotoxaemia

C57/BL6 mice (8–10 weeks, 20–25 g) were i.p. injected LPS(055: B5, Sigma) with four different doses of 10 mg, 15 mg, 20 mg and 25 mg to establish the endotoxin model.

## Droplet digital PCR (ddPCR)

A QX200 ddPCR system (Bio-Rad) was used to detect the absolute copy numbers of the genes of interest. This system uses

microfluidic technology to achieve partitioning on a massive scale, generating 20,000 highly uniform nanoliter-sized droplets per sample. RNA extraction and two-step RT–ddPCR were performed according to the manufacturer's instructions. Briefly, total RNA was extracted using TRIzol reagent and stored at 300–500 ng/µl in DEPC-treated water at −80 °C. cDNA was generated according to standard protocols using HiScript® II Q RT SuperMix for qPCR (+ gDNA wiper) (Vazyme). Once the reverse transcription reaction was complete, the concentration of cDNA was reduced to approximately 0.2 ng/µl RNA equivalent, and a 1 µl per droplet digital PCR (ddPCR) reaction (Bio-Rad QX200™ ddPCR™ Eva-Green® supermix, total volume of 20 µl) was performed. After droplet generation with the QX200 Droplet Generator, the droplets were carefully transferred into a 96-well plate. The plate was sealed with a PX1 PCR plate sealer. Thermal cycling and subsequent reading of the droplets were performed in the QX200 Droplet Reader. After data acquisition, samples in the well selector under analysis were selected. The concentration reported is the copies/µl of the final 1× ddPCR.

## Preparation of single-cell suspensions

Eight hours after the sham or CLP surgery, the mice were anesthetized, and whole blood (~600 µl per mouse) was collected from the retro-orbital sinus and placed in K3-EDTA tubes. The blood samples were temporarily stored at 4 °C for 1–1.5 h (for better separation). For PBMC isolation, an equal volume (600 µl) of whole-blood dilution buffer was added and mixed gently to avoid red blood cell destruction. Lymphocyte separation solution (3 ml) was subsequently added. If the total volume was less than 3 ml, 3 ml of lymphocyte separation solution was added; if the total volume was >3 ml, an equal volume of lymphocyte separation solution was added. The total volume did not exceed two-thirds of the centrifuge tube volume (15 ml). The mixture was subsequently centrifuged at $400 \times g$ for 30 min at 4 °C. After centrifugation, the upper layer of plasma, the lower layer of Ficoll–Paque media, the granulocyte layer, and the RBC layer were discarded (Appendix Fig. S6). The buffy coat, which contained all the mononuclear cells (PBMCs, including lymphocytes and myeloid cells), was aspirated. A thick Ficoll–Paque media layer was observed between the granulocytes and the buffy coat, and thus the neutrophils could be removed cleanly. The viability of the PBMCs was evaluated by Trypan blue staining (cell viability >98%). The single-cell suspension was then loaded in a 10x Chromium system to form a water-in-oil mixture with the Single-Cell 5′Library & Gel Bead Kit (10x Genomics).

## Single-cell RNA sequencing (scRNA-seq)

In brief, a single-cell suspension was loaded in a 10x Chromium system to form a water–oil mixture using the Single-Cell 5′Library & Gel Bead Kit (10x Genomics). The RNA from barcoded cells was reverse transcribed, and a sequencing library was constructed using the Chromium Single Cell 10x v2 Kit according to the manufacturer's instructions. Sequencing was performed on NovaSeq (Illumina) platforms. Quality control of the raw sequence reads was performed with Trimmomatic software. The raw sequencing data were compared to the mouse genome mm10 and demultiplexed with the CellRanger software (version 5.0, 10x Genomics) pipeline (https://support.10xgenomics.com/single-cell-gene-expression/ software/pipelines/latest/what-is-cell-ranger) to generate gene matrices of gene counts based on the cell barcodes.

## Preprocessing of scRNA-seq data

CellRanger (version 5.0, 10x Genomics) was used to filter each gene and each cell barcode, and a digital expression matrix was generated using the unique molecular identifiers. All downstream single-cell analyses were performed using Seurat (version 4.0, 10x Genomics). A second filtration by Seurat with R software was performed as follows: if (1) the number of cells expressing a gene was less than 3, (2) the number of genes expressed in a cell was less than 200; (3) the proportion of mitochondrial genes was not in the range of 5–30%; or (4) if the cells were identified as doublets or foreign cells, the raw sequencing reads were filtered out. After data normalization, dimensionality reduction, cell clustering, and differential expression analysis with the Seurat package, marker genes were screened with the EDGG R package. The cell clusters were defined according to the identified marker genes. The reference databases for the definition of marker genes were as follows: CellMarker (http://biocc.hrbmu.edu.cn/CellMarker/); PanglaoDB(https://panglaodb.se/), Mouse Cell Atlas (http://bis.zju.edu.cn/MCA/gallery.html), and the European Centre for Bioinformatics (https://www.ebi.ac.uk/gxa/sc/home). All subsequent analyses, including t-SNE, UMAP, and cell–cell interaction analyses, performed in this study were based on the above preprocessed data.

## Flow cytometry analysis

Anti-CD3 (Cat. 557984 and Cat. 746368) and hamster IgG2 λ isotype control (Cat. 3557985), anti-CD19 (Cat. 557398 and Cat. 753680) and rat IgG2a κ isotype control (Cat. 553929), anti-CD4 (Cat. 563151), anti-CD8 (Cat. 551162), anti-CD45 (Cat.557659), anti-CD47 (Cat. 563584), anti-CD11b (Cat. 557397), anti-Ly6G (Cat. 560601), anti-Ly6C (Cat. 563011), and anti-MHC II (Cat. 563415) antibodies were purchased from BD Bioscience. The single-cell suspensions were added to K3-EDTA anticoagulant tubes and mixed well. A blank tube containing 50 µl of the antibody-free cell suspension was used to adjust the voltage of the flow cytometer. For staining with a single antibody, the cells in the tubes were incubated with one of the antibodies used for the detection of biomarkers and were used for compensation and adjustment to exclude nonspecific expression. For staining with multiple antibodies, the cells in the tubes were incubated with an even mixture of all the antibodies. Approximately 50,000–100,000 cells/events were detected in each tube, and cells stained with the isotype control antibodies or a single antibody were used to adjust the compensation and thus remove the background. All antibodies were used for flow cytometry as follows: the antibody was incubated in the dark for 15 min, 450 µl of 1×FACs hemolysin was added to the tube, and the mixture was incubated for 15 min to fully lyse the red blood cells. The gating strategy was based on the schemes provided by BD Bioscience. Briefly, FSC-H and FSC-A were used to remove doublets, and SSC-A and FSC-A were used to remove small microparticles and debris. DAPI was used to differentiate dead and live cells (DAPI-negative cells were considered live cells), and CD45 was used as a lineage marker for all leukocytes.

The absolute number of each cell population was calculated using a microbead counter (Trucount Tubes, Cat. 340334, BD bioscience). The calculation formula was based on the manufacturer's instructions:

Absolute cell number = positive cell events ÷ bead events × (46800 bead events/50 μl). The simplified formula is as follows: (number of cells per tube ÷ number of beads per tube) × 936 (cells/μl). The cell numbers were labeled next to the corresponding cell population within the FCM graphs (cells/μl). The FCM assay revealed that after CLP surgery, the number of neutrophils increased most significantly, followed by the numbers of PMN-MDSCs and M-MDSCs. The trends of the changes in the cell populations detected by FCM and scRNA-seq were consistent.

## Cell–cell communication analysis

CellChat was used to analyze cell–cell communication. CellChat is a database that can analyze cell–cell signaling networks using scRNA-seq data (Jin et al, 2021) and includes CellChat Human and CellChat Mouse (species) and three subdatabases: (Peng et al, 2020) Secreted Signaling, (Boomer et al, 2011) Extracellular Matrix (ECM)–Receptor interactions, and (3) Cell–Cell contacts. The APP-CD74 interaction was screened using the Secreted Signaling subdatabase ("CellChatDB.use <- subsetDB(CellChatDB, search = "Secreted Signaling") and displayed in a netVisual_bubble diagram.

## RNA sequencing and data analysis

Total RNA was extracted from PBMCs and cells from the lymph nodes, spleen, and bone marrow. RNA purity was checked using a Nano Photometer spectrophotometer (IMPLEN, CA, USA), and RNA integrity was assessed using a RNA Nano 6000 Kit and a Bioanalyzer 2100 system (Agilent Technologies, CA, USA). A total of 1 mg of RNA/sample was used as the input material for the RNA sample preparation. The sequencing libraries were generated using the NEBNext UltraTM RNA Library Prep Kit for Illumina (NEB, USA) according to the manufacturer's instructions. The library quality was determined with an Agilent Bioanalyzer 2100 system. Sequencing was performed on the Illumina NovaSeq platform. FeatureCounts v1.5.0-p3 was used to count the number of reads mapped to each gene. The differential expression analysis was performed using the DESeq2 R package (1.16.1).

## Generation of CD47-KO mice

CD47-KO ($Cd47^{-/-}$) mice were generated from the C57BL/6J strain using the CRISPR/Cas9 system. Briefly, 2 gRNAs that target specific locations in exon 2 and exon 3 of the murine $Cd47$ gene were designed. The $Cd47$/Cas9 plasmids were injected into the zygotes of C57BL/6J mice. The sequencing results revealed that a total of 1228 bases were deleted, accompanied by frameshift mutations. No fetal death or neonatal death occurred in the CD47-KO mice. No discernible abnormalities were observed in terms of hair, altered behavior, food or water intake, or fertility. The detailed design and identification of the CD47-KO mice are shown in Appendix Fig. S16. Genomic DNA was extracted from mouse tails forgenotyping (#D7283S, Beyotime). Primers for CD47 genotyping of mice:CD47-KO-TF3A:AGTTACAGTCTACTGGCTGGTGTGC,CD47-KO-TR 3A:AACAACACTGCTGTCCGCAACTC;CD47-KO-TF1: TACAG

TCTACTGGCTGGTGTGC, CD47-wt-TR1:TTCAAAGAGGCAATGC CATTG.

## Blood bacterial load

Blood samples were collected from WT mice that received sham, mild CLP, or critical CLP surgery or from CD47-KO mice that received critical CLP surgery, and the sample collection time was 24 h or 48 h after surgery. Ten microliters from each sample were plated on blood agar plates, and bacterial colonies (CFU) were counted manually.

## Measurement of bacterial loads in various organs (liver, kidney, lung)

Samples of the liver, kidney, and lungs were collected from mice 8 h after underwent sham, mild CLP, or critical CLP surgery. Blood agar plates were used to detect the bacterial load in the tissues.

## Therapeutic experiments

Eight hours after the CLP surgery, the mice were administered a mouse-derived anti-CD47 antibody (2.5 mg/kg in 100 μl of saline through the tail vein, one-time injection) or isotype control IgG. The administration of the anti-CD47 or isotype control IgG antibody was performed by investigators who were blinded to the experimental design, including the groups. Data collection and analysis were not blinded. The mice treated with vehicle (saline) or the potential therapeutic agents were observed for 15 days. All the mice (8 male mice and 8 female mice per group) that underwent CLP surgery received antibiotics and fluid therapy. After the CLP surgery, the mice received a single intramuscular injection of ciprofloxacin at a dose of 20 mg/kg and were immediately subjected to subcutaneous fluid resuscitation with 800 μl of saline. The survival outcomes were analyzed by constructing Kaplan–Meier survival curves, and the significance of the differences was assessed using the log-rank test.

## Real-time RT–PCR

Total RNA was extracted from the tissues of interest using TRIzol reagent, which was obtained from Invitrogen. Reverse transcription was performed with a Superscript II two-step RT–PCR kit (Invitrogen). PCR was performed using SYBR® Green PCR Master Mix (Applied Biosystems). Gene expression was normalized to that of GAPDH. The primers used to determine the mRNA levels of the genes of interest are listed in the Reagents and Tools Table.

## Phagocytosis assay

Peritoneal macrophages from healthy control mice (sham surgery), septic mice (CLP surgery), or septic mice treated with the anti-CD47 antibody (1 h after CLP surgery, 2.5 mg/kg, one-time injection through the tail vein) were isolated from the peritoneal cavity of the mice 8 h after CLP surgery. The cells were seeded on 24-well plates, and $Escherichia\ coli$ expressing green fluorescent protein (transfected with a GFP expression plasmid) was added. The phagocytosis of $E.\ coli$ by macrophages was observed and photographed under an inverted fluorescence microscope.

The phagocytic activity was determined by measuring the number of macrophages that engulfed the fluorescent bacteria (using Image-Pro Plus software). The phagocytosis rate (%) = (phagocytosing cells/all cells in a $60 \times 60$ mm dish) $\times 100\%$. Each plot in the statistical diagram represents the phagocytosis rate (%) from one $60 \times 60$ mm dish, $n = 3$ (dishes). The phagocytic process was captured with a Nikon Ti2-U inverted fluorescence microscope and analyzed with the software NIS-Elements BR 5.21.00 64-bit.

## Transwell assay

A total of $2 \times 10^6$ CD11b$^+$ myeloid cells sorted (using magnetic beads) from the spleens of healthy mice (sham surgery) or septic mice (CLP surgery) were seeded in the upper chambers (Transwell inserts, pore size: 0.4 μm, Millipore) of six-well plates. The same number of B cells ($2 \times 10^6$) from the spleens of healthy mice were sorted using CD19$^+$ magnetic beads and seeded in the lower chambers. Following an incubation at 37 °C with 5% $CO_2$ for 8 h, the B cells in the lower chamber were collected, and the mRNA levels of CD19, *Pax5*, H2-Aa, and *Ciita* in the B cells were measured using real-time RT–PCR. B cells that were directly isolated from septic mice (8 h after CLP surgery) were used as positive controls.

## Immunoprecipitation–western blot (IP-WB) analysis

The cells were washed with ice-cold PBS and lysed with nondenaturing lysis buffer containing a cocktail of proteinase inhibitors. After centrifugation, the supernatant was extracted and incubated with the anti-*App* antibody (2 μg/ml) with continuous rotation at 4 °C overnight. The mixture was then incubated with agarose beads (Cat#HY-K0202, MCE) under continuous rotation at 4 °C for 4 h. After centrifugation, the supernatant was removed. The beads were washed with nondenaturing lysis buffer, mixed with loading buffer, and then boiled for 5 min. After centrifugation, the supernatant was collected and subjected to Western blot analysis (with an anti-CD74 antibody).

## ELISA

The culture medium of myeloid cells (in vitro) and the blood of healthy or CLP mice treated with the anti-CD47 antibody or E2609 (in vivo) were collected to detect the levels of Aβ1-42, IL-6, TNF-α, IL-1β, and MCP using an ELISA kit according to the manufacturer's instruction.

## HE staining

All animal experiments were approved by the Animal Ethics Committee of Sichuan University and performed according to institutional and national guidelines. The mice were anaesthetized and per fused transcardially with 4% paraformaldehyde in PBS for 10 min. The lung and liver were removed, post-fixed with 4% paraformaldehyde for 24 h, embedded in paraffin, sectioned at a 5-μm thickness, and and processed for HE staining. Briefly, tissue sections on glass slides were rehydrated with xylene and alcohol and counterstained with hematoxylin and eosin to label nuclear and cytoplasm, respectively, and finally visualized under the slide scanning system (VS200, Olympus, Tokyo, Japan).

## Statistics

When two groups were compared, each dataset was tested for a normal (Gaussian) distribution first, and Welch's *t* test (for data that passed the normality test) or the Mann–Whitney *U* test (for data that did not pass the normality test) was used. For multigroup comparisons, one-way ANOVA followed by Tukey's multiple comparison test was used, and the normality of the residuals was assessed using "Anderson–Darling (A2*)", "D'Agostino-Pearson omnibus (K2)", "Shapiro–Wilk (W)", and "Kolmogorov–Smirnov (distance)" tests. A *P* value <0.05 was considered statistically significant. Survival outcomes were analyzed by constructing Kaplan–Meier survival curves, and the significance of the differences was assessed using the log-rank test. Receiver operating characteristic (ROC) curves were analyzed to evaluate the diagnostic performance of a candidate gene to differentiate infected patients from healthy volunteers or to differentiate patients with a common infection from those who developed sepsis. The area under the curve (AUC), *p* values and cutoff values are marked on the ROC curves. The cutoff value of a candidate gene was defined as the value with the maximum specificity and sensitivity to discriminate between healthy controls and infected patients (or between patients with a common infection and those who developed sepsis).

## Data availability

The scRNA-seq data have been deposited in the NCBI/SRA repository under accession number GSE226648. The RNA-seq data have been deposited in the NCBI/SRA repository under accession numbers GSE226649 and GSE222996. The datasets produced in this study are available in S-BIAD1639: https://www.ebi.ac.uk/biostudies/bioimages/studies/S-BIAD1639.

The source data of this paper are collected in the following database record: biostudies:S-SCDT-10_1038-S44319-025-00442-4.

## Peer review information

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

## Acknowledgements

The authors thank Hao Yang (Sichuan University, China.) and Zhenling Wang (Sichuan University, China.); Sisi Wu, Jinhan Zhou, Jiayi Xu, Jiehao Chen, Xiaojiao Wang, Xin Luo, Yi Zhang, and Yufei Cheng (Core Facilities of West China Hospital, Sichuan University) for their technical supports and help. This work was supported by the National Natural Science Foundation of China (82172142 to Fei Xiao), the Natural Science Foundation of Sichuan Province (2024NSFSC1536 to Zhongxue Feng), the 1.3.5 Project for Disciplines of Excellence, West China Hospital, Sichuan University (ZYGD18020/ ZYJC18006 to Yan Kang), and the Sichuan Province Science and Technology Program (2023YFH0074 to Chunling Jiang).

## Author contributions

**Zhongxue Feng**: Conceptualization; Resources; Data curation; Formal analysis; Investigation; Methodology; Writing—original draft. **Lijun Wang**: Resources; Data curation; Formal analysis. **Yang Li**: Data curation; Investigation. **Yonggang Wei**: Validation; Methodology. **Yueyue Zhou**: Data curation; Investigation; Methodology. **Siying Wang**: Investigation; Methodology. **Xiaoqi Zhang**: Investigation; Methodology. **Chunling Jiang**: Methodology. **Xuelian Liao**: Resources. **Yan Kang**: Resources; Supervision; Funding acquisition; Project administration. **Fei Xiao**: Conceptualization; Resources; Supervision; Funding acquisition; Project administration. **Wei Zhang**: Conceptualization; Resources; Data curation; Formal analysis; Supervision; Funding acquisition; Methodology; Writing—original draft; Project administration; Writing—review and editing.

Source data underlying figure panels in this paper may have individual authorship assigned. Where available, figure panel/source data authorship is listed in the following database record: biostudies:S-SCDT-10_1038-S44319-025-00442-4.

## Disclosure and competing interests statement

The authors declare no competing interests.

# Expanded View Figures

**Figure EV1.  Whole-mount scans of H&E-stained sections.**

(**A**) Lung, (**B**) liver, and (**C**) kidney tissue sections from mice that underwent sham surgery, mild CLP surgery, and critical CLP surgery. Scale bars: full-slice scan, 1000 µm; first-level magnification field of view, 100 µm; second-level magnification field of view, 20 µm. ▶

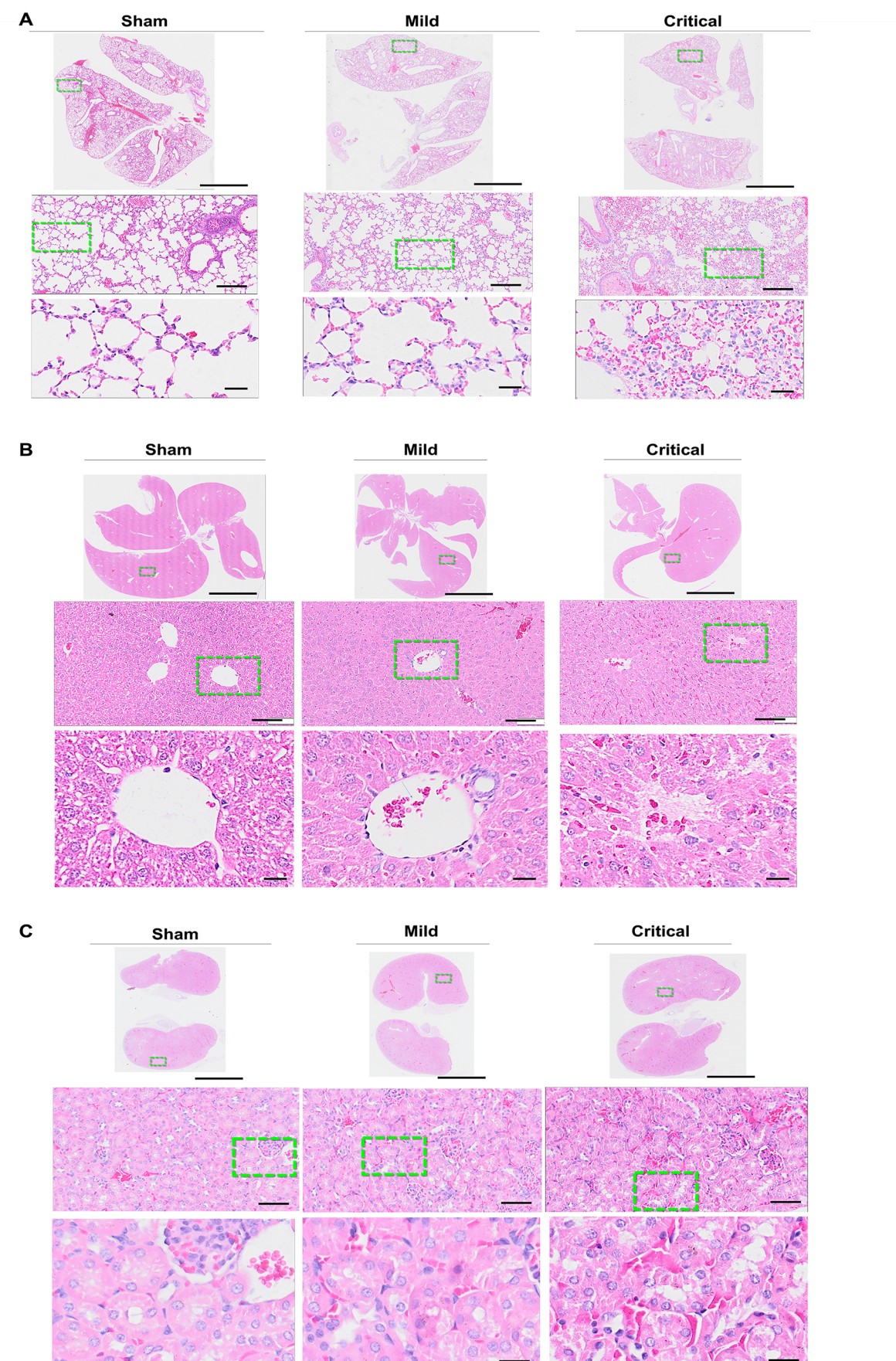

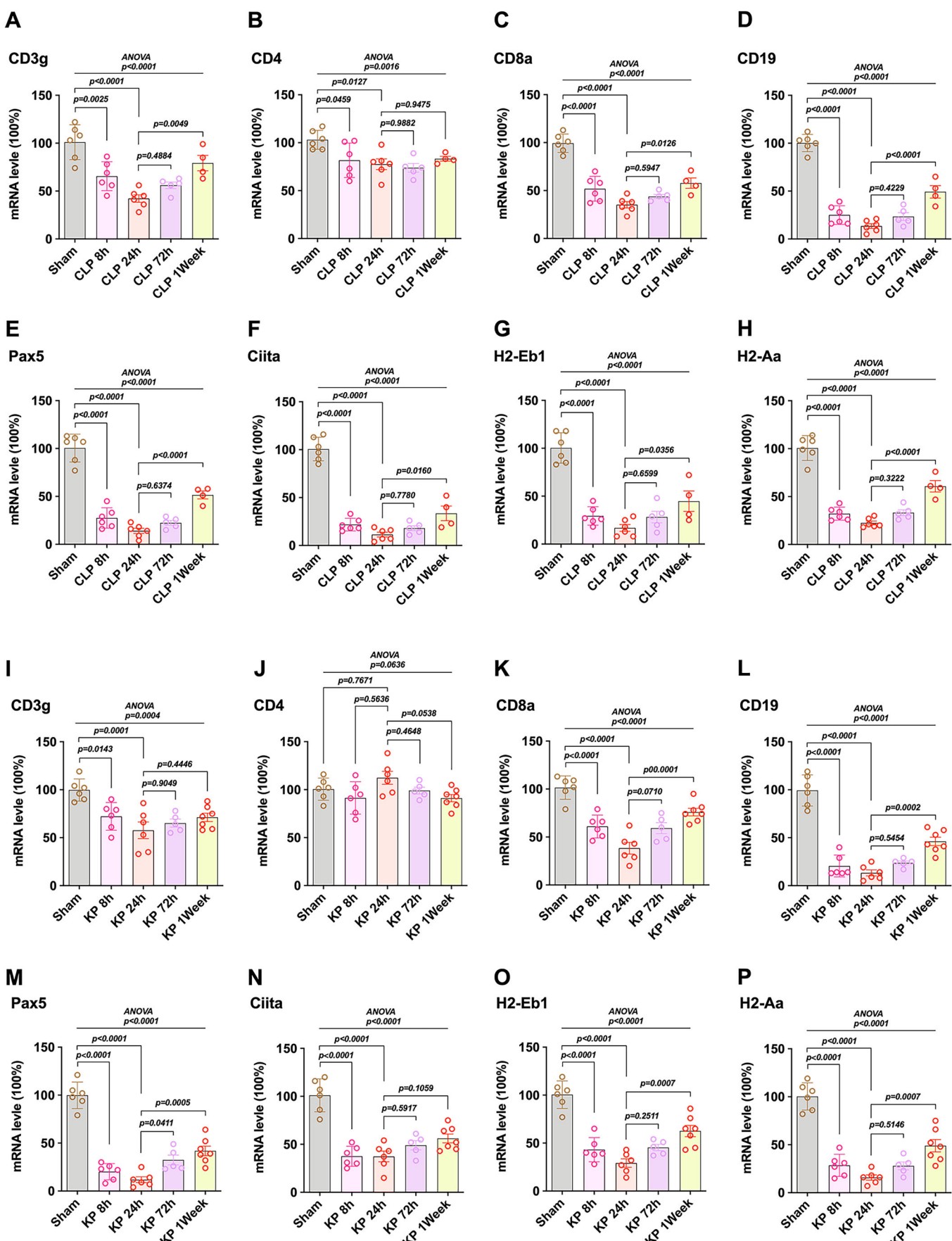

◀  **Figure EV2.  Expression of adaptive immune-related genes over time during sepsis.**

Real-time RT-PCR was used to detect the mRNA levels of adaptive immune-related genes at 8 h to 1 week after CLP surgery (**A–H**) and *K.p.* infusion (**I-P**). Data are shown as mean ± SD. The number of mice per group (n-number, biological replicates) was indicated by the points in the scatter plot. Significance was assessed by one-way ANOVA followed by Tukey's multiple comparison test. The exact *P* values are shown above the bars.

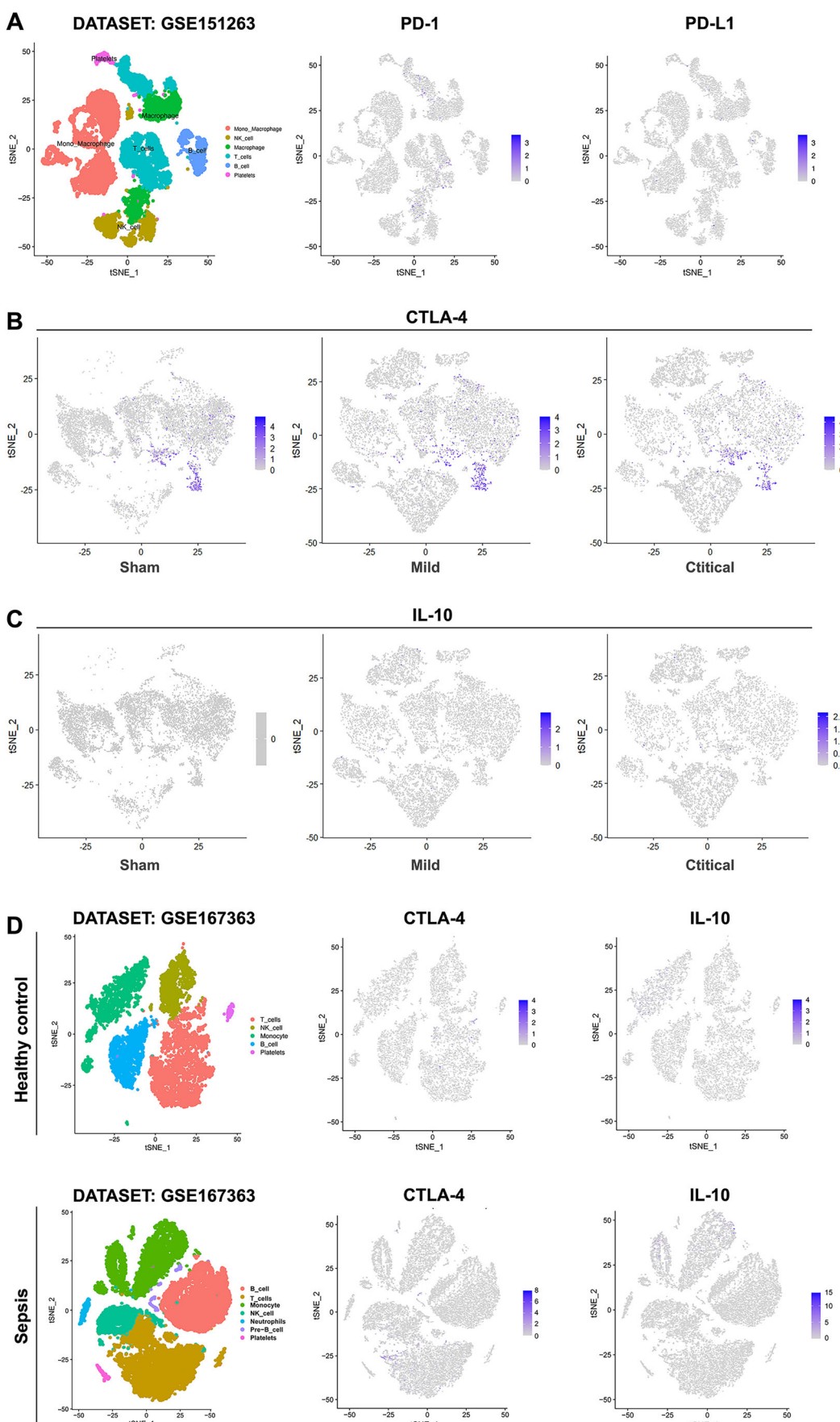

**Figure EV3. The expression level of canonical immunosuppression genes.**

(A) The scRNA-Seq public dataset GSE151263 shows the expression pattern of PD-1 and PD-L1 in human PBMCs. (B, C) The scRNA-Seq data shows the expression of CTLA-4 and IL-10 in mice received sham, mild or critical CLP surgery. (D) The scRNA-Seq public dataset GSE167363 shows the expression pattern of CTLA-4 and IL-10 in human PBMCs.

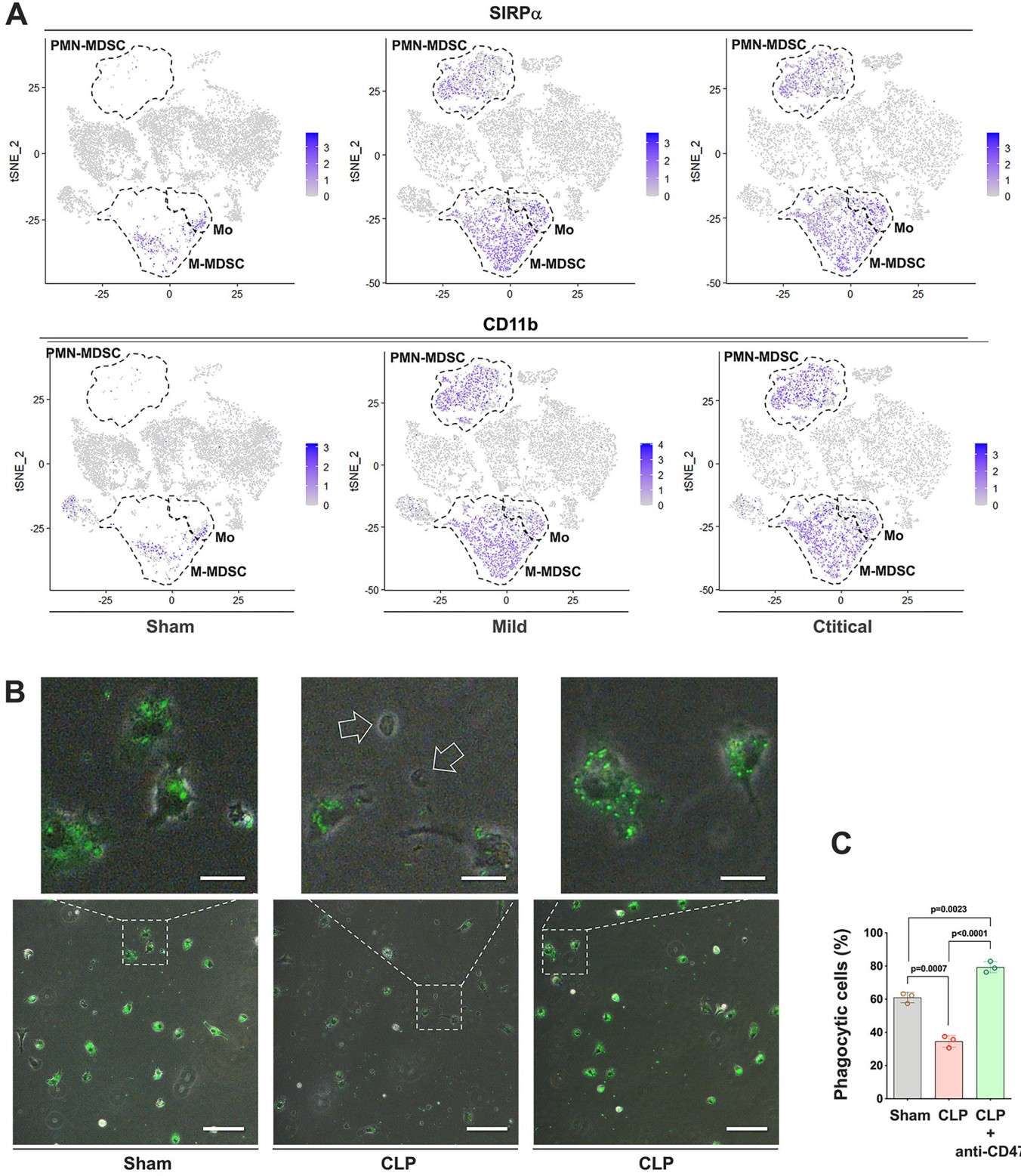

◀ **Figure EV4. CD47-SIRPα signaling reduces the phagocytotic ability of myeloid cells.**

(**A**) Representative images showing the process of *E. coli* phagocytosis by myeloid cells from healthy control mice, CLP mice, or CLP mice treated with anti-CD47. (**B**) Phagocytosis rate (%) = (phagocytosing cells/all cells in a 60 × 60 mm dish) × 100%; each plot represents the Phagocytosis rate (%) from one 60 × 60 mm dish. Scale bars: upper field of view, 20 µm; lower field of view, 100 µm. (**C**) Data are shown as mean ± SD ($n = 3$, biological replicates) and were assessed by the 2-tailed parametric Welch's *t* test. The exact *P* values are shown above the bars.

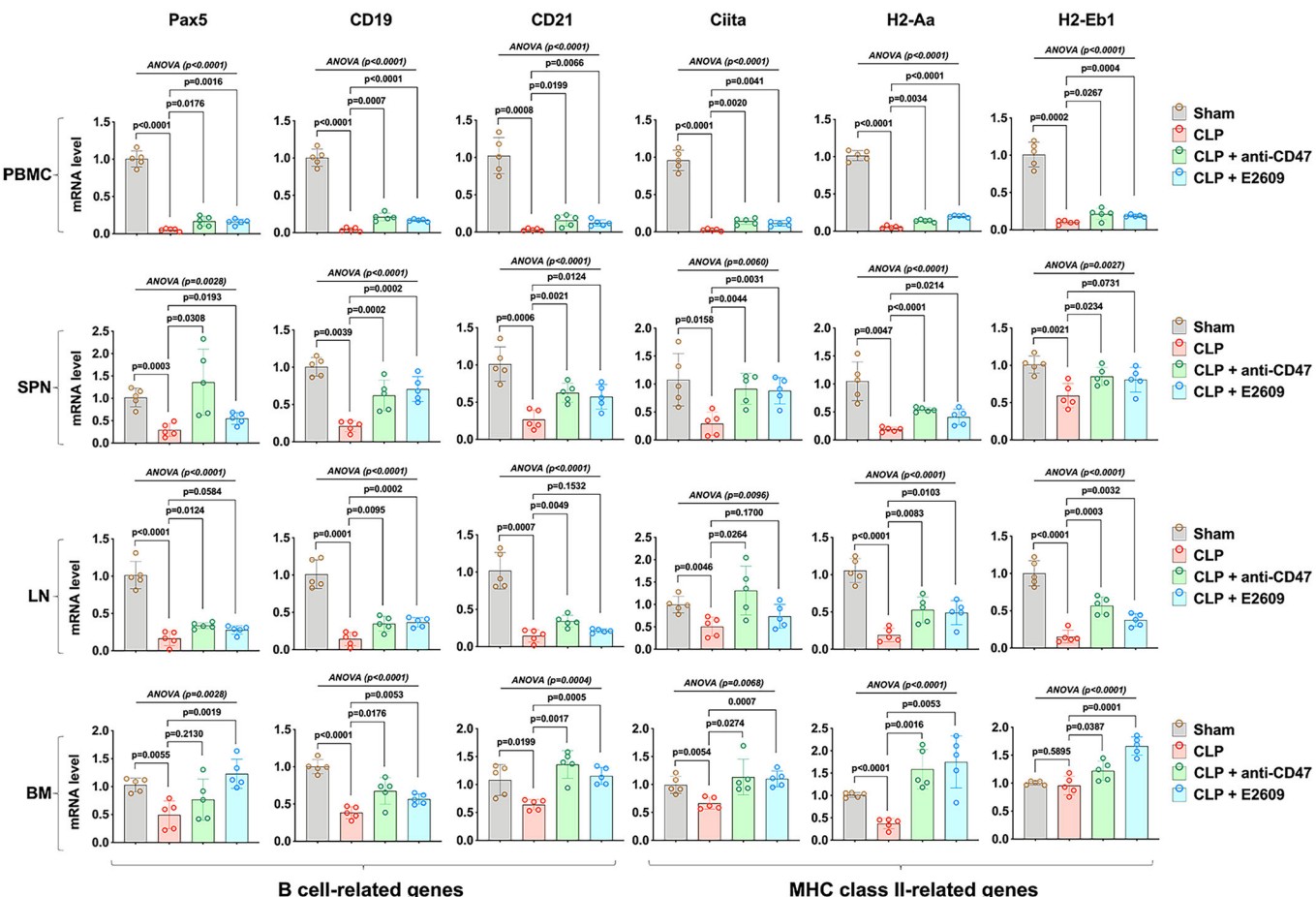

**Figure EV5. Blocking CD47-amyloid-β signaling restores the transcription of genes that had been strongly suppressed in immune compartments.**

The mRNA levels of Pax5, CD19, CD21, Ciita, H2-Aa, H2-Eb1 in PBMC, lymph nodes (LN), spleen (SPN), and bone marrow (BM) from control mice, CLP mice or CLP mice treated with anti-CD47 or E2609. The anti-CD47 antibody (2.5 mg/kg, i.v.) or Aβ inhibitor E2609 (6.5 mg/kg, i.p.) was injected 1 h after CLP surgery, and samples were collected 8 h after CLP. Data are shown as mean ± SD ($n = 5$, biological replicates) and were assessed by one-way ANOVA test followed by Tukey's multiple comparison. The exact $P$ values are shown above the bars.

