## [Peer Review File · EMBO Reports]

CD47-amyloid- β -CD74 signaling triggers adaptive immunosuppression in sepsis

Zhongxue Feng, Lijun Wang, Yang Li, Yonggang Wei, Yueyue Zhou, Siying Wang, Xiaoqi Zhang, Chunling Jiang, Xuelian Liao, Yan Kang, Fei Xiao, and Wei Zhang

Corresponding author(s): Wei Zhang (zhangwei197610@wchscu.edu.cn), Yan Kang (kangyan@scu.edu.cn), Fei Xiao (icuxiaofei@scu.edu.cn)

Review Timeline:

Submission Date:	8th Oct 24
Editorial Decision:	25th Oct 24
Appeal Received:	19th Dec 24
Editorial Decision:	10th Feb 25
Revision Received:	18th Feb 25
Editorial Decision:	11th Mar 25
Revision Received:	18th Mar 25
Accepted:	26th Mar 25

Editor: Achim Breiling

Transaction Report:

Dear Prof. Zhang,

Thank you for the transfer of your manuscript to EMBO reports. I have now received the reports from the three referees that were asked to evaluate your study, which can be found at the end of this email.

I am sorry to say that the evaluation of your manuscript is not a positive one. As you will see, all referees, in particular referees #2 and #3, have major concerns regarding the experimental and clinical data, state that major claims are not sufficiently supported and also indicate that the interpretation of the data is overstated. Moreover, all three referees note several technical and experimental shortcomings. As the reports are below, I will not further detail them here.

Given the comments of the referees, the amount of work required to address them, the fact that EMBO reports can only invite revision of papers that receive overall positive support from all the referees upon initial assessment, I cannot offer to publish your manuscript.

I am sorry to have to disappoint you this time. I nevertheless hope that the referee comments will be helpful in your continued work in this area, and I thank you once more for your interest in our journal.

Yours sincerely

Referee #1:

This is an interesting study, well designed, with a clear message and just a couple of flaws.

The study design is of interest and well argued. Sham (S) surgery is performed as well as mild (M) and critical (C) sepsis is induced. It is essential that the authors show (with data) how M and C these CLPs really are, in terms of survival of mice and pathophysiological data (organ damage for example). It should be discussed what is the impact of (lack of) antibiotic application in the mice, as this is an important factor which is determining severity and similarity to human peritoneal sepsis management.

In the paper, it should be better stated to what extent previous studies using mutant mice (RAG, SCID, Nudes) have found findings that are in line with the conclusions of this paper.

The stepwise approach (S,M,C) is interesting and logical, but there is not very much attention to kinetics, besides 1 figure. Nevertheless, this is important and would deserve much more attention. There might still happen a lot later than 24h after sepsis onset.

Moreover, in Fig2 E and later, the three groups (S,M,C) are suddenly reduced to two (Sham and CLP): this is scientifically not correct and must be split up in the three groups again.

In Fig 3, I suppose these data are based on bulk RNASEQ, and so this reviewer wonders how the authors correct for gene expression in the stromal cells of these organs?

The use of Kl. pneumoniae is interesting, but falls out of a blue sky. Why this model? Why not Streptococci, why not Pseudomonas? This needs more clarification.

Also, Kl. pneumoniae is used to stress the general nature of the findings, but the therapeutic data should then also be reproduced in Kl pneumoniae, besides the CLP model.

Finally, the text needs a very profound English editing by the authors, preferentially by a professional editor. There are many language errors which should be cleared.

Referee #2:

1. Does this manuscript report a single key finding? YES/NO

If YES, please describe it in one sentence.

The authors describe their finding that, in a mouse model of sepsis, the activation of CD47-SIRP signalling triggers the

expression of amyloid- precursor protein (APP) in myeloid cells, which subsequently secrete APP cleavage products that interact with CD74 on B cells, leading to B cell suppression.

2. Is the reported work of significance (YES), or does it describe a confirmatory finding or one that has already been documented using other methods or in other organisms etc (NO)? YES/NO

While some of the results are confirmatory, there are elements of the findings that are novel.

3. Is it of general interest to the molecular biology community? YES/NO

If YES, please say why, in a single sentence. If NO, please state which more specialized community you feel it is aimed at (or none), in a single word or phrase.

The central finding of APP as a mediator of B cell immunosuppression may have wider implications.

4. Is the single major finding robustly documented using independent lines of experimental evidence (YES), or is it really just a preliminary report requiring significant further data to become convincing, and thus more suited to a longer-format article (NO)? YES/NO

General comments:

The purpose of this study was to investigate the mechanism of dysregulated immunoregulation that accompanies sepsis. The investigators used a standard CLP murine model, but in an interesting modification, compared "mild" infection with a more conventional "severe" model, seeking to understand why some infections go on to result in a severe systemic, multi-organ process that often leads to death. First, they demonstrated that CLP-induced sepsis is quickly followed by widespread and marked depression of adaptive immunity; they provided evidence to suggest that the pattern of gene expression could be used to identify sepsis in patients; and finally they demonstrated that CD47 may trigger this immunosuppression by binding to SIRP - which results in the release of amyloid - which in turn binds to CD74 leading to B cell immunosuppression. Several of these findings deserve further consideration.

Major comments:

1. Immunosuppression following sepsis is well known and has been studied extensively, including the downregulation of MRC class II genes (e.g. Winkler PLoS One 2017 12(8):e0182427), likewise the potential role of MDSC's in mitigating the inflammatory response that follows lethal abdominal sepsis in mice (e.g. Harriett et al Front Cell Infect Microbiol 2022 12:898030). The authors should make clear that the finding of immunosuppression, per se, is not novel.

2. In further experiments they investigated the topography of immune cells to see if they "migrate[d] to the potential sites of infection". In these experiments (page 7) they concluded that T/B cells infiltrate the lungs in sepsis, but not the kidneys and liver, drawing the conclusion that the lungs were a focus of infection but not the other organs. I found this surprising since CLP is essentially a systemic infection. It would have been very interesting if the authors had made bacteriological studies of all the organs to determine if the liver and kidneys, for instance, were culture negative.

3. The use of the "mild" versus "severe" models was innovative and interesting, but the data they show could have other explanations. Could it be, for instance, that the "mild" infections eventually show the same cellular and genetic changes as the 'severe' group but just get there more slowly? This question could be answered relatively easily by performing some delayed experiments.

4. The studies with *K.pneumoniae* did look at the delayed response, but I have some concerns about the model. The direct inoculation of a high dose of bacteria into the trachea is hardly a model of pneumonia; furthermore, given the relatively high dose of bacteria, it is possible that this is just, in effect, an LPS challenge model. It would be informative to repeat some of these experiments using just an LPS challenge to provide reassurance that this is truly a model of sepsis.

5. The clinical data are intriguing but raise some significant questions. The entry criterion was that the patient was "diagnosed with infection". We are not told whether these are sequential patients, or if they were selected in some way (the patient numbers in column A of Supp Table 1 are not sequential). On what basis was the diagnosis of sepsis made? It is not clear if this was just the SOFA score; if so, what cut off was used? Patient 16, for instance, has a high SOFA (6) but no indication in the clinical data that she was listed as septic. It is unfortunate that no microbiological data are provided. Supp Table 1 has some slightly odd elements. For instance, the PCT data do not entirely make sense (compare for example pt 4 (sepsis, deceased, PCT 0.66 to patient 7 (sepsis, deceased, PCT 5.5). I have never seen the PaO₂/FI₀₂ shown to 7 decimal figures; some cells are incorrect (patient 69). More importantly, I think it would have been of considerable interest to include a control group with serious non-infective injury, such as trauma. Was the SOFA score calculated on data obtained at admission? Are the authors surprised that CD11b did not show significant differences between healthy and septic patients, given the quite marked differences in neutrophil counts?

6. I was surprised that they reported immunosuppressive molecules such as IL-10 were unaffected in their experiments, given the substantial literature in this area (e.g. Zhou et al Ann Med 2024; 56:2396569).

7. The interaction between CD74 and APP has been reported previously (Liu et al Front Pharmacol 2024 15:1437113). Interestingly, those authors demonstrated that this mediated an endothelial-macrophage interaction, and one wonders whether that might be playing a role, even a dominant role, in the result of CD74 blockade described in the current paper; this merits comment. Certainly their studies with anti-CD47, E2609 and the KO are striking. I was slightly confused by their observation that widespread changes were found in the lungs and kidneys of CLP mice (page 15)(compare their results on page 7, noted above).

8. The Discussion provides a cogent analysis of their results, but as the authors acknowledge, leaves the question of the dominant effect unclear. I was surprised that the authors did not try and assess this by bacteriological studies; it would be of great interest, for instance, to perform quantitative bacterial counts in blood and organs in their models, particularly after the therapeutic interventions. While the authors have, naturally enough, focused on their findings to try and explain the immunobiology of the immune suppression seen in sepsis it might have been helpful to put it into a wider context by acknowledging the hugely complex series of changes that occur, and in particular the literature on endotypes which suggests that there is not a single "simple" explanation for the immune profile in sepsis. As a final thought, I am not sure that their data provide very strong support for the claim that certain immunity-related genes could be used to distinguish septic patients from those with mild infections.

Minor comment:

1. There are minor errors of syntax throughout that could be corrected.

Referee #3:

Feng et al. investigate the role of CD47 and a possible CD47-amyloid- β -CD74 signaling axis in sepsis.

While some of the findings reported here are potentially interesting, the manuscript lacks mechanistic coherence and solid evidence to support its central claims.

There are also considerable imprecisions in either interpretation or labeling of the data. For example, in Figure 2, MDSCs do not increase from 10%-45% in CLP. Indeed, the increase from mild to severe CLP is small, which makes it hard to accept the central role of these populations in the dramatically different outcomes between mild (no mortality) and severe CLP (high mortality).

Also, contrary to the author's initial criteria, CD47 increases at the RNA level but not at the protein level. There are just more cells, which does not really support the author's hypothesis. The manuscript heavily relies on transcriptional data, and its interpretation is overstated.

Generally, figure legends and text have unnecessary text but lack key information, including the time of collection of samples. There are also mislabelings, including the fact that the CFUs in WT vs KOs are not in Figures 6A and B. Many experiments do not seem to have had biological replicates, and three animals are certainly not enough to represent the measurements of the bacterial burdens.

More substantially, the authors do not really make a compelling case for the role of immunosuppression in the observations they make when CD47 is inhibited. How is B cell suppression related to the control of the bacterial burden? What does E2609 do to the bacterial burden? It is plausible that CD47 deficiency increases bacterial burden control by myeloid cells, but what is the role of B cell suppression in this?

The authors should look at the kinetics of bacterial burden in controls vs different forms of CD47 block. They should also uncouple what can be attributed to myeloid and B cells. What is the effect in survival, bacterial burden and organ damage of combining CD47KO and E2609 treatment? For example, using a mouse line deficient in CD74 would be desirable. A

iso, how does immunosuppression lead to organ damage? Perhaps the opposite could be expected, as less tissue damage would occur.

** As a service to authors, EMBO Press provides authors with the ability to transfer a manuscript that one journal cannot offer to publish to another journal, without the author having to upload the manuscript data again. To transfer your manuscript to another EMBO Press journal using this service, please click on
Link Not Available

Point-by-point response:

Referee #1:

This is an interesting study, well designed, with a clear message and just a couple of flaws.

1. The study design is of interest and well argued. Sham (S) surgery is performed as well as mild (M) and critical (C) sepsis is induced. It is essential that the authors show (with data) how M and C these CLPs really are, in terms of survival of mice and pathophysiological data (organ damage for example). It should be discussed what is the impact of (lack of) antibiotic application in the mice, as this is an important factor which is determining severity and similarity to human peritoneal sepsis management.

Response:

We greatly appreciate the reviewer for the valuable comments and suggestions, which were very helpful for us to further improve our work and make the points of this study more explicit.

Indeed, examining organ injuries in the control, mild and critical infection groups will help us understand the pathophysiological characteristics of each group. In the revised manuscript, we provide full scans of HE-stained sections of the lung, liver, and kidney (the images were added to Fig. EV1A-C). Compared with those in the control group, no obvious organ injury was observed in the mild group, whereas significant pathological injuries were detected in the organs from the critical group (including thickened pulmonary septa, blurred hepatic sinusoids, dilated renal tubules, etc.).

We used antibiotics to treat CLP mice as a method to simulate actual clinical sepsis. All the mice that underwent CLP surgery received antibiotics and underwent fluid resuscitation. Briefly, the mice received a single intramuscular injection of ciprofloxacin at a dose of 20 mg/kg and were subjected to subcutaneous fluid resuscitation with 800 µl of saline immediately after surgery, as described in our previous study (Xiao *et al*, 2018). This information can be found in the *Methods* section (the previous version, page 23; the current revised version, page 28). This antibiotic treatment slightly reduced mortality but could not prevent sepsis-related death, similar to the clinical situation. Considering that the antibiotic intervention may affect the body's immune response to mild and severe infections, we did not use antibiotics before collecting blood samples for scRNA sequencing and bulk RNA sequencing.

-
2. In the paper, it should be better stated to what extent previous studies using mutant mice (RAG, SCID, Nudes) have found findings that are in line with the conclusions of this paper.

Response:

We agree with the reviewer that previous studies using immunodeficient mice should be discussed in detail. The infection/sepsis-related experimental results obtained from T/B-cell-deficient mouse models (including Rag^{-/-}, SCID, and nude mice) are consistent with the findings of our study. Shelley et al. reported that Rag1^{-/-} mice have increased mortality in burn and postburn sepsis models (Shelley *et al*, 2003). Reim et al. reported that Rag1^{-/-} mice presented increased susceptibility and impaired bacterial clearance in a model of acute septic peritonitis (Reim *et al*, 2009). Klingensmith et al. used Rag^{-/-} mice to establish a CLP model and found that the survival rate of Rag^{-/-} mice was much lower than that of WT mice (approximately 8% vs. 50%) (Joshi *et al*, 2012). Kelly-Scumpia reported that Rag1^{-/-} mice display deficient early inflammatory responses and reduced survival during sepsis (Kelly-Scumpia *et al*, 2011). Kadowaki's study revealed that the use of nude mice to establish a CLP model resulted in a higher mortality rate than the use of WT mice (Kadowaki *et al*, 2013). Joshi et al. used both SCID and nude mice and found that T cells play an essential role in the IsdB vaccine-mediated defense against invasive *S. aureus* infection in mice (Joshi *et al*, 2012). A study we published in 2023 showed that only approximately 20% of SCID mice survived mild CLP surgery, whereas 100% of WT mice survived (Feng *et al*, 2023).

Previous studies of sepsis have focused mostly on the inflammation and cytokine storms caused by infection. However, in recent decades, almost all drugs that target inflammatory factors have failed, suggesting that inflammation alone does not fully reflect the immune status of the body during sepsis. Adaptive immunosuppression in sepsis, which is caused mainly by decreased T/B lymphocyte numbers and function, is gradually being recognized. Recent studies have reported that the PD-1/PD-L1 pathway may be involved in sepsis-associated immunosuppression by inhibiting T-cell functions. Hosts with T/B-cell-related immunodeficiency, such as HIV patients (CD4⁺ T-cell deficiency), organ transplant patients receiving immunosuppressive drugs, and patients who have used glucocorticoids for a long period, are more susceptible to a fatal infection. Therefore, we believe that defects in T/B-cell

function increase the vulnerability of the host to infection and sepsis, which is theoretically logical and consistent with the experimental and observational results.

These studies involving mutant mice have been added to the Discussion section; please see page 19, the 1st and 2nd paragraphs, highlighted text.

3. The stepwise approach (S,M,C) is interesting and logical, but there is not very much attention to kinetics, besides 1 figure. Nevertheless, this is important and would deserve much more attention. There might still happen a lot later than 24h after sepsis onset. Moreover, in Fig2 E and later, the three groups (S,M,C) are suddenly reduced to two (Sham and CLP): this is scientifically not correct and must be split up in the three groups again.

Response:

We thank the reviewer for pointing out this issue. In the previous manuscript, we presented the scRNA-Seq and bulk RNA-Seq data from the S, M, and C groups. However, when we used flow cytometry to verify the sequencing results, we did not establish a mild group. At that time, the mice were divided into only the sham and CLP groups (severe) to reduce the workload, which is indeed a design flaw. In the revised manuscript, we added the *mild* group again, performed flow cytometry detection, and integrated these data into Fig. 2E-N and Fig. 5E-F. After Fig. 6, we focused on the effect of intervening in the CD47 pathway on sepsis. Because all the mice that underwent mild CLP surgery survived and the intervention could not further improve the already 100% survival rate, we did not add a mild group to the experiments shown in the figures after Fig. 6.

(The data from *the mild* group have been integrated into Fig. 2E-N and Fig. 5E-F.)

As the reviewer suggested, we have also considered detecting changes in the expression of immune-related genes for a longer period after sepsis to obtain a more comprehensive understanding of the host immune profile in response to infection. However, we found that the process of sepsis development in mice is different from that in humans. Most deaths in mice after CLP surgery occurred before 3–4 days, while deaths within two weeks of sepsis in humans can be considered rapid. If mice can survive for more than 3–4 days, the mortality rate decreases sharply, and most of these mice are in

a state of recovery rather than deterioration. If we collect blood samples after 3–4 days, they will most likely be derived from the mice in the recovery stage rather than from the mice whose condition is still worsening. This condition is different from clinical sepsis patients. From diagnosis to 3–4 days later, patients with sepsis are likely to be in the process of sepsis progression rather than in a state of recovery.

Since mice die after samples are collected, in the supplementary experiment, we could perform only a "snapshot" detection method to collect samples from two batches 3 days and 7 days later. In the first batch of mice, 5/10 were still alive 3 days after the CLP surgery, and so 5 samples were collected; in the second batch of mice, 4/10 were still alive 7 days after the CLP surgery, and so 4 samples were collected.

The pneumonia sepsis model established by *Klebsiella pneumoniae* infusion has a more distinct disease process than CLP surgery. Almost all the mice that survived 3 days after the *Klebsiella pneumoniae* infusion eventually recovered. In addition, although the same dose of *Klebsiella pneumoniae* (10^9 CFUs) was administered, the mortality rate of the mice in each batch may be different. We conducted two batches of *Klebsiella pneumoniae* infusion experiments. A total of 5/10 of the mice in the first batch survived for 3 days after the *Klebsiella pneumoniae* infusion, and 7/10 of the mice in the second batch survived.

The experimental results revealed that the expression of adaptive immune-related genes, such as CD3, CD8, CD19, Pax5, H2-Eb, and H2-Aa, recovered after 7 days compared with 24 hours, regardless of the CLP surgery or *K. pneumoniae* infusion, suggesting that the adaptive immune system of these mice was in a state of recovery. These results have been integrated into Fig. EV16 and Fig. EV17 in the revised manuscript.

-
4. In Fig 3, I suppose these data are based on bulk RNASEQ, and so this reviewer wonders how the authors correct for gene expression in the stromal cells of these organs?

Response:

As the reviewer pointed out, bulk RNA-seq cannot distinguish the cell from which the expressed

gene originates. Only scRNA-seq can distinguish which genes are expressed from which cells. In this study (Fig. 3), we used bulk RNA-seq to detect gene expression in the main immune compartments (peripheral blood, spleen, lymph nodes, and bone marrow), which contain various immune cells. When comparing gene expression, we used FPKM values instead of direct sequencing values (gene counts). FPKM corrects for the sequencing depth and gene length rather than direct read counts. Compared with the results of scRNA-Seq, the expression of adaptive immune-related genes, including CD19 (which is specifically expressed by B cells), CD3 (a T-cell marker), and MHC class II genes (which are expressed mainly by B cells and some myeloid cells in mice), decreased in a stepwise manner. We also performed bulk RNA-seq of lung, kidney, and liver tissues to analyze the expression of these immune-related genes. Since these immune-related genes are expressed by specific immune cells and are rarely expressed in the parenchymal and interstitial cells of these organs, the increase in their mRNA levels can be considered an indicator of whether specific immune cells have migrated to these organs. Therefore, we selected genes that are specifically expressed by certain immune cells for analysis, which may circumvent the issue that these genes may be expressed in other stromal cells.

5. The use of *Kl. pneumoniae* is interesting but falls out of a blue sky. Why this model? Why not *Streptococci*, why not *Pseudomonas*? This needs more clarification.

Response:

In recent years, we have attempted to establish more sepsis models other than those generated with CLP surgery, including those involving the use of *Klebsiella pneumoniae*, *Staphylococcus aureus*, *Pseudomonas aeruginosa*, and influenza virus. Airway intubation and injection of bacteria or viruses have been used to simulate pneumonia and sepsis caused by pathogenic microorganisms (Luna *et al*, 2009). London *et al.* used H5N1 influenza virus to directly infect mice and establish a viral sepsis model (London *et al*, 2010). Zhao *et al.* used *Klebsiella pneumoniae* (*K. pneumoniae*) to establish systemic infection-induced sepsis in mice (Zhao *et al*, 2024). Joshi *et al.* established a mouse sepsis model via the intranasal inoculation of *K. pneumoniae* (Joshi *et al*, 2024). Byrnes *et al.* established a rat sepsis model via the intratracheal instillation of *K. pneumoniae* (Byrnes *et al*, 2023). Sécher *et al.* established a mouse model of acute pneumonia by administering *Pseudomonas aeruginosa* through

airway intubation (Secher *et al*, 2019).

Clinical data show that *K. pneumoniae* infection is now one of the most common sources of nosocomial infection in the ICU. The experimental results revealed that the *K. pneumoniae*-induced infection model was stable and can cause severe sepsis at a dose of 10^9 CFUs (we tested 4 different doses of 10^7 CFUs, 10^8 CFUs, 10^9 CFUs and 10^{10} CFUs; please see Fig. EV15A). The mortality rate of sepsis caused by airway intubation in *K. pneumoniae* was approximately 30–70%, with slightly greater heterogeneity than CLP surgery. This deviation can be compensated for by increasing the number of mice in each group (generally ≥ 15 mice/group). Each animal sepsis model should simulate specific clinical events under existing technical conditions, such as CLP surgery, which can be used to simulate sepsis caused by severe abdominal infection; influenza virus infection, which can be used to simulate viral sepsis; and *K. pneumoniae* could be used to simulate sepsis caused by severe pneumonia. Although airway intubation of *K. pneumoniae* cannot completely replicate severe clinical pneumonia, considering that *K. pneumoniae* is a common nosocomial infection bacteria in the ICU, and the incidence of *K. pneumoniae* has increased in recent years, we believe that the use of *K. pneumoniae* is appropriate to establish a pneumonia-induced sepsis model.

We have added the rationale for using *K. pneumoniae* to model sepsis in the Discussion section. Please see page 20, the last paragraph, highlighted text.

-
6. Also, Kl. pneumoniae is used to stress the general nature of the findings, but the therapeutic data should then also be reproduced in Kl pneumoniae, besides the CLP model.

Response:

The survival curve of the *K. pneumoniae* model is slightly different from that of the CLP model. After day 4, the death of the experimental mice became rare. In this model, the therapeutic effect of blocking the CD47 pathway was also slightly lower than that in the CLP model. This result may be because the amount of bacteria infused at one time is too large, which results in greater "shock" to the mice than does CLP surgery. We have added these results to Fig. 8B. The main text was revised accordingly; please see page 17, the 1st paragraph, highlighted text.

-
7. Finally, the text needs a very profound English editing by the authors, preferentially by a professional editor. There are many language errors which should be cleared.

Response:

The revised manuscript was edited for proper English language, grammar, punctuation, spelling, and overall style by qualified English-speaking editors at AJE. This certificate can be verified on the AJE website using the verification code CEA-809F-A1D4-F6A7-5B0P.

Point-by-point response:

Referee #2:

General comments:

The purpose of this study was to investigate the mechanism of dysregulated immunoregulation that accompanies sepsis. The investigators used a standard CLP murine model, but in an interesting modification, compared "mild" infection with a more conventional "severe" model, seeking to understand why some infections go on to result in a severe systemic, multi-organ process that often leads to death. First, they demonstrated that CLP-induced sepsis is quickly followed by widespread and marked depression of adaptive immunity; they provided evidence to suggest that the pattern of gene expression could be used to identify sepsis in patients; and finally they demonstrated that CD47 may trigger this immunosuppression by binding to SIRP - which results in the release of amyloid - which in turn binds to CD74 leading to B cell immunosuppression. Several of these findings deserve further consideration.

Major comments:

1. Immunosuppression following sepsis is well known and has been studied extensively, including the downregulation of MRC class II genes (e.g. Winkler PLoS One 2017 12(8):e0182427), likewise the potential role of MDSC's in mitigating the inflammatory response that follows lethal abdominal sepsis in mice (e.g. Harriett et al Front Cell Infect Microbiol 2022 12:898030). The authors should make clear that the finding of immunosuppression, per se, is not novel.

Response:

We greatly appreciate the reviewer for the valuable comments and suggestions, which were very helpful for us to further improve our work and make the points of this study more explicit.

We completely agree with the reviewer that immunosuppression following sepsis has been proposed and intensively studied. Although the theory that an "uncontrolled inflammatory response causes

sepsis" has prevailed since the 1970s, investigators began to challenge this theory as early as the 1990s (Lederer *et al*, 1999; Natanson *et al*, 1994; Oberholzer *et al*, 2001; Weighardt *et al*, 2000). Weighardt reported that postoperative sepsis was associated with an immediate defect in the production of both inflammatory and anti-inflammatory cytokines by monocytes and that the survival of patients with sepsis was associated with the recovery of the inflammatory response but not the anti-inflammatory response, and they proposed that immunosuppression was a primary rather than a compensatory response to sepsis (Weighardt *et al.*, 2000). The role of immunosuppression in sepsis was reemphasized in 2013. Hotchkiss *et al.* published a famous review in *Nat Rev Immunol*, suggesting that immunosuppression may play an equally critical role as inflammation in the pathogenesis of sepsis (Hotchkiss *et al*, 2013). A reduction in HLA-DR in sepsis was first reported in 1994, in which Lin *et al.* found that HLA-DR expression was significantly reduced in patients with severe sepsis (Lin *et al*, 1994). HLA-DR has gradually become a biomarker for evaluating the immune status of patients with sepsis.

MDSCs were initially reported mainly in cancer research. Sander and colleagues may be the first to report that MDSCs may be involved in sepsis (Sander *et al*, 2010). We have cited and discussed previous studies of MDSCs in the Discussion section (please see previous version, page 16-17). Regarding the role of MDSCs in sepsis, the consensus is that this group of cells is characterized by anti-inflammatory and immunosuppressive activities. Researchers have not yet reached a consensus on whether MDSCs improve or worsen the survival of patients with sepsis (Lai *et al*, 2014). Earlier studies suggested that the anti-inflammatory and immunosuppressive properties of MDSCs could improve the survival of septic mice (Derive *et al*, 2012; Sander *et al.*, 2010). On the other hand, studies in recent years reported that reducing the number of MDSCs can ameliorate immunosuppression and improve the survival rate of septic mice (Chen *et al*, 2022; Liang & Shen, 2020; Zhang *et al*, 2021). Studies examining the effects of MDSCs on lymphocytes have focused mainly on their ability to regulate T cells. MDSCs may suppress T-cell activity via IL-10 (Bah *et al*, 2018) or PD-L1 (Ruan *et al*, 2020), but whether MDSCs affect B cells in individuals with sepsis has not yet been reported.

Here, we did not intend to reinvent the concept of immunosuppression in sepsis, nor did we intend to claim that we were the first to discover the reduction in HLA-DR expression and the increase in the number of MDSCs in a sepsis model. We apologize if the manuscript left such an impression. We will

review the manuscript to remove statements that may leave such an impression. The above description of previous studies on immunosuppression in sepsis has been added to the Discussion section; please see page 19-20, highlighted text.

2. In further experiments they investigated the topography of immune cells to see if they "migrate[d] to the potential sites of infection". In these experiments (page 7) they concluded that T/B cells infiltrate the lungs in sepsis, but not the kidneys and liver, drawing the conclusion that the lungs were a focus of infection but not the other organs. I found this surprising since CLP is essentially a systemic infection. It would have been very interesting if the authors had made bacteriological studies of all the organs to determine if the liver and kidneys, for instance, were culture negative.

Response:

We thank the reviewer for pointing out this important issue. CLP is a surgery that causes intestinal perforation and bacterial leakage, which leads to a severe abdominal infection. It was somewhat unexpected that the abdominal infection caused by CLP could result in such a high bacterial load in the peripheral blood. Moreover, whether such high bacterial loads exist in major organs, such as the kidneys, liver, and lungs, is an interesting and very important question worth exploring.

We performed bacterial tests on major organs (kidney, liver, and lung) and found that the systemic infection caused by CLP is likely to cause bacterial infections in other organs. Blood agar plate culture showed that a large number of bacteria was detected in the tissue homogenates of the kidney, liver, and lung of mice in the CLP group compared with the sham group. A small number of bacteria was also detected in the lungs of the sham group, but these bacteria were likely bacteria that colonized the lungs rather than bacteria from an active infection.

We thank the reviewer for raising such a good question. The overactivated host response induced by a specific infection site (such as the abdominal infection caused by CLP) is believed to damage other organs outside the initial infection site. These bacterial detection results suggest that the CLP-induced abdominal infection is likely to cause bacteria to spread to other organs. Therefore, the "battlefield" for the host to combat pathogens may not be limited to the initial site of infection but may spread to other major organs and cause organ damage, which may explain why multiple organ injury occurs in sepsis.

(The results of the bacteriological detection of major organs were added to Fig. EV22 in the revised manuscript.)

3. The use of the "mild" versus "severe" models was innovative and interesting, but the data they show could have other explanations. Could it be, for instance, that the "mild" infections eventually show the same cellular and genetic changes as the "severe" group but just get there more slowly? This question could be answered relatively easily by performing some delayed experiments.

Response:

In this study, we established mild and severe sepsis models through CLP surgery with different severities. Briefly, when the mild model was established, the cecum was ligated 2.5 mm from the cecal tip and perforated by a single "through and through" puncture with a very thin needle (26 gauge). The mice that underwent mild CLP surgery did not show severe symptoms, and no bacteria were detected in the blood (as shown in Fig. 6B); all the mice survived, and the symptoms caused by mild CLP surgery generally disappeared after 3–5 days. No mice in the mild group died within 30 days after surgery. For the severe model, the cecum was ligated 7.5 mm from the cecal tip and perforated with a thick needle (18 gauge); approximately 50-70% of the mice died from sepsis caused by a severe abdominal infection (the descriptions of CLP surgeries of different severities can be found on page 4 of the main text).

The difference between the mild and severe CLP models is mainly related to the length of the ligated intestinal segment and the thickness of the puncture needle, which causes different amounts of intestinal contents to leak into the abdominal cavity. The "severe" CLP surgery causes significantly more intestinal contents to leak into the abdominal cavity, resulting in a stronger host response and more severe systemic infection, as evidenced by the high bacterial burden in the blood and the death of more than half of the mice. On the other hand, the "mild" CLP surgery causes only a small amount of leakage of the intestinal contents and only mild symptoms. No bacteremia occurred, and no mice died. Therefore, the factors leading to a "mild" or "severe" infection are determined by the severity of the CLP surgery. According to the experimental results, the "mild" cases will not develop into "severe"

cases over time but recover quickly.

4. The studies with *K.pneumoniae* did look at the delayed response, but I have some concerns about the model. The direct inoculation of a high dose of bacteria into the trachea is hardly a model of pneumonia; furthermore, given the relatively high dose of bacteria, it is possible that this is just, in effect, an LPS challenge model. It would be informative to repeat some of these experiments using just an LPS challenge to provide reassurance that this is truly a model of sepsis.

Response:

Airway intubation and injections of bacteria or viruses have been used to simulate pneumonia and sepsis caused by pathogenic microorganisms (Luna *et al.*, 2009). London et al. used H5N1 influenza virus to directly infect mice and establish a viral sepsis model (London *et al.*, 2010). Zhao et al. used *Klebsiella pneumoniae* (*K. pneumoniae*) to establish a mouse model of systemic infection-induced sepsis (Zhao *et al.*, 2024). Joshi et al. established a mouse sepsis model via the intranasal inoculation of *K. pneumoniae* (Joshi *et al.*, 2024). Byrnes et al. established a rat sepsis model via the intratracheal instillation of *K. pneumoniae* (Byrnes *et al.*, 2023). Sécher et al. established a mouse model of acute pneumonia by administering *Pseudomonas aeruginosa* through airway intubation (Secher *et al.*, 2019).

In recent years, we have attempted to establish more sepsis models other than CLP surgery, including the use of *Klebsiella pneumoniae*, *Staphylococcus aureus*, *Pseudomonas aeruginosa*, and influenza virus. Clinical data show that *K. pneumoniae* infection is now one of the most common sources of nosocomial infection in the ICU. The experimental results showed that the *K. pneumoniae*-induced infection model was stable and could cause severe sepsis at a dose of 10^9 CFUs (we tested 4 different doses of 10^7 CFUs, 10^8 CFUs, 10^9 CFUs and 10^{10} CFUs; please see Fig. EV15). The mortality rate of sepsis caused by airway intubation with *K. pneumoniae* is approximately 30–70%, and its heterogeneity is slightly greater than that of the CLP surgery. This deviation can be compensated for by increasing the number of mice in each group (generally ≥ 15 mice/group). Each animal sepsis model should simulate specific clinical events under the existing technical conditions, such as CLP surgery, which can be used to simulate sepsis caused by severe abdominal infection; influenza virus infection, which can be used to simulate viral sepsis; and *K. pneumoniae*, which can be used to simulate sepsis

caused by severe pneumonia. Although airway intubation with *K. pneumoniae* cannot completely replicate severe pneumonia in the clinic, considering that *K. pneumoniae* is a common nosocomial infection bacterium in the ICU and that the incidence of *K. pneumoniae* has been increasing in recent years, we believe that the use of *K. pneumoniae* to establish a pneumonia sepsis model is appropriate.

We analyzed the mortality of mice caused by different doses of LPS. The lethal dose of LPS for mice was determined by initially administering an intraperitoneal injection of 10 mg/kg of LPS (a concentration lower than 5 mg/kg generally causes symptoms but not death). According to our experimental results, a dose of 10 mg/kg can cause approximately 20% mortality; when the dose reaches 25 mg/kg, the mortality rate can reach 100%. The death patterns caused by LPS, CLP surgery, or bacterial administration differ. After CLP surgery or bacterial administration, most of the deaths of the mice occurred before 3–4 days, and most of the surviving mice remained alive thereafter. These findings suggest that if a mouse can survive for more than 3–4 days, the mortality rate is significantly reduced, and the surviving mice are in a state of recovery rather than deterioration. In the previous version of this manuscript, we examined the expression of immune-related genes in mice 8 h and 24 h after CLP surgery or *K. pneumoniae* delivery. In the revised manuscript, we added an observational experiment to analyze the expression of immune-related genes 3 days and 7 days after CLP or *K. pneumoniae* insult. The results revealed that the expression of immune-related genes, such as CD3, CD8, CD19, Pax5, H2-Eb, and H2-Aa, recovered after 7 days compared with 24 hours, suggesting that the adaptive immune system of these mice was likely to be in a state of recovery.

However, the death pattern caused by LPS injection was different. As shown in the Kaplan–Meier curve, only 20% of the mice died within 3 days after the injection of 25 mg/kg LPS, whereas 80% of the mice died from the 4th day onward. These findings suggest that the death caused by LPS is similar to gradual organ failure caused by toxic reactions. In sepsis induced by bacterial infection, the mice that survived after 4–5 days were generally in the recovery stage; however, 4–5 days after the injection of a large dose of LPS, the mice continued to die in large numbers, indicating that organ failure was still progressing.

The mechanism of sepsis caused by the airway administration of bacteria is similar to that of CLP surgery, both of which induce sepsis by introducing a large number of bacteria into the body in a short period. The mechanism of LPS is somewhat different. LPS is one of the components of bacteria and

cannot reproduce itself in the body like bacteria. The immune response caused by LPS is also different from that caused by bacteria. According to a consensus meeting on animal sepsis models held in Vienna in May 2017, several principles were recommended to improve the quality of preclinical studies, including that the microorganisms used in animal models should replicate those commonly found in humans with sepsis, that sepsis models could be initiated at sites other than the peritoneal cavity, and that challenge with LPS is not an appropriate model for replicating sepsis in humans (Osuchowski *et al*, 2018). A good animal sepsis model should simulate a specific clinical event under the current technical conditions, but no specific clinical event corresponds to the administration of large doses of LPS. The commonly used doses of LPS (5–25 mg/kg) result in blood concentrations (50–250 µg/ml) in mice that are approximately 10^4 – 10^5 times higher than the median endotoxin level in the blood of patients with sepsis (100–700 pg/ml). The death of animals caused by high-dose LPS is more likely due to toxic reactions rather than a dysregulated host immune response caused by bacteria. Therefore, some investigators have proposed that LPS injection may serve as a model for endotoxic shock but not for sepsis (Riedemann *et al*, 2003).

(Supplemental LPS challenge experiments and prolonged observations of immune-related gene expression in the CLP and *K. pneumoniae* models have been added to Figs. EV15-17; the main text has been revised accordingly. Please see page 9, the high-light text.)

5. The clinical data are intriguing but raise some significant questions. The entry criterion was that the patient was "diagnosed with infection". We are not told whether these are sequential patients, or if they were selected in some way (the patient numbers in column A of Supp Table 1 are not sequential). On what basis was the diagnosis of sepsis made? It is not clear if this was just the SOFA score; if so, what cut off was used? Patient 16, for instance, has a high SOFA (6) but no indication in the clinical data that she was listed as septic. It is unfortunate that no microbiological data are provided. Supp Table 1 has some slightly odd elements. For instance, the PCT data do not entirely make sense (compare for example pt 4 (sepsis, deceased, PCT 0.66 to patient 7 (sepsis, deceased, PCT 5.5). I have never seen the PaO₂/FI₀2 shown to 7 decimal figures; some cells are incorrect (patient 69). More importantly, I think it would have been of considerable interest to include a control group with serious non-infective injury, such as trauma. Was the SOFA score

calculated on data obtained at admission? Are the authors surprised that CD11b did not show significant differences between healthy and septic patients, given the quite marked differences in neutrophil counts?

Response:

We thank the reviewer for pointing out these detailed issues, and addressing them one by one will help us improve the quality of the manuscript.

The entry criterion was that the patient was "diagnosed with an infection", which means that these patients were diagnosed with an "infection" by the emergency physician on duty. At this time point, these patients generally only underwent routine blood tests, and the emergency physician made a preliminary diagnosis based on the chief complaint, clinical symptoms and routine blood test results. The test results used for SOFA scoring were either not performed or had not yet returned. The reason for choosing this time point is that we wanted to assess the patient's immune status earlier than SOFA scoring by detecting the expression levels of the immune markers of interest. They were not sequential patients because not all emergency department patients were diagnosed with an infection, and they were not included on the same day; therefore, their record numbers were not consecutive.

The diagnosis of sepsis was made after a SOFA score was calculated for the patient after sufficient clinical and laboratory results had been returned, including the PaO₂/FiO₂ (mmHg), platelet count (10⁹/L), bilirubin level (μmol/L), mean arterial pressure, GCS, and creatinine level (μmol/L) (the above information can be found on page 10 of the manuscript). The diagnosis of sepsis follows the criteria of Sepsis-3: infection plus evidence of organ dysfunction (SOFA_≥2) can be diagnosed as sepsis.

Patient #16 is a special case. According to the evaluation of the clinical team, the patient's low P/F value and impaired consciousness were temporary symptoms caused by neck swelling rather than an increase in the SOFA score attributed to organ dysfunctions, such as those in the lungs, heart, and brain, caused by infection. Therefore, she was not diagnosed with sepsis during the clinical consultation. We have specifically provided a note in the last column (Remarks) in Supplementary Table 1: "Patient #16 had a SOFA score of 6 but was not diagnosed with sepsis because the low PaO₂/FiO₂ and GCS scores were likely due to asphyxia because of neck soft tissue swelling rather than an infection".

We thank the reviewer for suggesting that the microbiological test results should be included. Since

these patients came from the emergency department, some of them with mild symptoms were discharged quickly and therefore did not undergo microbiological testing. However, as the reviewer noted, microbiological tests are important tests that can help us interpret the results of the study; thus, in the revised manuscript, we traced the test results by the patient's medical record numbers and added the available microbiological data to Supplementary Table 1. The symbol "\" indicates that the patient did not undergo microbiological testing before discharge.

PCT is a biomarker widely used in clinical practice to determine whether a patient has an infection. PCT can distinguish whether a patient has an infection or a noninfectious disease well, but its sensitivity and specificity are not sufficient to distinguish sepsis from a common infection (or SIRS). A systematic review published in *Lancet Infect Dis* in 2007 reported that "...Overall, the diagnostic performance of procalcitonin was low, with mean values of both sensitivity and specificity being 71% (95% CI 67-76) and an area under the summary receiver operating characteristic curve of 0.78 (95% CI 0.73-0.83)... "(Tang *et al*, 2007). A multicenter, prospective, observational study reported that "...both CRP and PCT elevated was also not an independent predictor in multivariate analysis. Initial levels of CRP and PCT alone and their combinations in septic shock patients had a limitation to predict 28-day mortality..."(Ryoo *et al*, 2019). From the evidence sources in the available literature, the power of PCT in differentiating between infectious and noninfectious forms of systemic inflammatory response syndrome in adults and in stratifying morbidity and mortality risks is limited(Aloisio *et al*, 2019). A recent systematic review published in 2023 analyzed 60 studies including 15,681 patients and reported that PCT, CRP, IL-6, and sCD14 were not shown to help predict the mortality of critically ill patients with sepsis(Molano-Franco *et al*, 2023). Overall, PCT is helpful in distinguishing infected patients from noninfected patients, and the existing evidence supports the use of PCT as a biomarker to improve the diagnosis of bacterial infections and guide antibiotic therapy. However, the power of PCT is not sufficient to predict the sepsis prognosis. A significant correlation between high or low PCT levels and the final outcome of survival or death was not observed. Therefore, for an individual patient, predicting the prognosis based on the PCT value is difficult.

The value of PaO₂/FiO₂ is obtained by dividing the former (PaO₂) by the latter (FiO₂). Some Excel cells were formatted to 7 decimal places, resulting in 7 decimal figures when generating this value. The use of these significant digits is due to our personal negligence. We did not notice this error when

checking the table. We are very grateful to the reviewer for this reminder.

According to the reviewers' suggestion, we included 10 noninfectious, posttraumatic or postsurgical patients with SOFA scores ranging from 3 to 8. Droplet digital PCR revealed that the immune-related gene expression profiles of these posttraumatic/operative patients were similar to those of patients with common infections, except that the expression levels of CD5 and CD8A were low, similar to those of patients with sepsis.

(The newly added data were integrated into Fig. 4; the information of the newly added patients was integrated into Supplementary Table 1.)

For CD11b, after peripheral blood samples were collected, neutrophils were removed from the blood via standard Ficoll gradient centrifugation to isolate PBMCs (please see Fig. EV6). Among all leukocytes in peripheral blood, neutrophils account for the vast majority, especially after bacterial infection. The reason for removing neutrophils from PBMCs is that the proportion of neutrophils in the blood after infection is too high (usually >80%), which will mask changes in the expression of genes in other immune cells. As the reviewer noted, if neutrophils remain in PBMCs for testing, CD11b expression will be significantly increased. The ROC curve in Figure 4I shows that there is no difference in the expression of CD11b in PBMCs between patients with sepsis and patients with common infections. Nevertheless, if only healthy people and infected patients are compared, CD11b expression is actually increased even in neutrophil-depleted PBMCs (*Mann-Whitney test, p=0.0165*).

-
6. I was surprised that they reported immunosuppressive molecules such as IL-10 were unaffected in their experiments, given the substantial literature in this area (e.g. Zhou et al *Ann Med* 2024; 56:2396569).

Response:

We thank the reviewer for noting this issue, and we assessed the expression of IL-10 again. The results of single-cell sequencing revealed that in this batch of experiments, the expression of IL-10 in the peripheral blood PBMCs from the sham (Sh), mild (M), and critical (C) groups was low, and a significant difference was not observed among the three groups (Supplementary Fig. EV18B, C). Since

the results were from the scRNA-Seq result in the CLP mouse model, we analyzed the public human scRNA-Seq dataset (GSE167363) and found that the expression of CTLA4 and IL-10 in PBMCs of healthy volunteers and sepsis patients was low and did not show significant differences. We speculate that this may be because IL-10 is a secretory cytokine that may be released into the blood once when sepsis occurs, but it may not be continuously produced in immune cells, so we did not observe obvious changes in the mRNA level. We thank the reviewer for pointing out this issue. We have added the supplemented results to Supplementary Fig. EV18D of the revised manuscript.

7. The interaction between CD74 and APP has been reported previously (Liu et al Front Pharmacol 2024 15:1437113). Interestingly, those authors demonstrated that this mediated an endothelial-macrophage interaction, and one wonders whether that might be playing a role, even a dominant role, in the result of CD74 blockade described in the current paper; this merits comment. Certainly their studies with anti-CD47, E2609 and the KO are striking. I was slightly confused by their observation that widespread changes were found in the lungs and kidneys of CLP mice (page 15)(compare their results on page 7, noted above).

Response:

Our study started in 2021, and thus we did not retrieve this paper (Liu et al., Front Pharmacol 2024 15:1437113) during the period of our study. We thank the reviewer for providing this information.

In this study, we focused on the immune pathway mediated by APP-CD74 signaling because the samples we used for scRNA-seq and bulk RNA-seq were from the immune compartments, including the peripheral blood, spleen, lymph nodes, and bone marrow. However, our results are somewhat different from those of Liu. In this study, we found that APP was specifically expressed in myeloid cells and that the expression of APP almost overlapped with the expression of CD11b (a pan marker of all myeloid cells) (Please see Fig. 6A and Fig. 7D, where the expression profiles of CD11b, SIRP α and APP can be compared; we combined the images and put them below). Moreover, CD74 was expressed mainly in mouse B cells, and a small portion of CD74 was expressed in myeloid cells (Fig. 7C, D). On the other hand, Liu reported that APP was expressed by endothelial cells, whereas macrophages expressed CD74, which was different from our results. These results indicate that the cell-cell

communication between immune cells and nonimmune cells in parenchymal organs may differ from that between immune cells and nonimmune cells.

In this study, we focused mainly on the APP–CD74 interaction between immune cells. Undoubtedly, the APP–CD74 pathway may also play an important role in the interactions between immune cells and nonimmune cells. For example, it would be interesting to explore whether the interaction between myeloid cell-derived APP and CD74 on endothelial cells or other types of cells is involved in the progression of Alzheimer's disease. The potential mechanisms and biological effects of APP-CD74 signaling between myeloid cells and B cells or between myeloid cells and other nonimmune cells deserve further exploration.

Acute lung injury (ALI) and acute kidney injury (AKI) are the most common complications of sepsis and septic shock. Although CLP simulates sepsis caused by the leakage of many bacteria from the intestine into the abdominal cavity, it can also cause ALI and AKI in mice, as verified by a pathological

examination. Previous studies(Deng *et al*, 2018; Weis *et al*, 2017), our previous study(Xiao *et al*, 2018), and our ongoing studies revealed that CLP can cause ALI and AKI, which are manifested mainly by the thickening of the lung septa, increased lung exudation, and thinning of the brush border of the renal tubules.

However, CLP surgery causes a severe abdominal infection, and how it causes ALI and AKI remains to be elucidated. According to the classic theory of sepsis pathogenesis, ALI and AKI are likely caused by the systemic inflammatory response (SIRS). This concept was first proposed by Lewis Thomas in a review published in *The New England Journal of Medicine* in 1972, noting that "...It is our response that makes the disease. Our arsenals for fighting off bacteria are so powerful... that we are more in danger from them than the invaders."(Thomas, 1972) This theory implies that although an actual infection may not be present in these organs, the strong immune response induced by the severe abdominal infection caused by CLP surgery may also damage these organs.

Interestingly, according to the reviewer's suggestion, we conducted bacteriological tests on the lungs, kidneys, and liver, and detected high bacterial burden in these organs, indicating that the systemic infection caused by CLP may cause bacteria to spread to these organs and that the pathological changes in the lungs and kidneys may be because these organs themselves have become "battlefields" rather than "collateral damage" caused by the immune system. We have long believed that a systemic immune response is initiated by a specific site of infection and that this systemic response can lead to damage to organs beyond the initial site of infection. The high bacterial burden in major organs suggest that the systemic infection caused by CLP is likely to cause bacteria to spread to other organs. This result suggests that the "battlefield" for the host against pathogens may not be limited to the initial site of infection, and the "battlefield" for the host to combat invaders also takes place in other major organs, which may be a potential mechanism to explain the organ injury in individuals with sepsis. We thank the reviewer very much for this suggestion, which provides a new perspective for analyzing the organ dysfunction associated with sepsis.

Then why do the H&E staining results of the lungs and kidneys on page 15 show significant pathological changes, while the bulk RNA-Seq results on page 7 only show obvious changes in the expression of immune-related genes in the lungs, but no obvious changes in the kidneys? We speculate that this difference may be because the lungs are the end of the respiratory tract, which is directly

connected to the outside environment, different types of colonizing bacteria exist even when the host is in a healthy state, and immune cells themselves tend to migrate toward the lungs and fight potential infections. However, the kidneys are not directly connected to the outside environment, and they usually do not have colonizing bacteria. The probability of direct contact with invading pathogens in the urinary tract is also lower than that in the respiratory tract. This observation may explain why changes in the expression of immune-related genes in the kidneys are not as significant as those in the lungs. Nevertheless, although the changes in the expression of adaptive immune-related genes were not significant in the kidney and liver, as shown in Fig. 3X, the level of CD11b, a marker of all monocytes–macrophages, was significantly increased, suggesting that monocytes–macrophages may migrate to the infection site. These experimental results suggest that both innate and adaptive immune responses may be prominent in the lungs, but in the kidney, the mobilization of innate immune cells may be dominant.

8. The Discussion provides a cogent analysis of their results, but as the authors acknowledge, leaves the question of the dominant effect unclear. I was surprised that the authors did not try and assess this by bacteriological studies; it would be of great interest, for instance, to perform quantitative bacterial counts in blood and organs in their models, particularly after the therapeutic interventions. While the authors have, naturally enough, focused on their findings to try and explain the immunobiology of the immune suppression seen in sepsis it might have been helpful to put it into a wider context by acknowledging the hugely complex series of changes that occur, and in particular the literature on endotypes which suggests that there is not a single "simple" explanation for the immune profile in sepsis. As a final thought, I am not sure that their data provide very strong support for the claim that certain immunity-related genes could be used to distinguish septic patients from those with mild infections.

Response:

The reviewer gave us a good suggestion to perform bacteriological testing on major organs in addition to blood, which provides new insights into the pathogenesis of sepsis. Based on the traditional concept that an uncontrolled inflammatory response causes sepsis, performing bacteriological testing on every organ seems unnecessary. Since "...It is our response that makes the disease (sepsis)...we are

more in danger from them than the invaders..."(Thomas, 1972), it seems less important to determine whether bacteria have spread to other organs outside the primary infection site. However, the results of bacterial detection in major organs indicate that the systemic infection caused by CLP is likely to cause bacteria to spread to other organs. This result suggests that the "battlefield" for the host against pathogens may not be limited to the initial site of infection, and the "battlefield" for the host to combat bacteria also takes place in other major organs, which may be a potential mechanism to explain multiorgan injuries in individuals with sepsis.

We fully agree with the reviewer that a "simple" explanation for the immune profile in patients with sepsis may not exist. Some researchers regard the inflammatory response as a "necessary evil" because inflammatory mediators activate both the innate and adaptive immune responses (Nedeva *et al*, 2019). Conceivably, during and after a fierce battle (host immunity vs. microorganisms), many traces will remain on the battlefield. The increased levels of inflammatory factors indicate the intensity of the battle, and the increase in the number of dead cells (including dead immune cells, epithelial cells, endothelial cells, parenchymal cells, etc.) is the debris left on the battlefield. Focusing too much on these battle traces is unlikely to provide much help in analyzing the nature of sepsis, and trying to erase the traces of war rather than eliminating the causes that triggered the war is unlikely to cure sepsis.

We agree with the reviewer that more evidence is needed to support the claim that certain immune-related genes can be used to distinguish patients with sepsis from patients with mild infection. At this stage, we included 20 healthy volunteers, 51 infected patients, and 10 non-infected patients with organ dysfunction (SOFA>2). Since the sample size is not large enough, it can be regarded as a preliminary proof-of-concept experiment. In the next stage, these immune-biomarker combinations will be continuously optimized by deleting or adding new candidates. We hope to eventually obtain a set of immune-biomarkers to establish an immune scoring system for the diagnosis and immunophenotyping of sepsis. In order to avoid the impression that the immune-related genes mentioned in the paper have been proven to distinguish sepsis from common infections, we have added the above to the *Discussion* section, please see page 23, the last paragraph of *Discussion*, high-lighted text.

Minor comment:

There are minor errors of syntax throughout that could be corrected.

Response:

The revised manuscript was edited for proper English language, grammar, punctuation, spelling, and overall style by qualified English-speaking editors at AJE. This certificate can be verified on the AJE website using the verification code CEA-809F-A1D4-F6A7-5B0P.

Point-by-point response:

Referee #3:

1. Feng et al. investigate the role of CD47 and a possible CD47-amyloid- β -CD74 signaling axis in sepsis. While some of the findings reported here are potentially interesting, the manuscript lacks mechanistic coherence and solid evidence to support its central claims. There are also considerable imprecisions in either interpretation or labeling of the data. For example, in Figure 2, MDSCs do not increase from 10%-45% in CLP. Indeed, the increase from mild to severe CLP is small, which makes it hard to accept the central role of these populations in the dramatically different outcomes between mild (no mortality) and severe CLP (high mortality).

Response:

We thank the reviewer for taking the time to review our manuscript. In Fig. 2, we show that the most obvious changes were in the ratio and number of myeloid cells and lymphocytes. The proportion of lymphocytes decreased from 90% in the Sham group to 45% in the Critical infection group, while MDSCs increased from 10% in the Sham group to 45% in the Critical group. Fig. 2D (please see below) clearly showed that the trend of change was from normal state (Sham) to Critical (sepsis), not from Mild to Critical.

(Fig. 2D)

We must clarify here that our main claim is not that the difference in MDSC numbers between the mild and critical groups plays a central role and leads to different severity. As we have often observed in clinical practice, two patients may both have increased WBC counts (>10,000) and increased neutrophil ratios (>80%), but one is just suffering from a common infection while the other is a sepsis patient. If we only look at the increase in the number of myeloid cells, then the increase from the Mild group to the Critical group does not seem to be much. In fact, we explore this question throughout this study and provide our answer through a series of analyses and experiments: adaptive immune suppression may be the key difference between critical infection (sepsis) and common infection. This is also consistent with the analysis of clinical data (Fig. 4), which show that adaptive immunity-related genes, including B-cell-related genes Pax5, CD5, and CD19, T-cell-related gene CD8A, and MHC-II-related genes CIITA, HLA-DQ1, HLA-DR, and HLA-DO, may have the potential to differentiate patients with sepsis from those with common infections. In contrast, CD11b, a pan-marker of myeloid cells, showed no significant difference between patients with common infection and sepsis. This suggests that the number of myeloid cells alone is not sufficient to distinguish sepsis from common infections, and we must clarify again that this is not the central role we proposed.

2. Also, contrary to the author's initial criteria, CD47 increases at the RNA level but not at the protein level. There are just more cells, which does not really support the author's hypothesis.

Response:

We also have to clarify that our hypothesis is not that CD47 is increased at the protein level. CD47 is not increased at the RNA level either. In the manuscript, we stated that "Thus, the increase in overall CD47 mRNA level found in RNA-Seq is likely due to an increase in the number of cells expressing it, rather than an increase in CD47 expression in a certain cell type" (please see page 10, last paragraph). Our hypothesis does not include "an increase in CD47 at the RNA level" or "an increase in CD47 protein expression level". What we proposed is that when sepsis occurs, the MDSCs that have increased in large numbers in PBMCs all highly express CD47, which is likely the reason for the increase in the overall CD47 mRNA level in PBMCs and makes it rational to block CD47 to inhibit the immunosuppressive roles of MDSCs.

3. The manuscript heavily relies on transcriptional data, and its interpretation is overstated.

Response:

In this manuscript, only Fig. 1 to Fig. 3 are scRNA-seq and bulk RNA-seq analyses. In addition, In Fig. 2, only Fig. 2A-D are single-cell sequencing analyses, while Fig. 2E-M are wet experiments (flow cytometry analysis). Fig. 4 is a clinical data analysis, and Fig. 5, Fig. 6, Fig. 7 and Fig. 8 plus supplementary Fig. EV15-EV25 are functional experiments, mechanism analysis, and survival analysis. Therefore, this study did not rely entirely on transcriptional data, but includes a large number of wet experiments.

This study is not a pure bioinformatics analysis study. Of course, it starts with a bioinformatics analysis, because this is a data-driven study rather than a hypothesis-driven study. The data-driven paradigm can avoid pre-set conclusions or propose a hypothesis in advance, but let big data tell us which genes may play a role in sepsis. In terms of objectivity, data-driven research is less susceptible to the influence of preconceptions and subjectivity than hypothesis-driven research.

4. Generally, figure legends and text have unnecessary text but lack key information, including the time of collection of samples. There are also mislabelings, including the fact that the CFUs in WT vs KOs are not in Figures 6A and B. Many experiments do not seem to have had biological replicates, and three animals are certainly not enough to represent the measurements of the bacterial burdens.

Response:

In the legend of Figure 1, we marked the time when the blood samples were collected "Blood samples were collected 8 hours after the CLP surgery...".

In the legend of Figure 3, we also marked the time when the organ samples were collected "... major organs, including the kidney, liver and lung, were collected 8 hours after the sham or CLP surgery and analyzed by RNA-seq (3 parallel samples per group)".

Fig. 6A shows the expression profiles of SIRP α and CD11b in PBMCs (using scRNA-seq data), and Fig. 6B is a photo of bacteria cultured on a blood agar plate, in which CFU does not need to be labelled. Fig. 6C is a bar graph of the bacterial culture results of each group, where the vertical axis is the CFU value.

As for the number of animals or experiments used per group, for intervention and survival experiments, the number of mice used in each intervention is generally 6 to 10 per group, because the heterogeneity of the disease model must be taken into account. In this study, the number of mice used in the intervention experiments was $n = 16$ (Fig. 8A, B). For the detection of specific indicators, 3 samples per group ($n = 3$) were generally used. Most laboratory research published in academic journals uses $n = 3$ for flow cytometry, WB, IP and detection experiments. From the perspective of animal ethics, unless the purpose is to test the therapeutic effect of a preclinical drug, it is usually not necessary to use too many inbred animals in a single group (theoretically, these inbred mice, including C57BL/6 and Balb/c mice, are of the same genotype because they are all derived from multiple generations of brother-sister hybrids). Even if it is used sparingly, a study generally requires hundreds or even thousands of mice to complete. Therefore, most studies, including those papers published in top journals generally use $n = 2$ or $n = 3$ in detection experiments(Weber *et al*, 2015; Wei *et al*, 2024; Zhang *et al*, 2019). Since the assays performed using cell lines or inbred animals have good consistency, $n = 3$ is usually considered sufficient to reflect experimental differences.

-
5. More substantially, the authors do not really make a compelling case for the role of immunosuppression in the observations they make when CD47 is inhibited. How is B cell suppression related to the control of the bacterial burden? What does E2609 do to the bacterial burden? It is plausible that CD47 deficiency increases bacterial burden control by myeloid cells, but what is the role of B cell suppression in this?

Response:

In this study, we proposed a molecular mechanism that myeloid cells secrete amyloid- β to act on CD74 of B cells to cause B cell inhibition, thereby triggering subsequent T/B lymphocyte inhibition, which may be one of the cause of immunosuppression in sepsis. The schematic diagram of this

mechanism is shown below:

(Fig. EV26 in the revised manuscript)

As we mentioned in the Discussion section (page 22, last paragraph), CD47-expression myeloid cells are upregulated during infection, and the increase is more pronounced during sepsis. If this change is a natural response established during evolution, what is its benefit for the host in combatting infection? According to our data, A β might be the answer. Various cleaved forms of A β have been found to have antimicrobial functions. A β exerts antimicrobial activity against 8 common and clinically relevant microorganisms (Soscia *et al*, 2010), and the amyloidogenic peptides A β 1-42, A β 2-42, and A β 3p-42 are able to bind to the surfaces of various bacterial strains and *Candida albicans* and exert potent antimicrobial activities against all these pathogens (Spitzer *et al*, 2016). The common β -sheet structure shared by A β and other host defense peptides can interact and form channels on the membranes of microbes, which may be responsible for its antimicrobial function (Jang *et al*, 2011; Kagan *et al*, 2012). Therefore, CD47-SIRP α signaling may promote the release of A β from myeloid cells to defend against microbial invasion.

So, is CD47-SIRP α -promoted A β secretion beneficial or harmful during infection? The answer may not be easy to determine. Not all immune responses established during evolution benefit the host under all circumstances, such as inflammation, which can cause a variety of pathological changes, especially autoimmune syndromes and cytokine storms in patients with sepsis. We hypothesize that in the case of

mild infections, A β secretion, along with other innate immune responses, may provide the first line of defense for the host to eliminate microorganisms. On the other hand, in the case of sepsis, excessive A β causes severe B-cell suppression, resulting in adaptive immunosuppression, which is detrimental to survival in sepsis.

The following question is, if A β is an antimicrobial peptide that directly kills bacteria, why does blocking A β increase survival of septic mice, and how does B cell suppression relate to controlling of bacterial burden? According to our results, because the binding of CD47 and SIRP α transmits a "don't eat me" signal, blocking CD47 can directly increase the phagocytic activity of myeloid cells, and this process may not require the direct involvement of B cells. On the other hand, when CD47-SIRP α -A β pathway is blocked by inhibiting CD47, B cell suppression may be reversed, thereby activating adaptive immunity, promoting antigen recognition, presentation, antibody production, and ADCC effects, etc., so that innate immunity and adaptive immunity can cooperate more effectively to fight against invading pathogens.

Traditional concept generally regards the adaptive immune system, in which T/B cells play a major role, and the innate immune system, in which myeloid cells play a major role, as two separate and unrelated immune systems, and that B cells are not involved in bacterial sepsis at all. This concept needs to be updated based on recent advances. Currently, more than 600 papers on B cells and sepsis, and more than 1,600 papers on T cells and sepsis can be found on PubMed. A recent study reported that patients with PICS and sepsis had reduced naive and memory B cells and proliferated plasma cells(Sun *et al*, 2025). Our recently published study found that adaptive immune suppression caused by impaired T cells is closely associated with sepsis-related death(Yang *et al*, 2025). It has been widely reported that adaptive immune suppression caused by T/B cell suppression is strongly associated with sepsis mortality, and hundreds of papers have been published to study this phenomenon. B cells and T cells may participate in the pathogenesis of sepsis through pathogen recognition, antigen presentation, antibody production, direct antibody killing and antibody-assisted immune cell killing, immune cell assistance and immune memory. In humans, it usually takes 4–7 days for the adaptive immune system to mount a significant response, and in mice it may work faster, but it generally takes effect after the innate immune response. More studies are needed to explore the potential mechanisms by which B- and T-cell-mediated adaptive immunity participate in sepsis.

We appreciate the reviewer's suggestion to discuss how B cells and adaptive immunity are involved in bacterial sepsis. We have integrated the above content into the *Discussion* section (please see page 22, highlighted text).

6. The authors should look at the kinetics of bacterial burden in controls vs different forms of CD47 block. They should also uncouple what can be attributed to myeloid and B cells. What is the effect in survival, bacterial burden and organ damage of combining CD47KO and E2609 treatment? For example, using a mouse line deficient in CD74 would be desirable. Also, how does immunosuppression lead to organ damage? Perhaps the opposite could be expected, as less tissue damage would occur.

Response:

We thank the reviewer for the suggestion to examine the kinetics of bacterial burden in controls vs different forms of CD47 block. Blood plate culture test showed that the bacterial load in the CLP group was higher at 48h compared with 24h, indicating that the disease was still progressing. When anti-CD47 antibody intervention was given 1 h after CLP surgery, the blood bacterial burden decreased at 24h, and the decrease was greater at 48h. CD47-KO mice showed a similar phenotype (The results of bacterial burden kinetic tests were integrated into Fig. 6B and 6C).

The above result suggests that blocking the CD47 pathway does enhance the body's ability to clear invading bacteria. Given that the binding of CD47 and its receptor SIRP α can trigger a "don't eat me" signal, thereby reducing the phagocytic ability of myeloid cells, we believe that blocking CD47 can directly enhance the phagocytic effect of myeloid cells, which may not require the assistance of B cells (Fig. 6B, C and Fig. EV19, EV20).

On the other hand, blocking the CD47 pathway or using E2609 can alleviate B cell suppression by blocking the secretion of A β , thereby restoring the expression of multiple adaptive immune markers (Fig. 7 and Fig. EV23). This improves the overall immune capacity of the host, which may be beneficial for the body to coordinate innate and adaptive immunity as a whole to respond to invading pathogens. As for the role of CD74, we established the CLP model using a conventional CD74-KO mouse strain (please see Supplementary Material 1, Cd74-genotyping file). Kaplan-Meier survival

analysis showed that CD74 gene-deficient mice showed lower survival rate after CLP surgery compared with WT mice (this result was added to Fig. EV24). Because we newly set up a group of WT mice for CLP surgery as a control, and the mortality rate of each batch of CLP mice was different, this result was not integrated into Fig. 8A. This result is consistent with our hypothesis that A β needs to bind and block CD74 on B cells to lead to B cell suppression. Nevertheless, it should be noted that CD74 itself may also be involved in other important immune pathways, so more research is needed to explore the function of CD74 and the upstream and downstream of its signals.

Enhanced immune response leads to organ damage, while immunosuppression can alleviate organ damage, which comes from the concept that "excessive immune response causes sepsis". Since Lewis Thomas first proposed in the *New England Journal of Medicine* in 1972 that "it is our response that makes the disease (sepsis)..." (Thomas, 1972), numerous drugs have been developed to suppress the inflammatory mediators, but none of them has achieved the desired effect. The continued failure has led investigators to question whether organ injuries in septic patients are indeed caused by the "overactivated" immune response. In 2013, Hotchkiss published a famous review in *Nat Rev Immunol* (Hotchkiss *et al.*, 2013), suggesting that immunosuppression may play an equally important (if not more important) role in sepsis than inflammatory response. Since then, immunosuppression has become a rapidly developing field in the study of sepsis pathogenesis.

We have recently conducted a survey and completed a review, and found that more than 100,000 research papers may have been published in the field of sepsis, in which hundreds of treatments have been proposed to have dramatic therapeutic effects on sepsis models, but the only clinical applications are still antibiotics and life support treatments. Hundreds of new biomarkers have been reported for sepsis diagnosis, but currently only PCT and CRP are used clinically. The survey suggests that research papers in the field of sepsis are becoming increasingly out of touch with reality and have become a self-propagating paper system to reinforce a concept that may not be true. The gap between this "paper universe" and the clinical reality is so large that we must re-evaluate our previous concepts about sepsis.

Contrary to the concept of "our response causes sepsis", clinical trials aimed at alleviating immunosuppression and improving adaptive immune response have become a rapidly expanding area of sepsis treatment. Now there are more than 40 clinical trials for promoting immune response that

have been conducted or are underway (please see the table below), including non-specific immune-promoting therapies such as immunoglobulins (8 trials), colony-stimulating factors (17 trials), thymosin α (8 trials), IFN- γ (5 trials), and targeted therapies such as IL-7 injection (3 trials) and PD-1/PD-L1 blockade (3 trials). On the contrary, trials targeting inflammatory factors have significantly declined, with only tocilizumab (which inhibits the inflammatory factor IL-6) being tried for COVID-19 treatments (although the effectiveness remains controversial) and low-dose glucocorticoids being used in a few trials for the treatment of sepsis. Adalimumab (Humira) is an anti-TNF- α monoclonal antibody approved by the FDA in 2002 and is widely used to treat autoimmune diseases, including psoriasis, rheumatoid arthritis, ankylosing spondylitis, inflammatory bowel disease, etc. In 2017, Adalimumab's sales reached \$18 billion, making it the world's highest-selling drug. However, the most common adverse reaction to Adalimumab is infection, and the serious adverse reactions are severe infections and sepsis, including active pulmonary tuberculosis, fungal infections, etc. The theoretical expectations of blocking TNF- α are inconsistent with or even contrary to the actual clinical results, suggesting that the concept of targeting inflammatory mediators to treat sepsis may be a dead end and needs to be re-evaluated. According to the clinical trials and the latest research progress in recent years, immunomodulatory therapy to alleviate immunosuppression may be the right direction for treating sepsis.

TABLE. Clinical trials of drugs aimed at alleviating immunosuppression

Drugs	Target	Study phase	Conclusion	Sources
Immunoglobulins	IgG\IgM\IgA	Phase II, Phase III	Immunoglobulin infusion is generally safe, but conclusive evidence to show that immunoglobulin has a significant therapeutic effect on sepsis is lacking.	1980(Jones et al , 1980), 2017(Iizuka et al , 2017), 2006(Hentrich et al , 2006), 2021(Biagioni et al , 2021), 2019(Gharebaghi et al , 2020), 2021(Tabarsi et al , 2021), 2022(Group, 2022), 2023(Alemanly et al , 2023)
G/GM-CSF	Neutrophils, Monocyte-Macrophages	Phase I/II, Phase III	G-CSF and GM-CSF have been confirmed to increase the number and function of leukocytes and reduce the mortality of some neonatal sepsis patients, but more evidence is needed to determine whether these agents can reduce the mortality of adult sepsis patients.	1994(Gillan et al , 1994), 1995(Cairo et al , 1995), 1998(Drossou-Agakidou et al , 1998), 2012(El-Ganzoury et al , 2012), 1999(Carr et al , 1999), 2001(Bilgin et al , 2001), 2009(Carr et al , 2009), 2013(Marlow et al , 2013), 2009(Meisel et al , 2009), 1998(Schibler et al , 1998), 2001(Miura et al , 2001), 2013(Borjanyazdi et al , 2013), 2012(Gathwala et al , 2012), 2012(Chaudhuri et al , 2012),

				2007(Cheng et al , 2007), 2002(Presneill et al , 2002), 2021(Cheng et al , 2021)
Thymosin α 1	Thymosin α 1	Phase III	There is controversy about the therapeutic effect of thymosin α 1; two clinical trials reported it to be effective, but another six reported it to be ineffective. It might have beneficial effects in patients aged 60 and older and those with chronic conditions.	2013(Wu et al , 2013), 2020(Wu et al , 2020), 2022(Shetty et al , 2022), 2021(Sun et al , 2021), 2021(Huang et al , 2021), 2021(Wang et al , 2021), 2021(Liu et al , 2021), 2025(Wu et al , 2025)
IFN- γ	IFN- γ	Phase II, Phase III	The ability of IFN- γ to improve immune status and reduce secondary infections in patients with sepsis needs to be further evaluated.	2012(Leentjens et al , 2012), 2013(Delsing et al , 2014), 2019(Payen et al , 2019), NCT03332225 (Ongoing), NCT01649921(Not reported)
IL-7	IL-7	Phase II	IL-7 therapy is helpful in the recovery of lymphatic counts in patients with sepsis.	2018(Francois et al , 2018), 2023(Daix et al , 2023), 2020(Laterre et al , 2020)
BMS-936559	PD-L1	Phase Ib	BMS-936559 was well tolerated but no conclusion can be drawn about whether BMS-936559 affects survival rate of sepsis.	NCT02576457 (Terminated)
Nivolumab	PD-1	Phase I/II	Nivolumab has shown safety in patients with sepsis, but the effect of reducing mortality has not been observed.	2020(Watanabe et al , 2020), 2019(Hotchkiss et al , 2019)

References:

- Aleman A, Millat-Martinez P, Corbacho-Monne M, Suner C, Galvan-Casas C, Carrera C, Ouchi D, Prat N, Ara J, Nadal N *et al* (2023) Subcutaneous anti-COVID-19 hyperimmune immunoglobulin for prevention of disease in asymptomatic individuals with SARS-CoV-2 infection: a double-blind, placebo-controlled, randomised clinical trial. *EClinicalMedicine* 57: 101898
- Aloisio E, Dolci A, Panteghini M (2019) Procalcitonin: Between evidence and critical issues. *Clin Chim Acta* 496: 7-12
- Bah I, Kumbhare A, Nguyen L, McCall CE, El Gazzar M (2018) IL-10 induces an immune repressor pathway in sepsis by promoting S100A9 nuclear localization and MDSC development. *Cell Immunol* 332: 32-38
- Biagioni E, Tosi M, Berlot G, Castiglione G, Corona A, De Cristofaro MG, Donati A, Feltracco P, Forfori F, Fraganza F *et al* (2021) Adjunctive IgM-enriched immunoglobulin therapy with a personalised dose based on serum IgM-titres versus standard dose in the treatment of septic shock: a randomised controlled trial (IgM-fat trial). *BMJ Open* 11: e036616
- Bilgin K, Yaramis A, Haspolat K, Tas MA, Gunbey S, Derman O (2001) A randomized trial of granulocyte-macrophage colony-stimulating factor in neonates with sepsis and neutropenia. *Pediatrics* 107: 36-41
- Borjanyazdi L, Froomandi M, Noori Shadkam M, Hashemi A, Fallah R (2013) The effect of granulocyte colony stimulating factor administration on preterm infant with neutropenia and clinical sepsis: a randomized clinical trial. *Iran J Ped Hematol Oncol* 3: 64-68
- Byrnes D, Masterson CH, Brady J, Alagesan S, Gonzalez HE, McCarthy SD, Fandino J, O'Toole DP, Laffey JG (2023) Differential Effects of Cytokine Versus Hypoxic Preconditioning of Human Mesenchymal Stromal Cells in Pulmonary Sepsis Induced by Antimicrobial-Resistant *Klebsiella pneumoniae*. *Pharmaceuticals (Basel)* 16
- Cairo MS, Christensen R, Sender LS, Ellis R, Rosenthal J, van de Ven C, Worcester C, Agosti JM (1995) Results of a phase I/II trial of recombinant human granulocyte-macrophage colony-stimulating factor in very low birthweight neonates: significant induction of circulatory neutrophils, monocytes, platelets, and bone marrow neutrophils. *Blood* 86: 2509-2515
- Carr R, Brocklehurst P, Dore CJ, Modi N (2009) Granulocyte-macrophage colony stimulating factor administered as prophylaxis for reduction of sepsis in extremely preterm, small for gestational age neonates (the PROGRAMS trial): a single-blind, multicentre, randomised controlled trial. *Lancet* 373: 226-233
- Carr R, Modi N, Dore CJ, El-Rifai R, Lindo D (1999) A randomized, controlled trial of prophylactic granulocyte-macrophage colony-stimulating factor in human newborns less than 32 weeks gestation. *Pediatrics* 103: 796-802
- Chaudhuri J, Mitra S, Mukhopadhyay D, Chakraborty S, Chatterjee S (2012) Granulocyte Colony-stimulating Factor for Preterms with Sepsis and Neutropenia: A Randomized Controlled Trial. *J Clin Neonatol* 1: 202-206
- Chen J, Cai S, Li R, Xie J, Yang F, Liu T (2022) Blockade of Cyclooxygenase-2 ameliorates sepsis induced immune-suppression by regulating myeloid-derived suppressor cells. *Int Immunopharmacol* 104: 108506

Cheng AC, Limmathurotsakul D, Chierakul W, Getchalarat N, Wuthiekanun V, Stephens DP, Day NP, White NJ, Chaowagul W, Currie BJ *et al* (2007) A randomized controlled trial of granulocyte colony-stimulating factor for the treatment of severe sepsis due to melioidosis in Thailand. *Clin Infect Dis* 45: 308-314

Cheng LL, Guan WJ, Duan CY, Zhang NF, Lei CL, Hu Y, Chen AL, Li SY, Zhuo C, Deng XL *et al* (2021) Effect of Recombinant Human Granulocyte Colony-Stimulating Factor for Patients With Coronavirus Disease 2019 (COVID-19) and Lymphopenia: A Randomized Clinical Trial. *JAMA Intern Med* 181: 71-78

Daix T, Mathonnet A, Brakenridge S, Dequin PF, Mira JP, Berbille F, Morre M, Jeannet R, Blood T, Unsinger J *et al* (2023) Intravenously administered interleukin-7 to reverse lymphopenia in patients with septic shock: a double-blind, randomized, placebo-controlled trial. *Ann Intensive Care* 13: 17

Delsing CE, Gresnigt MS, Leentjens J, Preijers F, Frager FA, Kox M, Monneret G, Venet F, Bleeker-Rovers CP, van de Veerdonk FL *et al* (2014) Interferon-gamma as adjunctive immunotherapy for invasive fungal infections: a case series. *BMC Infect Dis* 14: 166

Deng W, Zhu S, Zeng L, Liu J, Kang R, Yang M, Cao L, Wang H, Billiar TR, Jiang J *et al* (2018) The Circadian Clock Controls Immune Checkpoint Pathway in Sepsis. *Cell Rep* 24: 366-378

Derive M, Bouazza Y, Alauzet C, Gibot S (2012) Myeloid-derived suppressor cells control microbial sepsis. *Intensive Care Med* 38: 1040-1049

Drossou-Agakidou V, Kanakoudi-Tsakalidou F, Sarafidis K, Taparkou A, Tzimouli V, Tsandali H, Kremenopoulos G (1998) Administration of recombinant human granulocyte-colony stimulating factor to septic neonates induces neutrophilia and enhances the neutrophil respiratory burst and beta2 integrin expression. Results of a randomized controlled trial. *Eur J Pediatr* 157: 583-588

El-Ganzoury MM, El-Farrash RA, Saad AA, Mohamed AG, El-Sherbini IG (2012) In vivo effect of recombinant human granulocyte colony-stimulating factor on neutrophilic expression of CD11b in septic neonates: a randomized controlled trial. *Pediatr Hematol Oncol* 29: 272-284

Feng Z, Li M, Ma A, Wei Y, Huang L, Kong L, Kang Y, Wang Z, Xiao F, Zhang W (2023) Intermedin (adrenomedullin 2) plays a protective role in sepsis by regulating T- and B-cell proliferation and activity. *Int Immunopharmacol* 121: 110488

Francois B, Jeannet R, Daix T, Walton AH, Shotwell MS, Unsinger J, Monneret G, Rimmele T, Blood T, Morre M *et al* (2018) Interleukin-7 restores lymphocytes in septic shock: the IRIS-7 randomized clinical trial. *JCI Insight* 3

Gathwala G, Walia M, Bala H, Singh S (2012) Recombinant human granulocyte colony-stimulating factor in preterm neonates with sepsis and relative neutropenia: a randomized, single-blind, non-placebo-controlled trial. *J Trop Pediatr* 58: 12-18

Gharebaghi N, Nejadrahim R, Mousavi SJ, Sadat-Ebrahimi SR, Hajizadeh R (2020) The use of intravenous immunoglobulin gamma for the treatment of severe coronavirus disease 2019: a randomized placebo-controlled double-blind clinical trial. *BMC Infect Dis* 20: 786

Gillan ER, Christensen RD, Suen Y, Ellis R, van de Ven C, Cairo MS (1994) A randomized, placebo-controlled trial of recombinant human granulocyte colony-stimulating factor administration in newborn infants with presumed sepsis: significant induction of peripheral and bone marrow neutrophilia. *Blood* 84: 1427-1433

Group IS (2022) Hyperimmune immunoglobulin for hospitalised patients with COVID-19 (ITAC): a double-blind, placebo-controlled, phase 3, randomised trial. *Lancet* 399: 530-540

Hentrich M, Fehnle K, Ostermann H, Kienast J, Cornely O, Salat C, Ubelacker R, Buchheidt D, Behre

G, Hiddemann W *et al* (2006) IgMA-enriched immunoglobulin in neutropenic patients with sepsis syndrome and septic shock: a randomized, controlled, multiple-center trial. *Critical care medicine* 34: 1319-1325

Hotchkiss RS, Colston E, Yende S, Crouser ED, Martin GS, Albertson T, Bartz RR, Brakenridge SC, Delano MJ, Park PK *et al* (2019) Immune checkpoint inhibition in sepsis: a Phase 1b randomized study to evaluate the safety, tolerability, pharmacokinetics, and pharmacodynamics of nivolumab. *Intensive Care Med* 45: 1360-1371

Hotchkiss RS, Monneret G, Payen D (2013) Sepsis-induced immunosuppression: from cellular dysfunctions to immunotherapy. *Nat Rev Immunol* 13: 862-874

Huang C, Fei L, Xu W, Li W, Xie X, Li Q, Chen L (2021) Efficacy Evaluation of Thymosin Alpha 1 in Non-severe Patients With COVID-19: A Retrospective Cohort Study Based on Propensity Score Matching. *Front Med (Lausanne)* 8: 664776

Iizuka Y, Sanui M, Sasabuchi Y, Lefor AK, Hayakawa M, Saito S, Uchino S, Yamakawa K, Kudo D, Takimoto K *et al* (2017) Low-dose immunoglobulin G is not associated with mortality in patients with sepsis and septic shock. *Critical care* 21: 181

Jang H, Arce FT, Mustata M, Ramachandran S, Capone R, Nussinov R, Lal R (2011) Antimicrobial protegrin-1 forms amyloid-like fibrils with rapid kinetics suggesting a functional link. *Biophys J* 100: 1775-1783

Jones RJ, Roe EA, Gupta JL (1980) Controlled trial of Pseudomonas immunoglobulin and vaccine in burn patients. *Lancet* 2: 1263-1265

Joshi A, Pancari G, Cope L, Bowman EP, Cua D, Proctor RA, McNeely T (2012) Immunization with Staphylococcus aureus iron regulated surface determinant B (IsdB) confers protection via Th17/IL17 pathway in a murine sepsis model. *Hum Vaccin Immunother* 8: 336-346

Joshi PR, Adhikari S, Onah C, Carrier C, Judd A, Mack M, Baral P (2024) Lung-innervating nociceptor sensory neurons promote pneumonic sepsis during carbapenem-resistant Klebsiella pneumoniae lung infection. *Sci Adv* 10: eadl6162

Kadowaki T, Morishita A, Niki T, Hara J, Sato M, Tani J, Miyoshi H, Yoneyama H, Masaki T, Hattori T *et al* (2013) Galectin-9 prolongs the survival of septic mice by expanding Tim-3-expressing natural killer T cells and PDCA-1+ CD11c+ macrophages. *Critical care* 17: R284

Kagan BL, Jang H, Capone R, Teran Arce F, Ramachandran S, Lal R, Nussinov R (2012) Antimicrobial properties of amyloid peptides. *Mol Pharm* 9: 708-717

Kelly-Scumpia KM, Scumpia PO, Weinstein JS, Delano MJ, Cuenca AG, Nacionales DC, Wynn JL, Lee PY, Kumagai Y, Efron PA *et al* (2011) B cells enhance early innate immune responses during bacterial sepsis. *J Exp Med* 208: 1673-1682

Lai D, Qin C, Shu Q (2014) Myeloid-derived suppressor cells in sepsis. *Biomed Res Int* 2014: 598654

Laterre PF, Francois B, Collienne C, Hantson P, Jeannet R, Remy KE, Hotchkiss RS (2020) Association of Interleukin 7 Immunotherapy With Lymphocyte Counts Among Patients With Severe Coronavirus Disease 2019 (COVID-19). *JAMA Netw Open* 3: e2016485

Lederer JA, Rodrick ML, Mannick JA (1999) The effects of injury on the adaptive immune response. *Shock* 11: 153-159

Leentjens J, Kox M, Koch RM, Preijers F, Joosten LA, van der Hoeven JG, Netea MG, Pickkers P (2012) Reversal of immunoparalysis in humans in vivo: a double-blind, placebo-controlled, randomized pilot study. *Am J Respir Crit Care Med* 186: 838-845

Liang H, Shen X (2020) LXR activation radiosensitizes non-small cell lung cancer by restricting

myeloid-derived suppressor cells. *Biochemical and biophysical research communications* 528: 330-335

Lin RY, Astiz ME, Saxon JC, Saha DC, Rackow EC (1994) Relationships between plasma cytokine concentrations and leukocyte functional antigen expression in patients with sepsis. *Critical care medicine* 22: 1595-1602

Liu J, Shen Y, Wen Z, Xu Q, Wu Z, Feng H, Li Z, Dong X, Huang S, Guo J *et al* (2021) Efficacy of Thymosin Alpha 1 in the Treatment of COVID-19: A Multicenter Cohort Study. *Front Immunol* 12: 673693

London NR, Zhu W, Bozza FA, Smith MC, Greif DM, Sorensen LK, Chen L, Kaminoh Y, Chan AC, Passi SF *et al* (2010) Targeting Robo4-dependent Slit signaling to survive the cytokine storm in sepsis and influenza. *Sci Transl Med* 2: 23ra19

Luna CM, Sibila O, Agusti C, Torres A (2009) Animal models of ventilator-associated pneumonia. *Eur Respir J* 33: 182-188

Marlow N, Morris T, Brocklehurst P, Carr R, Cowan FM, Patel N, Petrou S, Redshaw ME, Modi N, Dore C (2013) A randomised trial of granulocyte-macrophage colony-stimulating factor for neonatal sepsis: outcomes at 2 years. *Arch Dis Child Fetal Neonatal Ed* 98: F46-53

Meisel C, Schefold JC, Pschowski R, Baumann T, Hetzger K, Gregor J, Weber-Carstens S, Hasper D, Keh D, Zuckermann H *et al* (2009) Granulocyte-macrophage colony-stimulating factor to reverse sepsis-associated immunosuppression: a double-blind, randomized, placebo-controlled multicenter trial. *Am J Respir Crit Care Med* 180: 640-648

Miura E, Procianoy RS, Bittar C, Miura CS, Miura MS, Mello C, Christensen RD (2001) A randomized, double-masked, placebo-controlled trial of recombinant granulocyte colony-stimulating factor administration to preterm infants with the clinical diagnosis of early-onset sepsis. *Pediatrics* 107: 30-35

Molano-Franco D, Arevalo-Rodriguez I, Muriel A, Del Campo-Albendea L, Fernandez-Garcia S, Alvarez-Mendez A, Simancas-Racines D, Viteri A, Sanchez G, Fernandez-Felix B *et al* (2023) Basal procalcitonin, C-reactive protein, interleukin-6, and presepsin for prediction of mortality in critically ill septic patients: a systematic review and meta-analysis. *Diagn Progn Res* 7: 15

Natanson C, Hoffman WD, Suffredini AF, Eichacker PQ, Danner RL (1994) Selected treatment strategies for septic shock based on proposed mechanisms of pathogenesis. *Ann Intern Med* 120: 771-783

Nedeva C, Menassa J, Puthalakath H (2019) Sepsis: Inflammation Is a Necessary Evil. *Front Cell Dev Biol* 7: 108

Oberholzer A, Oberholzer C, Moldawer LL (2001) Sepsis syndromes: understanding the role of innate and acquired immunity. *Shock* 16: 83-96

Osuchowski MF, Ayala A, Bahrami S, Bauer M, Boros M, Cavaillon JM, Chaudry IH, Coopersmith CM, Deutschman C, Drechsler S *et al* (2018) Minimum quality threshold in pre-clinical sepsis studies (MQTiPSS): an international expert consensus initiative for improvement of animal modeling in sepsis. *Intensive Care Med Exp* 6: 26

Payen D, Faivre V, Miatello J, Leentjens J, Brumpt C, Tissieres P, Dupuis C, Pickkers P, Lukaszewicz AC (2019) Multicentric experience with interferon gamma therapy in sepsis induced immunosuppression. A case series. *BMC Infect Dis* 19: 931

Presneill JJ, Harris T, Stewart AG, Cade JF, Wilson JW (2002) A randomized phase II trial of granulocyte-macrophage colony-stimulating factor therapy in severe sepsis with respiratory dysfunction. *Am J Respir Crit Care Med* 166: 138-143

Reim D, Westenfelder K, Kaiser-Moore S, Schlautkotter S, Holzmann B, Weighardt H (2009) Role of T

cells for cytokine production and outcome in a model of acute septic peritonitis. *Shock* 31: 245-250

Riedemann NC, Guo RF, Ward PA (2003) The enigma of sepsis. *J Clin Invest* 112: 460-467

Ruan WS, Feng MX, Xu J, Xu YG, Song CY, Lin LY, Li L, Lu YQ (2020) Early Activation of Myeloid-Derived Suppressor Cells Participate in Sepsis-Induced Immune Suppression via PD-L1/PD-1 Axis. *Front Immunol* 11: 1299

Ryoo SM, Han KS, Ahn S, Shin TG, Hwang SY, Chung SP, Hwang YJ, Park YS, Jo YH, Chang HL *et al* (2019) The usefulness of C-reactive protein and procalcitonin to predict prognosis in septic shock patients: A multicenter prospective registry-based observational study. *Sci Rep* 9: 6579

Sander LE, Sackett SD, Dierssen U, Beraza N, Linke RP, Muller M, Blander JM, Tacke F, Trautwein C (2010) Hepatic acute-phase proteins control innate immune responses during infection by promoting myeloid-derived suppressor cell function. *J Exp Med* 207: 1453-1464

Schibler KR, Osborne KA, Leung LY, Le TV, Baker SI, Thompson DD (1998) A randomized, placebo-controlled trial of granulocyte colony-stimulating factor administration to newborn infants with neutropenia and clinical signs of early-onset sepsis. *Pediatrics* 102: 6-13

Secher T, Dalonneau E, Ferreira M, Parent C, Azzopardi N, Paintaud G, Si-Tahar M, Heuze-Vourc'h N (2019) In a murine model of acute lung infection, airway administration of a therapeutic antibody confers greater protection than parenteral administration. *J Control Release* 303: 24-33

Shelley O, Murphy T, Paterson H, Mannick JA, Lederer JA (2003) Interaction between the innate and adaptive immune systems is required to survive sepsis and control inflammation after injury. *Shock* 20: 123-129

Shetty A, Chandrakant NS, Darnule RA, Manjunath BG, Sathe P (2022) A Double-blind Multicenter Two-arm Randomized Placebo-controlled Phase-III Clinical Study to Evaluate the Effectiveness and Safety of Thymosin alpha1 as an Add-on Treatment to Existing Standard of Care Treatment in Moderate-to-severe COVID-19 Patients. *Indian J Crit Care Med* 26: 913-919

Soscia SJ, Kirby JE, Washicosky KJ, Tucker SM, Ingelsson M, Hyman B, Burton MA, Goldstein LE, Duong S, Tanzi RE *et al* (2010) The Alzheimer's disease-associated amyloid beta-protein is an antimicrobial peptide. *PLoS one* 5: e9505

Spitzer P, Condic M, Herrmann M, Oberstein TJ, Scharin-Mehlmann M, Gilbert DF, Friedrich O, Gromer T, Kornhuber J, Lang R *et al* (2016) Amyloidogenic amyloid-beta-peptide variants induce microbial agglutination and exert antimicrobial activity. *Sci Rep* 6: 32228

Sun Q, Xie J, Zheng R, Li X, Chen H, Tong Z, Du B, Qiu H, Yang Y (2021) The effect of thymosin alpha1 on mortality of critical COVID-19 patients: A multicenter retrospective study. *Int Immunopharmacol* 90: 107143

Sun XF, Luo WC, Huang SQ, Zheng YJ, Xiao L, Zhang ZW, Liu RH, Zhong ZW, Song JQ, Nan K *et al* (2025) Immune-cell signatures of persistent inflammation, immunosuppression, and catabolism syndrome after sepsis. *Med*

Tabarsi P, Barati S, Jamaati H, Haseli S, Marjani M, Moniri A, Abtahian Z, Dastan A, Yousefian S, Eskandari R *et al* (2021) Evaluating the effects of Intravenous Immunoglobulin (IVIg) on the management of severe COVID-19 cases: A randomized controlled trial. *Int Immunopharmacol* 90: 107205

Tang BM, Eslick GD, Craig JC, McLean AS (2007) Accuracy of procalcitonin for sepsis diagnosis in critically ill patients: systematic review and meta-analysis. *Lancet Infect Dis* 7: 210-217

Thomas L (1972) Germs. *The New England journal of medicine* 287: 553-555

Wang Z, Chen J, Zhu C, Liu L, Qi T, Shen Y, Zhang Y, Xu L, Li T, Qian Z *et al* (2021) Thymosin

Alpha-1 Has no Beneficial Effect on Restoring CD4+ and CD8+ T Lymphocyte Counts in COVID-19 Patients. *Front Immunol* 12: 568789

Watanabe E, Nishida O, Kakihana Y, Odani M, Okamura T, Harada T, Oda S (2020) Pharmacokinetics, Pharmacodynamics, and Safety of Nivolumab in Patients With Sepsis-Induced Immunosuppression: A Multicenter, Open-Label Phase 1/2 Study. *Shock* 53: 686-694

Weber GF, Chousterman BG, He S, Fenn AM, Nairz M, Anzai A, Brenner T, Uhle F, Iwamoto Y, Robbins CS *et al* (2015) Interleukin-3 amplifies acute inflammation and is a potential therapeutic target in sepsis. *Science* 347: 1260-1265

Wei Y, Li R, Meng Y, Hu T, Zhao J, Gao Y, Bai Q, Li N, Zhao Y (2024) Transport mechanism and pharmacology of the human GlyT1. *Cell* 187: 1719-1732 e1714

Weighardt H, Heidecke CD, Emmanuilidis K, Maier S, Bartels H, Siewert JR, Holzmann B (2000) Sepsis after major visceral surgery is associated with sustained and interferon-gamma-resistant defects of monocyte cytokine production. *Surgery* 127: 309-315

Weis S, Carlos AR, Moita MR, Singh S, Blankenhaus B, Cardoso S, Larsen R, Rebelo S, Schauble S, Del Barrio L *et al* (2017) Metabolic Adaptation Establishes Disease Tolerance to Sepsis. *Cell* 169: 1263-1275 e1214

Wu J, Pei F, Zhou L, Li W, Sun R, Li Y, Wang Z, He Z, Zhang X, Jin X *et al* (2025) The efficacy and safety of thymosin alpha1 for sepsis (TESTS): multicentre, double blinded, randomised, placebo controlled, phase 3 trial. *BMJ* 388: e082583

Wu J, Zhou L, Liu J, Ma G, Kou Q, He Z, Chen J, Ou-Yang B, Chen M, Li Y *et al* (2013) The efficacy of thymosin alpha 1 for severe sepsis (ETASS): a multicenter, single-blind, randomized and controlled trial. *Crit Care* 17: R8

Wu M, Ji JJ, Zhong L, Shao ZY, Xie QF, Liu ZY, Wang CL, Su L, Feng YW, Liu ZF *et al* (2020) Thymosin alpha1 therapy in critically ill patients with COVID-19: A multicenter retrospective cohort study. *Int Immunopharmacol* 88: 106873

Xiao F, Wang D, Kong L, Li M, Feng Z, Shuai B, Wang L, Wei Y, Li H, Wu S *et al* (2018) Intermedin protects against sepsis by concurrently re-establishing the endothelial barrier and alleviating inflammatory responses. *Nature communications* 9: 2644

Yang J, Ma C, Feng Z, Xiao F, Kang Y, Zhang W, Liao X (2025) Soluble CD72 concurrently impairs T cell functions while enhances inflammatory response in sepsis. *Int Immunopharmacol* 147: 113981

Zhang D, Tang Z, Huang H, Zhou G, Cui C, Weng Y, Liu W, Kim S, Lee S, Perez-Neut M *et al* (2019) Metabolic regulation of gene expression by histone lactylation. *Nature* 574: 575-580

Zhang W, Luo M, Zhou Y, Hu J, Li C, Liu K, Liu M, Zhu Y, Chen H, Zhang H (2021) Liver X receptor agonist GW3965 protects against sepsis by promoting myeloid derived suppressor cells apoptosis in mice. *Life sciences* 276: 119434

Zhao D, Tang M, Ma Z, Hu P, Fu Q, Yao Z, Zhou C, Zhou T, Cao J (2024) Synergy of bacteriophage depolymerase with host immunity rescues sepsis mice infected with hypervirulent *Klebsiella pneumoniae* of capsule type K2. *Virulence* 15: 2415945

Dear Prof. Zhang,

Thank you for the submission of your revised manuscript to EMBO reports. I have now received the reports from the three referees that were asked to re-evaluate your study, which can be found at the end of this email.

As you will see, referee #1 is satisfied by the revision and supports publication. Referee #3 remains critical, but does not indicate specifically which concerns have not been addressed. Referee #2 thinks that the findings are of interest, but has remaining concerns. During cross-commenting, referee #1 agreed with the remaining points of referee #2 but nevertheless indicated that the study should be published after further revisions.

Given the constructive comments of referees #1 and #2, I would like to invite you to revise your manuscript further with the understanding that the remaining concerns of referee #2 must be addressed in the revised manuscript and in a detailed point-by-point response.

Moreover, and very importantly, we request that the human data (patient information) provided is anonymized!

When submitting your further revised manuscript, we will require:

1) a .docx formatted version of the final manuscript text (including legends for main figures, EV figures and tables), but without the figures included. Figure legends should be compiled at the end of the manuscript text.

2) individual production quality figure files as .eps, .tif, .jpg (one file per figure), of main figures and EV figures. Please upload these as separate, individual files upon re-submission.

3) a .docx formatted letter INCLUDING the reviewers' reports on the revision and your detailed point-by-point responses to their remaining concerns. As part of the EMBO Press transparent editorial process, the point-by-point response is part of the Review Process File (RPF), which will be published alongside your paper.

4) a complete author checklist, which you can download from our author guidelines

(<https://www.embopress.org/page/journal/14693178/authorguide>). Please insert page numbers in the checklist to indicate where the requested information can be found in the manuscript. The completed author checklist will also be part of the RPF.

5) that primary datasets produced in this study (e.g. RNA-seq, ChIP-seq, structural and array data) are deposited in an appropriate public database. If no primary datasets have been deposited, please also state this in a dedicated section (e.g. 'No primary datasets have been generated and deposited'), see below.

The accession numbers and database should be listed in a formal "Data Availability" section that follows the model below. This is now mandatory (like the COI statement). Please note that the Data Availability Section is restricted to new primary data that are part of this study. This section is mandatory. As indicated above, if no primary datasets have been deposited, please state this in this section

Data availability

8) Regarding data quantification and statistics, please make sure that the number "n" for how many independent experiments were performed, their nature (biological versus technical replicates), the bars and error bars (e.g. SEM, SD) and the test used to calculate p-values is indicated in the respective figure legends (also for EV and Appendix figures). Please also check that all the p-values are explained in the legend, and that these fit to those shown in the figure. Please provide statistical testing where applicable. Please avoid the phrase 'independent experiment', but clearly state if these were biological or technical replicates. Please also indicate (e.g. with n.s.) if testing was performed, but the differences are not significant. In case n=2, please show the data as separate datapoints without error bars and statistics. See also: <http://www.embopress.org/page/journal/14693178/authorguide#statisticalanalysis>

9) Please add scale bars of similar style and thickness to microscopic images, using clearly visible black or white bars (depending on the background). Please place these in the lower right corner of the images themselves. Please do not write on or near the bars in the image but define the size in the respective figure legend.

10) Please also note our reference format:

12) We now use CRediT to specify the contributions of each author in the journal submission system. CRediT replaces the author contribution section. Please use the free text box to provide more detailed descriptions and do NOT provide your final manuscript text file with an author contributions section. See also our guide to authors: <https://www.embopress.org/page/journal/14693178/authorguide#authorshipguidelines>

13) All Materials and Methods need to be described in the main text using our 'Structured Methods' format, which is required for all research articles. According to this format, the Methods section should include a Reagents and Tools Table (listing key reagents, experimental models, software, and relevant equipment and including their sources and relevant identifiers), uploaded

as separate file, and a Methods section in which we encourage the authors to describe their methods using a step-by-step protocol format with bullet points, to facilitate the adoption of the methodologies across labs. More information on how to adhere to this format as well as downloadable templates (.doc) for the Reagents and Tools Table can be found in our author guidelines (section 'Structured Methods'):

14) Please order the sections like this, using these names:

Title page - Abstract - Keywords - Introduction - Results - Discussion - Methods - Data availability section - Acknowledgements (including the funding information) - Disclosure and Competing Interests Statement - References - Figure legends - Expanded View Figure legends

15) Please make sure that all the funding information is also entered into the online submission system and that it is complete and similar to the one in the acknowledgement section of the manuscript text file.

Finally, please note that all corresponding authors are required to supply an institutional e-mail address and an ORCID ID for their name upon submission of the revised manuscript. Please do that for your co-corresponding authors.

Please find instructions on how to link the ORCID ID to the account in our manuscript tracking system in our Author guidelines: <http://www.embopress.org/page/journal/14693178/authorguide#authorshipguidelines>

I look forward to seeing a revised form of your manuscript when it is ready.

Yours sincerely,

Referee #1:

The authors have taken my comments serious and used them to make a better paper.

Referee #2:

I thank the authors for their careful consideration of the points I raised in my first review. I am content that my comments # 1, 3 and 8 have been suitably addressed.

With respect to #4 (K. pneumoniae model) I think the authors have slightly misinterpreted my comments, for which I apologise if I was not clear. I completely agree with them that a live-infection model is much superior to an LPS model, which indeed is not really suitable, and I also agree that K.pneumoniae is a very appropriate organism to study. My concern was that injecting a very large number of bacteria directly into the trachea may indeed not really be a model of pneumonia, but in effect a direct LPS challenge model. The authors have addressed this to some extent by providing evidence of different lethality outcomes, although they compared intraperitoneal infection of LPS, not intratracheal. A more specific issue is whether the effects they have seen in gene expression profiles differ in LPS challenge and their infection model. However, I accept that this is probably a step too far for this manuscript.

At #2 I pointed out that it would be interesting to examine bacteriological studies from other organs. The authors have done that and (unsurprisingly) found that indeed there was evidence of systemic infection. They originally pointed to the fact that immune

cells infiltrated the lungs but not the other organs, concluding that lungs alone were likely a focus of infection. In fact, their new data contradicts this conclusion but they do not seem to have acknowledged this or provided any potential explanation.

The clinical data (#5) remain something of a concern. Underlying this component of the work is that they can confidently separate septic patients from those with "mild" infection and from normal or trauma controls. However, careful examination of the Supplementary Table reveals, for example, that there are many cells with missing data, the use of "ok" in the PaO₂/FiO₂ column (what does that mean?), examples where the PaO₂/FiO₂ ratio is at a level which would be considered septic but the patient is not so designated (e.g. patient 93), incorrect entries (patient 69), inconsistent PCT data, inadequate bacteriological data etc. The fact that patients were not sequential is relevant because it implies that they were "chosen" in some way, which reduces the significance of the findings. The Table has units but no normal values/ranges for the various criteria. I am not wholly confident that these data allow the subsequent conclusions regarding the use of cell markers to identify septic patients, which in my view would have been better done as a separate, properly designed study. One of the reasons that I suggested that it would be informative to include some "disease controls" (i.e. trauma patients) was in order to compare their gene expression profiles with those of the infected patients. Very interestingly, the authors new data do indeed show that at least in some cases the profiles were the same in trauma and infected patients. While they have provided the data in Figure 4 they have not commented on it at all, a particularly significant absence given that this was one of the cornerstones of their paper.

The authors have acknowledged (#6) that it is slightly odd that IL-10 levels were not raised. I am not completely persuaded by their explanation, which in any case does not appear in the Discussion. I think that the recent paper I identified (#7) should have been cited in the revised paper.

Referee #3:

The authors have only partially addressed my concerns, having argued more than implemented the suggestions for improvement of the manuscript. In addition, the lack of mechanist substance and coherence remain.

Point-by-point response:

Referee #2:

I thank the authors for their careful consideration of the points I raised in my first review. I am content that my comments # 1, 3 and 8 have been suitably addressed. With respect to #4 (K. pneumoniae model) I think the authors have slightly misinterpreted my comments, for which I apologise if I was not clear. I completely agree with them that a live-infection model is much superior to an LPS model, which indeed is not really suitable, and I also agree that K.pneumoniae is a very appropriate organism to study. My concern was that injecting a very large number of bacteria directly into the trachea may indeed not really be a model of pneumonia, but in effect a direct LPS challenge model. The authors have addressed this to some extent by providing evidence of different lethality outcomes, although they compared intraperitoneal infection of LPS, not intratracheal. A more specific issue is whether the effects they have seen in gene expression profiles differ in LPS challenge and their infection model. However, I accept that this is probably a step too far for this manuscript.

Response:

We are very grateful for the valuable comments and suggestions from the reviewer. We agree with the reviewer that a single infusion of a large dose of bacteria into the lungs may be different from the actual infection that occurs in the clinic. The reason for choosing this dose of 10^9 CFU is that lower doses ($\leq 10^8$ CFU) do not cause death in mice, and therefore cannot create a successful model of pneumonia-induced sepsis death.

In clinical practice, active pneumonia caused by bacterial infection, such as *Klebsiella p.*, often occurs in patients with long-term hospitalization. However, it is difficult to induce active pneumonia or sepsis with a small dose of bacteria using normal mice. On the other hand, if immunodeficient mice such as nude mice or SCID mice are used, it is inconsistent with clinical practice because sepsis patients generally do not have congenital T/B cell immunodeficiency. Therefore, using a large dose of bacteria to induce pneumonia sepsis is an alternative before an ideal animal model appears. In future studies, we plan to immobilize bacteria in a clot to enable their gradual release over time, thereby avoiding the shock of a single large bacterial infusion. Similar studies have been conducted in the past. For example, in 2000, Mathiak reported a sepsis model induced by intraperitoneal implantation of

fibrin clots containing *E. coli* ($5-7 \times 10^8$ CFU/mL) in rats (Mathiak *et al*, 2000). We will refer to similar models and evaluate their advantages and disadvantages.

As for the effects of high-dose LPS injection and direct bacterial infection on the host, we speculate that the gene expression profiles induced by these two challenges are likely to be very different. In this regard, we plan to initiate a study on animal models of sepsis in the near future. This will include conventional LPS and CLP models, as well as direct pathogen challenge models such as *Klebsiella pneumoniae*, *Acinetobacter baumannii*, fungi (*Candida albicans*, *etc.*), and viruses (*Influenza A*, *etc.*). We will analyze differences in mortality rates, patterns of death, organ injury patterns, sites of pathogen invasion, and gene expression profiles across different types of sepsis models, and summarize their advantages and disadvantages and applicable scenarios. Such a comprehensive animal model evaluation will certainly be helpful to our future sepsis research and may provide valuable insights into sepsis models for other researchers.

We thank the reviewer for pointing out the potential limitations of the animal sepsis model. The aforementioned content will be added to the *Discussion* section (please see page 17, the first paragraph, highlighted text).

At #2 I pointed out that it would be interesting to examine bacteriological studies from other organs. The authors have done that and (unsurprisingly) found that indeed there was evidence of systemic infection. They originally pointed to the fact that immune cells infiltrated the lungs but not the other organs, concluding that lungs alone were likely a focus of infection. In fact, their new data contradicts this conclusion but they do not seem to have acknowledged this or provided any potential explanation.

Response:

This is an obvious oversight, and we appreciate the reviewer for pointing it out. In the initial version, we identified T/B cell infiltration primarily in the lungs through bulk RNA-sequencing, with limited T/B cell infiltration observed in the liver and kidneys. However, after supplementing with bacterial detection, we found substantial bacterial presence in the lungs, liver, and kidneys. This indicates that the systemic infection originating from abdominal infection due to intestinal perforation can not only affect the lungs but also other major organs including the liver and kidneys. In the revised manuscript,

we made corresponding revisions in the following places:

1. On page 7, paragraph 2, we revised the previous description as follows and highlighted it:

"...indicating that T/B cells may mainly infiltrate the lungs, with limited presence in the liver and kidneys. On the other hand, as a pan marker of myeloid cells, the expression of CD11b in the kidney, liver and lung increased stepwise. These results suggest that T/B cells may mainly infiltrate the lungs when sepsis occurs, while myeloid cells migrate to all major organs, including the kidneys, liver and lungs, and the degree of migration increased with the severity of infection".

2. On page 11, the 2nd paragraph, we revised the previous description as follows and highlighted it:

"This indicates the host's battle against pathogens is not limited to the initial infection site, but may spread to major organs including the lungs, liver, and kidneys and cause multi-organ dysfunction".

3. On page 16, the 2nd paragraph, we revised the previous description as follows and highlighted it:

"Acute organ injury (lung, kidney, liver, etc.) is common in sepsis, but its mechanism remains unclear. According to the classic sepsis pathogenesis theory, such injury is likely caused by a systemic inflammatory response. This theory suggests that even without direct infection, the over-activated immune response triggered by the primary infection site (e.g., CLP-induced abdominal infection) may cause "collateral damage" to organs beyond the initial infection site. However, bacterial detection of the lung, kidney, and liver revealed high bacterial burdens, indicating that CLP-induced systemic infection may spread to these organs and that the organ injuries may be because these organs themselves have become "battlefields" rather than "collateral damage" caused by the immune system. This may be a potential mechanism to explain multi-organ injuries in sepsis".

The clinical data (#5) remain something of a concern. Underlying this component of the work is that they can confidently separate septic patients from those with "mild" infection and from normal or trauma controls. However, careful examination of the Supplementary Table reveals, for example, that there are many cells with missing data, the use of "ok" in the PaO₂/FiO₂ column (what does that mean?), examples where the PaO₂/FiO₂ ratio is at a level which would be considered septic but the patient is not so designated (e.g. patient 93), incorrect entries (patient 69), inconsistent PCT data,

inadequate bacteriological data etc. The fact that patients were not sequential is relevant because it implies that they were "chosen" in some way, which reduces the significance of the findings. The Table has units but no normal values/ranges for the various criteria. I am not wholly confident that these data allow the subsequent conclusions regarding the use of cell markers to identify septic patients, which in my view would have been better done as a separate, properly designed study. One of the reasons that I suggested that it would be informative to include some "disease controls" (i.e. trauma patients) was in order to compare their gene expression profiles with those of the infected patients. Very interestingly, the authors new data do indeed show that at least in some cases the profiles were the same in trauma and infected patients. While they have provided the data in Figure 4 they have not commented on it at all, a particularly significant absence given that this was one of the cornerstones of their paper.

Response:

We thank the reviewer for pointing out the omissions in the Supplementary Table of patient clinical information. Indeed, certain details within the table require further refinement, as listed below:

1. In the PaO₂/FiO₂ (mmHg) column, "OK" means it is in the normal range, that is, ≥ 400 was recorded as "OK" by the researcher in charge of recording at that time; in the Glasgow score (GCS) column, "OK" means ≥ 15 . In the revised version, we marked all "OK" with a unified value, that is, ≥ 400 or ≥ 15 , to avoid misunderstanding.
2. In the column "Fractional concentration of inspired oxygen (FiO₂)", patients who were not receiving oxygen or ventilator therapy were all 21% (oxygen content in the atmosphere), which was previously left blank in the table. In the revised version, these blanks are all marked as 21%.
3. For patient #93, the oxygenation index was less than 400, which was recorded as 1 point; platelets ≥ 150 , which was recorded as 0 points; bilirubin < 20 , which was recorded as 0 points; mean arterial pressure ≥ 70 , which was recorded as 0 points; GCS score ≥ 15 , which was recorded as 0 points; and creatinine < 110 , which was recorded as 0 points. Therefore, the patient's SOFA score was 1 and was not diagnosed with sepsis.
4. The third column of patient #69 was mistakenly filled in the text, and we thank the reviewer for pointing out this error.

5. The normal value range for each criteria has been marked in the table (below the criteria title).
6. PCT is helpful in distinguishing infected patients from noninfected patients, and the existing evidence supports the use of PCT as a biomarker to improve the diagnosis of bacterial infections and guide antibiotic therapy. However, the power of PCT is not sufficient to predict the sepsis prognosis, and a significant correlation between high or low PCT levels and the final outcome of survival or death was not observed (Aloisio *et al*, 2019; Molano-Franco *et al*, 2023; Ryoo *et al*, 2019; Tang *et al*, 2007). Therefore, for an individual patient, predicting the prognosis based on the PCT value is difficult. Here we do not presume whether the PCT value is correlated with the diagnosis of sepsis or death, but only record its value for subsequent analysis.
7. In the revised version, we tracked the patients by their medical record numbers and added the available microbiological test results to Supplementary Table 1 (in the revised version, the table name was changed to Appendix Table 1). These patients came from the emergency department, so some with mild symptoms were discharged quickly and did not undergo microbiological testing (the symbol "\" indicates that the patient did not undergo microbiological testing until discharge). We agree that bacteriological test results are important for clinical judgment, nevertheless, it is difficult to get all emergency patients to undergo bacterial culture testing.
8. Each patient's specific number is a unique hospitalization number, for example, patient #1 is 0034592996, patient #2 is 0034595247, etc. The record numbers 1#, 2#, etc. are the numbers assigned to the patients by the researchers who handled them at the time. Since some patients were initially considered infected but were later excluded by clinicians based on WBC or other results, the record numbers are not continuous. In this sense, these patients were indeed selected based on whether they were infected or not. But this is not a subjective selection of some infected patients and the exclusion of others. It's just that the patients who were assigned numbers were later determined to be non-infected and therefore were not included.

We agree with the reviewer that this data does not allow us to conclude that these immune markers can distinguish patients with sepsis from those with common infections, but can only say that they may have the potential to be used in this way in the future. We have made corresponding changes to the text where this statement is mentioned (page 3, the last paragraph; page 8, subheading; page 19, the 2nd

paragraph; highlighted text). It can be considered as a preliminary validation experiment. As the reviewer suggested, a separate, well-designed, large-sample study is warranted to evaluate whether these immune markers can be used to differentiate sepsis from infection. We have initiated a study to retrospectively validate these markers in public sequencing datasets, followed by a multicenter prospective study to investigate their diagnostic and predictive performance.

Indeed, we presented test results of the immune markers in trauma patients in the results, but did not elaborate on them in the discussion, which is indeed an omission. In critically patients with severe trauma leading to multiple organ dysfunction (SOFA>2), some similar characteristics were also shown, manifested as a significant decrease in T/B cell-related markers such as CD8A/CD5/CD19, suggesting that trauma, like severe infection, may cause acute suppression of adaptive immunity, making the host more vulnerable to pathogens and further worsening the patient's condition. We have added the above to the *Discussion* section, please see page 15, the 2nd paragraph, highlighted text.

The authors have acknowledged (#6) that it is slightly odd that IL-10 levels were not raised. I am not completely persuaded by their explanation, which in any case does not appear in the Discussion. I think that the recent paper I identified (#7) should have been cited in the revised paper.

Response:

We thank the reviewer for pointing out that the result of IL-10 detection should be discussed and a recent paper mentioned in comment #7 (Liu *et al*, 2024) should be cited. IL-10 has been reported to be involved in the immunosuppression of sepsis (Zhou *et al*, 2024). However, according to the scRNA-Seq results of the CLP mouse model and the human scRNA-Seq public dataset, the mRNA level of IL-10 in PBMCs of healthy volunteers and sepsis patients was low and did not show significant differences. This may be because IL-10, as a secretory cytokine, is released into the blood upon sepsis onset but not continuously produced in immune cells, so obvious changes in the mRNA level was not observed. The possible role of immunosuppressive factors such as IL-10 in sepsis requires more researches. The aforementioned discussion has been added to the *Discussion* section, please see page 15, paragraph 2, highlighted text. The reference (Zhou *et al*, 2024) has been cited in page 11, the 2nd paragraph.

References:

- Aloisio E, Dolci A, Panteghini M (2019) Procalcitonin: Between evidence and critical issues. *Clin Chim Acta* 496: 7-12
- Liu B, Li F, Wang Y, Gao X, Li Y, Wang Y, Zhou H (2024) APP-CD74 axis mediates endothelial cell-macrophage communication to promote kidney injury and fibrosis. *Front Pharmacol* 15: 1437113
- Mathiak G, Szewczyk D, Abdullah F, Ovadia P, Feuerstein G, Rabinovici R (2000) An improved clinically relevant sepsis model in the conscious rat. *Critical care medicine* 28: 1947-1952
- Molano-Franco D, Arevalo-Rodriguez I, Muriel A, Del Campo-Albendea L, Fernandez-Garcia S, Alvarez-Mendez A, Simancas-Racines D, Viteri A, Sanchez G, Fernandez-Felix B *et al* (2023) Basal procalcitonin, C-reactive protein, interleukin-6, and presepsin for prediction of mortality in critically ill septic patients: a systematic review and meta-analysis. *Diagn Progn Res* 7: 15
- Ryoo SM, Han KS, Ahn S, Shin TG, Hwang SY, Chung SP, Hwang YJ, Park YS, Jo YH, Chang HL *et al* (2019) The usefulness of C-reactive protein and procalcitonin to predict prognosis in septic shock patients: A multicenter prospective registry-based observational study. *Sci Rep* 9: 6579
- Tang BM, Eslick GD, Craig JC, McLean AS (2007) Accuracy of procalcitonin for sepsis diagnosis in critically ill patients: systematic review and meta-analysis. *Lancet Infect Dis* 7: 210-217
- Zhou X, Liu C, Xu Z, Song J, Jin H, Wu H, Cheng Q, Deng W, He D, Yang J *et al* (2024) Combining host immune response biomarkers and clinical scores for early prediction of sepsis in infection patients. *Ann Med* 56: 2396569

Dear Prof. Zhang,

Thank you for the submission of your further revised manuscript to our editorial offices. I have now received the report from the referee that was asked to re-evaluate the study, you will find below. As you will see, the referee now fully supports the publication of the study in EMBO reports. The referee has a final comment. Please revise the abstract accordingly.

Before I can proceed with formal acceptance, I have these editorial requests I ask you to address in a final revised manuscript:

- I would suggest a more active title, e.g.:

CD47-amyloid- β -CD74 signaling triggers adaptive immunosuppression in sepsis

- Please provide the abstract written in present tense throughout.

- Please remove the ORCID IDs from the title page. Please find instructions on how to link the ORCID ID to the account in our manuscript tracking system in our Author guidelines:

<http://www.embopress.org/page/journal/14693178/authorguide#authorshipguidelines>

Please do this for the three co-corresponding authors.

- Please order the manuscript sections like this, using these names:

Title page - Abstract - Keywords - Introduction - Results - Discussion - Methods - Data availability section - Acknowledgements - Disclosure and Competing Interests Statement - References - Figure legends - Expanded View Figure legends

- Most figures, in particular Figures 2,3,6,7 and 8, are extremely crowded, and some of the smaller panels won't show very well in the final online version of the paper. Please re-assemble these figures, remove some data, and move these to the Appendix (as further Appendix Figures or adding the information to existing Appendix figures). Please then also update all the call-outs in the manuscript text accordingly. See also our guide for figure preparation:

http://www.embopress.org/sites/default/files/EMBOPress_Figure_Guidelines_061115.pdf

- Please remove now the referee access information from the Data Availability section and make sure the datasets are public latest upon online publication of the paper. Moreover, please remove the sentence 'All the data that support the findings of this study are available from the corresponding author upon reasonable request' from the DAS.

- Appendix Figure S13 is a table. Please name this Appendix Table S1, move it below the figures and update the callouts.

- The Appendix Table uploaded (patient data) is a datasets. Please upload this as dataset file using the nomenclature Dataset EV1. Please put a legend on the first TAB of the excel file and update all callouts accordingly.

- Please add scale bars of similar style and thickness to microscopic images (main, EV and Appendix figures), using clearly visible black or white bars (depending on the background). Please place these in the lower right corner of the images themselves. Please do not write on or near the bars in the image but define the size in the respective figure legend. Moreover, scale bars are missing in panel EV4B and S15B.

- Please make sure that all figure panels are called out separately and sequentially. Presently, there seems to be no callout for Appendix Figure S8. Please check. A 'Supplementary Table 1' is called out. Please use the correct callout.

- Please check again that the number "n" for how many independent experiments were performed, their nature (biological versus technical replicates), the bars and error bars (e.g. SEM, SD) and the test used to calculate p-values is indicated in the respective figure legends (main, EV and Appendix Figures). Please also check that all the p-values are explained in the legend, and that these fit to those shown in the figure. Please provide statistical testing where applicable. Please avoid the phrase 'independent experiment', but clearly state if these were biological or technical replicates. Please also indicate (e.g. with n.s.) if testing was performed, but the differences are not significant. In case n=2, please show the data as separate datapoints without error bars and statistics. See also:

<http://www.embopress.org/page/journal/14693178/authorguide#statisticalanalysis>

If n<5, please show single datapoints for diagrams. Presently, several blots show only partial statistics or 'n.s.' is missing. Moreover:

- Please note that, figure 2N is not provided in the manuscript, however legend for the same is provided. Kindly rectify this.

- Please note that the exact p values are not provided in the legends of figures 3A-P,T, U, X; 4A-I; 6C, 7N, 8D, F; EV2 A, B; EV4 C; EV5

- Please indicate what */ **/ ***/ **** represents; if this represents p value(s), please indicate the statistical test used and where appropriate, and the exact p value in the legend(s) of figure(s) 5D

- Please indicate the statistical test used for data analysis in the legends of figures 5E, 6C
- Please note that information related to n is missing in the legends of figures 1C, 5B, D;
- Although 'n' is provided, please describe the nature of entity for 'n' in the legends of figures 3A-X; 5E, G; 5E, G; 6C, E-H; 7E-L, N; 8D, F, G-P; EV4 C, EV5
- Please note that the error bars are not defined in the legends of figures 2M, 5E, G; 6C
- Please note that the measure of center for the error bars needs to be defined in the legends of figures 2F-K; 3A-X; 4A-I; 5B, D; 6E-H; 7E-L, N; 8D, F, G-P; EV2 A, B; EV4 C, EV5.

- Please remove the Reagents & Tools table from the manuscript text file. This should be provided only as separate file. Please also add callouts for the table in the methods section were appropriate.

- Please remove the uploaded 'Table 1' (identical to the Author Checklist) and make sure the Author Checklist is uploaded only once.

- The following information needs to be moved from the references to the DAS:
[Dataset] NCBI Gene Expression Omnibus, GSE151263, <https://www.ncbi.nlm.nih.gov/geo/query/acc.cgi?acc=GSE151263>

- There seems to be a re-use of panels between Figure 8C and Appendix figure S19 and Figure 8E and Appendix figure S20. Or it seems the same data is shown in 8C/S19 and 8E/S20. Is that necessary? If the reuse is intentional, then please explain and mention this in the respective figure legends.

In addition, I would need from you uploaded separately:

Best,

Referee #2:

The authors have provided constructive and robust responses to my comments, for which I thank them. There is one final minor point, that in the Abstract they still write: "Clinical data showed that these adaptive immunity-related genes may be used to distinguish patients with sepsis from those with common infections". Elsewhere in the paper they have appropriately toned this down to reflect that they "...may have the potential to distinguish..." and I think this formulation should also appear in the abstract.

All editorial and formatting issues were resolved by the authors.

Prof. Wei Zhang
West China Hospital, Sichuan University
Institute of Critical Care Medicine
No. 2222, Frontier Medical Center, Xin Chuan Road, Zhong He Street
Chengdu, Sichuan 610212
China

Dear Prof. Zhang,

I am very pleased to accept your manuscript for publication in the next available issue of EMBO reports. Thank you for your contribution to our journal.

Yours sincerely,
